# Non-asymptotic Global Convergence Analysis of BFGS with the Armijo-Wolfe Line Search

**Qiujiang Jin**
ECE, UT Austin
qiujiangjin0@gmail.com

**Ruichen Jiang**
ECE, UT Austin
rjiang@utexas.edu

**Aryan Mokhtari**
ECE, UT Austin
mokhtari@austin.utexas.edu

## Abstract

In this paper, we present the first explicit and non-asymptotic global convergence rates of the BFGS method when implemented with an inexact line search scheme satisfying the Armijo-Wolfe conditions. We show that BFGS achieves a global linear convergence rate of $(1-\frac{1}{\kappa})^t$ for $\mu$-strongly convex functions with $L$-Lipschitz gradients, where $\kappa = \frac{L}{\mu}$ represents the condition number. Additionally, if the objective function's Hessian is Lipschitz, BFGS with the Armijo-Wolfe line search achieves a linear convergence rate that depends solely on the line search parameters, independent of the condition number. We also establish a global superlinear convergence rate of $\mathcal{O}((\frac{1}{t})^t)$. These global bounds are all valid for any starting point $x_0$ and any symmetric positive definite initial Hessian approximation matrix $B_0$, though the choice of $B_0$ impacts the number of iterations needed to achieve these rates. By synthesizing these results, we outline the first global complexity characterization of BFGS with the Armijo-Wolfe line search. Additionally, we clearly define a mechanism for selecting the step size to satisfy the Armijo-Wolfe conditions and characterize its overall complexity.

## 1 Introduction

In this paper, we focus on solving the following unconstrained convex minimization problem

$$\min_{x\in\mathbb{R}^d} f(x), \tag{1}$$

where $f : \mathbb{R}^d \to \mathbb{R}$ is strongly convex and twice differentiable. Quasi-Newton methods are among the most popular algorithms for solving this class of problems due to their simplicity and fast convergence. Like gradient descent-type methods, they require only gradient information for implementation, while they aim to mimic the behavior of Newton's method by using gradient information to approximate the curvature of the objective function. There are several variations of quasi-Newton methods, primarily distinguished by their update rules for the Hessian approximation matrices. The most well-known among these include the Davidon-Fletcher-Powell (DFP) method [1, 2], the Broyden-Fletcher-Goldfarb-Shanno (BFGS) method [3–6], the Symmetric Rank-One (SR1) method [7, 8], and the Broyden method [9]. Apart from these classical methods, other variants have also been proposed in the literature, including randomized quasi-Newton methods [10–14], greedy quasi-Newton methods [13–16], and those based on online learning techniques [17, 18]. In this paper, we mainly focus on the global analysis of the BFGS method, arguably the most successful quasi-Newton method in practice.

The classic analyses of BFGS, including [19–28], primarily focused on demonstrating local asymptotic superlinear convergence without addressing an explicit global convergence rate when BFGS is deployed with a line-search scheme. While attempts have been made to establish global convergence for quasi-Newton methods using line search or trust-region techniques in previous studies [8, 29–33], these efforts provided only asymptotic convergence guarantees without explicit global convergence rates, thus not fully characterizing the global convergence rate of classical quasi-Newton methods.

38th Conference on Neural Information Processing Systems (NeurIPS 2024).

In recent years, there have been efforts to characterize the explicit convergence rate of BFGS within a local neighborhood of the solution, establishing a superlinear convergence rate of the form $(\frac{1}{\sqrt{t}})^t$; see, for example, [34–37]. However, these results focus solely on local convergence analysis of BFGS under conditions where the stepsize is consistently set to one, the iterate remains close to the optimal solution, and the initial Hessian approximation matrix meets certain necessary conditions. Consequently, these analyses do not extend to providing a global convergence guarantee. For more details on this subject, we refer the reader to the discussion section in [38].

To the best of our knowledge, only few papers are closely related to our work and establish a global non-asymptotic guarantee for BFGS. In [39], it was shown that BFGS with exact line search achieves a global linear rate of $(1 - \frac{2\mu^3}{L^3}(1 + \frac{\mu \, \mathbf{Tr}(B_0^{-1})}{t})^{-1}(1 + \frac{\mathbf{Tr}(B_0)}{Lt})^{-1})^t$, where $\mu$ is the strong convexity parameter, $L$ is the Lipschitz constant of the gradient, $B_0$ is the initial Hessian approximation matrix, and $\mathbf{Tr}(\cdot)$ denotes the trace of a matrix. After $t = O(d)$ iterations, this rate approaches $(1 - \frac{2\mu^3}{L^3})^t$, which is significantly slower than the convergence rate of gradient descent. Additionally, a recent draft in [40] studied the global convergence of BFGS under an inexact line search. While this work establishes a local superlinear rate, it only shows a global linear rate of the form $(1 - \frac{\mu^2}{L^2})^t$. Hence, both these results fail to prove any global advantage for BFGS over gradient descent. In [38], the authors improved upon [39] by showing a better global linear convergence rate and a faster superlinear rate for BFGS with exact line search. Specifically, for an $L$-Lipschitz and $\mu$-strongly convex function, BFGS initialized with $B_0 = LI$ achieves a global linear rate of $(1 - \frac{\mu^{3/2}}{L^{3/2}})^t$ for $t \geq 1$, while BFGS with $B_0 = \mu I$ achieves the same rate after $d \log \kappa$ iterations. With the additional assumption that the objective's Hessian is Lipschitz, an improved linear rate of $(1 - \frac{\mu}{L})^t$ is achieved after $\mathcal{O}(\kappa)$ iterations when $B_0 = LI$ and after $\mathcal{O}(d \log \kappa + \kappa)$ when $B_0 = \mu I$, matching the rate of gradient descent. A superlinear rate of $(1/\sqrt{t})^t$ was also shown when the number of iterations exceeds specific thresholds.

**Contributions.** In this paper, we analyze the BFGS method combined with the Armijo-Wolfe line search, the most commonly used line search criteria in practical BFGS applications; see, e.g., [41]. For minimizing an $L$-smooth and $\mu$-strongly convex function, we present a global convergence rate of $(1 - \frac{\mu}{L})^t$. To the best of our knowledge, this is the first result demonstrating a global linear convergence rate for BFGS that matches the rate of gradient descent under these assumptions. Furthermore, we show that if the objective function's Hessian is Lipschitz continuous, BFGS with the Armijo-Wolfe line search converges at a linear rate determined solely by the line search parameters and not the problem's condition number, $\kappa = L/\mu$, when the number of iterations is sufficiently large. Finally, we prove a global non-asymptotic superlinear convergence rate of $(h(d, \kappa, C_0)/t)^t$, where $h(d, \kappa, C_0)$ depends on the condition number $\kappa$, the dimension $d$, and the weighted distance between the initial point $x_0$ and the optimal solution $x_*$, denoted by $C_0$. We summarize our results in Table 1. By combining these convergence results, we establish the total iteration complexity of BFGS with the Armijo-Wolfe line search. We also specify the line search complexity by investigating a bisection algorithm for choosing the step size that satisfies the Armijo-Wolfe conditions. Our result is one of the first non-asymptotic analysis characterizing the global convergence complexity of the BFGS quasi-Newton method with an inexact line search.

**Notation.** We denote the $\ell_2$-norm by $\|\cdot\|$, the set of $d \times d$ symmetric positive definite matrices by $\mathbb{S}_{++}^d$, and use $A \preceq B$ to mean $B - A$ is symmetric positive semi-definite. The trace and determinant of a matrix $A$ are represented as $\mathbf{Tr}(A)$ and $\mathbf{Det}(A)$, respectively.

## 2 Preliminaries

In this section, we present the assumptions, notations, and intermediate results useful for the global convergence analysis. First, we state the following assumptions on the objective function $f$.

**Assumption 2.1.** *The function $f$ is twice differentiable and strongly convex with parameter $\mu > 0$.*

**Assumption 2.2.** *The gradient of $f$ is Lipschitz continuous with parameter $L > 0$.*

These assumptions are common in the convergence analysis of quasi-Newton methods. Under these, we show a global linear convergence rate of $O((1 - \frac{\mu}{L})^t)$. To achieve a faster linear convergence rate that is independent of the problem condition number, and a global superlinear rate, we require an additional assumption that the objective function Hessian is Lipschitz continuous, as stated next.

**Assumption 2.3.** *The Hessian of $f$ is Lipschitz continuous with parameter $M > 0$, i.e., for $x, y \in \mathbb{R}^d$, we have $\|\nabla^2 f(x) - \nabla^2 f(y)\| \leq M\|x - y\|$.*

| Initial Matrix | Convergence Phase | Convergence Rate | Starting moment |
|---|---|---|---|
| $B_0$ | Linear phase I | $\left(1 - \frac{1}{\kappa}\right)^t$ | $\Psi(\bar{B}_0)$ |
| $B_0$ | Linear phase II | $\left(1 - \frac{1}{3}\right)^t$ | $\Psi(\tilde{B}_0) + C_0 \Psi(\bar{B}_0) + C_0 \kappa$ |
| $B_0$ | Superlinear phase | $\left(\frac{\Psi(\bar{B}_0) + C_0 \Psi(\bar{B}_0) + C_0 \kappa}{t}\right)^t$ | $\Psi(\tilde{B}_0) + C_0 \Psi(\bar{B}_0) + C_0 \kappa$ |
| $LI$ | Linear phase I | $\left(1 - \frac{1}{\kappa}\right)^t$ | $1$ |
| $LI$ | Linear phase II | $\left(1 - \frac{1}{3}\right)^t$ | $d\kappa + C_0 \kappa$ |
| $LI$ | Superlinear phase | $\left(\frac{d\kappa + C_0 \kappa}{t}\right)^t$ | $d\kappa + C_0 \kappa$ |
| $\mu I$ | Linear phase I | $\left(1 - \frac{1}{\kappa}\right)^t$ | $d \log \kappa$ |
| $\mu I$ | Linear phase II | $\left(1 - \frac{1}{3}\right)^t$ | $(1 + C_0) d \log \kappa + C_0 \kappa$ |
| $\mu I$ | Superlinear phase | $\left(\frac{(1 + C_0) d \log \kappa + C_0 \kappa}{t}\right)^t$ | $(1 + C_0) d \log \kappa + C_0 \kappa$ |

Table 1: Summary of our results for (i) an arbitrary positive definite $B_0$, (ii) $B_0 = LI$, and (iii) $B_0 = \mu I$. Here, $\Psi(A) := \mathbf{Tr}(A) - d - \log \mathbf{Det}(A)$, $\bar{B}_0 = \frac{1}{L} B_0$ and $\tilde{B}_0 = \nabla^2 f(x_*)^{-\frac{1}{2}} B_0 \nabla^2 f(x_*)^{-\frac{1}{2}}$. The last column shows the number of iterations required to achieve the corresponding linear or superlinear convergence phase. For brevity, the absolute constants are dropped.

Note that the above regularity condition on the Hessian assumption is also common for establishing the superlinear convergence rate of quasi-Newton methods [19–28].

**BFGS Update.** Next, we state the general update rule of BFGS. If we denote $x_t$ as the iterate at time $t$, the vector $g_t = \nabla f(x_t)$ as the objective function gradient at $x_t$, and $B_t$ as the Hessian approximation matrix at step $t$, then the update is given by

$$x_{t+1} = x_t + \eta_t d_t, \qquad d_t = -B_t^{-1} g_t, \tag{2}$$

where $\eta_t > 0$ is the step size and $d_t$ is the descent direction. By defining the variable difference $s_t := x_{t+1} - x_t$ and the gradient difference $y_t := \nabla f(x_{t+1}) - \nabla f(x_t)$, we can present the Hessian approximation matrix update for BFGS as follows:

$$B_{t+1} = B_t - \frac{B_t s_t s_t^\top B_t}{s_t^\top B_t s_t} + \frac{y_t y_t^\top}{s_t^\top y_t}. \tag{3}$$

To avoid the costly operation of inverting the matrix $B_t$, one can define the inverse Hessian approximation matrix as $H_t := B_t^{-1}$ and apply the Sherman-Morrison-Woodbury formula to obtain

$$H_{t+1} := \left(I - \frac{s_t y_t^\top}{y_t^\top s_t}\right) H_t \left(I - \frac{y_t s_t^\top}{s_t^\top y_t}\right) + \frac{s_t s_t^\top}{y_t^\top s_t}. \tag{4}$$

It is well-known that for a strongly convex objective function, the Hessian approximation matrices $B_t$ remain symmetric and positive definite if the initial matrix $B_0$ is symmetric positive definite [41]. Therefore, all matrices $B_t$ and $H_t$ are symmetric positive definite throughout this paper.

As mentioned earlier, establishing a global convergence guarantee for BFGS requires pairing it with a line search scheme to select the stepsize $\eta_t$. This paper focuses on implementing BFGS with the Armijo-Wolfe line search, detailed in the following subsection.

**Armijo-Wolfe Line Search.** We consider a stepsize $\eta_t > 0$ that satisfies the Armijo-Wolfe conditions

$$f(x_t + \eta_t d_t) \leq f(x_t) + \alpha \eta_t \nabla f(x_t)^\top d_t, \tag{5}$$

$$\nabla f(x_t + \eta_t d_t)^\top d_t \geq \beta \nabla f(x_t)^\top d_t, \tag{6}$$

where $\alpha$ and $\beta$ are the line search parameters, satisfying $0 < \alpha < \beta < 1$ and $0 < \alpha < \frac{1}{2}$. The condition in (5) is the Armijo condition, ensuring that the step size $\eta_t$ provides a sufficient decrease in the objective function $f$. The condition in (6) is the curvature condition, which guarantees that the slope $\nabla f(x_t + \eta_t d_t)^\top d_t$ at $\eta_t$ is not strongly negative, indicating that further movement along $d_t$ would significantly decrease the function value. These conditions provide upper and lower bounds on the admissible step size $\eta_t$. In some references, the Armijo-Wolfe line search conditions are known as the weak Wolfe conditions [42, 43]. The procedure for finding $\eta_t$ that satisfies these conditions is described in Section 7. Next lemma presents key properties of the Armijo-Wolfe conditions.

**Lemma 2.1.** *Consider the BFGS method with Armijo-Wolfe inexact line search, where the step size satisfies the conditions in (5) and (6). Then, for any initial point $x_0$ and any symmetric positive definite initial Hessian approximation matrix $B_0$, the following results hold for all $t \geq 0$:*

$$\frac{f(x_t) - f(x_{t+1})}{-g_t^\top s_t} \geq \alpha, \qquad \frac{y_t^\top s_t}{-g_t^\top s_t} \geq 1 - \beta, \qquad and \qquad f(x_{t+1}) \leq f(x_t). \tag{7}$$

**Remark 2.1.** *While in this paper we only focus on the Armijo-Wolfe line search, our results are also valid for some other line search schemes that require stricter conditions. For instance, in the strong Wolfe line search, given $0 < \alpha < \beta < 1$ and $0 < \alpha < \frac{1}{2}$, the required conditions for the step size are*

$$f(x_t + \eta_t d_t) \leq f(x_t) + \alpha \eta_t \nabla f(x_t)^\top d_t, \qquad |\nabla f(x_t + \eta_t d_t)^\top d_t| \leq \beta \nabla f(x_t)^\top d_t,$$

*Indeed, if $\eta_t$ satisfies the strong Wolfe conditions, it also satisfies the Armijo-Wolfe conditions.*

*Another commonly employed line search scheme is Armijo–Goldstein, which imposes the conditions*

$$-c_1 \eta_t \nabla f(x_t)^\top d_t \leq f(x_t) - f(x_t + \eta_t d_t) \leq -c_2 \eta_t \nabla f(x_t)^\top d_t,$$

*with $0 < c_1 \leq c_2 < 1$. The lower bound on $f(x_t) - f(x_t + \eta_t d_t)$ in the Armijo–Goldstein line search indicates that $\eta_t$ satisfies the sufficient decrease condition in (5) required for the Armijo-Wolfe conditions, with $\alpha = c_1$. Moreover, given the convexity of $f$, the upper bound on $f(x_t) - f(x_t + \eta_t d_t)$ in the Armijo–Goldstein line search suggests $-\eta_t \nabla f(x_t + \eta_t d_t)^\top d_t \leq f(x_t) - f(x_t + \eta_t d_t) \leq -c_2 \eta_t \nabla f(x_t)^\top d_t$. Thus, $\eta_t$ also meets the curvature condition in (6) required in the Armijo-Wolfe conditions with $\beta = c_2$. Hence, all our results derived under the Armijo-Wolfe line search are also valid for both the strong Wolfe line search and the Armijo–Goldstein line search.*

## 3 Convergence Analysis

In this section, we present our theoretical framework for analyzing the global linear convergence rates of BFGS with the Armijo-Wolfe line search scheme. To start, we introduce some necessary definitions and notations. We define the average Hessian matrices $J_t$ and $G_t$ as

$$J_t := \int_0^1 \nabla^2 f(x_t + \tau(x_{t+1} - x_t))d\tau, \qquad G_t := \int_0^1 \nabla^2 f(x_t + \tau(x_* - x_t))d\tau. \tag{8}$$

Further, for measuring the suboptimality of the iterates we define the sequence $C_t$ as

$$C_t := \frac{M}{\mu^{\frac{3}{2}}} \sqrt{2(f(x_t) - f(x_*))}, \qquad \forall t \geq 0, \tag{9}$$

where $M$ is the Lipschitz constant of the Hessian defined in Assumption 2.3 and $\mu$ is the strong convexity parameter introduced in Assumption 2.1.To analyze the dynamics of the Hessian approximation matrices $\{B_t\}_{t=0}^{+\infty}$, we use the function $\Psi(A)$

$$\Psi(A) := \mathbf{Tr}(A) - d - \log \mathbf{Det}(A), \tag{10}$$

well-defined for any $A \in \mathbb{S}_{++}^d$. It was introduced in [32] to capture the discrepancy between $A$ and the identity matrix $I$. Note that $\Psi(A) \geq 0$ for any $A \in \mathbb{S}_{++}^d$ and $\Psi(A) = 0$ if and only if $A = I$.

Before we start convergence analysis, given any weight matrix $P \in \mathbb{S}_{++}^d$, we define the weighted versions of the vectors $g_t$, $s_t$, $y_t$, $d_t$ and the matrix $B_t$, $J_t$ as

$$\hat{g}_t = P^{-\frac{1}{2}} g_t, \qquad \hat{s}_t = P^{\frac{1}{2}} s_t, \qquad \hat{y}_t = P^{-\frac{1}{2}} y_t, \qquad \hat{d}_t = P^{\frac{1}{2}} d_t. \tag{11}$$

$$\hat{B}_t = P^{-\frac{1}{2}} B_t P^{-\frac{1}{2}}, \qquad \hat{J}_t = P^{-\frac{1}{2}} J_t P^{-\frac{1}{2}}. \tag{12}$$

Note that these weighted matrices and vectors preserve many properties of their unweighted counterparts. For instance, two of these main properties are $\hat{g}_t^\top \hat{s}_t = g_t^\top s_t$ and $\hat{y}_t^\top \hat{s}_t = y_t^\top s_t$. Similarly, the update for the weighted version of Hessian approximation matrices closely mirrors the update of their unweighted counterparts, as noted in the following expression:

$$\hat{B}_{t+1} = \hat{B}_t - \frac{\hat{B}_t \hat{s}_t \hat{s}_t^\top \hat{B}_t}{\hat{s}_t^\top \hat{B}_t \hat{s}_t} + \frac{\hat{y}_t \hat{y}_t^\top}{\hat{s}_t^\top \hat{y}_t}, \qquad \forall t \geq 0. \tag{13}$$

Finally, we define a crucial quantity, $\hat{\theta}_t$, which measures the angle between the weighted descent direction and the negative of the weighted gradient direction, satisfying

$$\cos(\hat{\theta}_t) = \frac{-\hat{g}_t^\top \hat{s}_t}{\|\hat{g}_t\| \|\hat{s}_t\|}. \tag{14}$$

## 3.1 Intermediate Results

In this section, we present our framework for analyzing the convergence of BFGS with an inexact line search. We first characterize the relationship between the function value decrease at each iteration and key quantities, including the angle $\hat{\theta}_t$ defined in (14).

**Proposition 3.1.** *Let $\{x_t\}_{t\geq 0}$ be the iterates generated by BFGS. Recall the definitions of weighted vectors in (11). Then, for any weight matrix $P$ and for all $t \geq 1$, we have*

$$\frac{f(x_t) - f(x_*)}{f(x_0) - f(x_*)} \leq \left(1 - \left(\prod_{i=0}^{t-1} \hat{p}_i \hat{q}_i \hat{n}_i \frac{\cos^2(\hat{\theta}_i)}{\hat{m}_i}\right)^{\frac{1}{t}}\right)^t. \tag{15}$$

*where $\hat{p}_t$, $\hat{q}_t$, $\hat{m}_t$ and $\hat{n}_t$ are defined as*

$$\hat{p}_t := \frac{f(x_t) - f(x_{t+1})}{-\hat{g}_t^\top \hat{s}_t}, \quad \hat{q}_t := \frac{\|\hat{g}_t\|^2}{f(x_t) - f(x_*)}, \quad \hat{m}_t := \frac{\hat{y}_t^\top \hat{s}_t}{\|\hat{s}_t\|^2}, \quad \hat{n}_t = \frac{\hat{y}_t^\top \hat{s}_t}{-\hat{g}_t^\top \hat{s}_t}. \tag{16}$$

This result shows the convergence rate of BFGS with Armijo-Wolfe line search depends on four products: $\prod_{i=0}^{t-1} \hat{p}_i$, $\prod_{i=0}^{t-1} \hat{q}_i$, $\prod_{i=0}^{t-1} \hat{n}_i$, and $\prod_{i=0}^{t-1} \frac{\cos^2(\hat{\theta}_i)}{\hat{m}_i}$. To establish an explicit rate, we need lower bounds on these products. Lemma 2.1 shows that the lower bounds for $\prod_{i=0}^{t-1} \hat{p}_i$ and $\prod_{i=0}^{t-1} \hat{n}_i$ depend on the inexact line search parameters $\alpha$ and $\beta$. We will further prove that if the unit step size $\eta_t = 1$ satisfies the Armijo-Wolfe conditions, better lower bounds can be obtained for these products. The lower bounds for $\prod_{i=0}^{t-1} \hat{q}_i$ and $\prod_{i=0}^{t-1} \frac{\cos^2(\hat{\theta}_i)}{\hat{m}_i}$ were established in previous work [38] as presented in Appendix D. Specifically, the bounds for $\prod_{i=0}^{t-1} \hat{q}_i$ depend on the choice of the weight matrix, which varies in different sections of the paper, requiring separate bounds for each case. However, the bound for $\prod_{i=0}^{t-1} \frac{\cos^2(\hat{\theta}_i)}{\hat{m}_i}$ does not require separate treatment. This is explicitly established in Proposition D.1, a classical result, as discussed in [41, Section 6.4]. We build all our linear and superlinear results by establishing different bounds on the terms in (15).

## 4 Global Linear Convergence Rates

Building on the tools introduced in Section 3, we establish explicit global linear convergence rates for BFGS with the Armijo-Wolfe line search, requiring only the strong convexity and gradient Lipschitz conditions from Assumptions 2.1 and 2.2. Our proof leverages the fundamental inequality in (15) from Proposition 3.1 and lower bounds on the terms that appear in the contraction factor. Here, we set the weight matrix $P$ to $P = LI$ and hence define the initial weighted matrix $\bar{B}_0$ as $\bar{B}_0 = \frac{1}{L} B_0$. The following theorem presents our first global linear convergence rate of BFGS for any $B_0 \in \mathbb{S}_{++}^d$.

**Theorem 4.1.** *Suppose Assumptions 2.1 and 2.2 hold. Let $\{x_t\}_{t\geq 0}$ be the iterates generated by BFGS, where the step size satisfies the Armijo-Wolfe conditions in (5) and (6). For any initial point $x_0 \in \mathbb{R}^d$ and any initial Hessian approximation matrix $B_0 \in \mathbb{S}_{++}^d$, we have*

$$\frac{f(x_t) - f(x_*)}{f(x_0) - f(x_*)} \leq \left(1 - e^{-\frac{\Psi(\bar{B}_0)}{t}} \frac{2\alpha(1-\beta)}{\kappa}\right)^t, \quad \forall t \geq 1. \tag{17}$$

**Remark 4.1.** *In [38], the authors analyzed BFGS with exact line search and established a global linear rate of $(1 - e^{-\frac{\Psi(\bar{B}_0)}{t}} \frac{1}{\kappa(1+\sqrt{\kappa})})^t$. In comparison, our result in (17) achieves a faster linear rate by eliminating the $\sqrt{\kappa}$ factor in the denominator. This improvement arises from using the Armijo-Wolfe conditions. Specifically, under these conditions, we show $\frac{f(x_t) - f(x_{t+1})}{-g_t^\top s_t} \geq \alpha$ as shown in Lemma 2.1, where $\alpha \in (0, 1/2)$ is a line search parameter. In contrast, using exact line search, the authors in [38] proved that $\frac{f(x_t) - f(x_{t+1})}{-g_t^\top s_t} \geq \frac{2}{\sqrt{\kappa}+1}$, thus leading to the extra $\sqrt{\kappa}$ factor in their rate.*

From Theorem 4.1, we observe that the linear convergence rate is determined by the quantity $\Psi(\bar{B}_0)$. Thus, to simplify our bounds, we consider two different initializations: $B_0 = LI$ and $B_0 = \mu I$.

**Corollary 4.2.** *Suppose Assumptions 2.1 and 2.2 hold, $\{x_t\}_{t\geq 0}$ are generated by BFGS with step size satisfying the Armijo-Wolfe conditions in (5) and (6), and $x_0 \in \mathbb{R}^d$ is an arbitrary initial point.*

- *If the initial Hessian approximation matrix is set as $B_0 = LI$, then for any $t \geq 1$*

$$\frac{f(x_t) - f(x_*)}{f(x_0) - f(x_*)} \leq \left(1 - \frac{2\alpha(1-\beta)}{\kappa}\right)^t. \tag{18}$$

- *If the initial Hessian approximation matrix is set as $B_0 = \mu I$, then for any $t \geq 1$ we have* $\frac{f(x_t)-f(x_*)}{f(x_0)-f(x_*)} \leq (1 - e^{-\frac{d\log\kappa}{t}} \frac{2\alpha(1-\beta)}{\kappa})^t$. *Moreover, for $t \geq d\log\kappa$, we have*

$$\frac{f(x_t) - f(x_*)}{f(x_0) - f(x_*)} \leq \left(1 - \frac{2\alpha(1-\beta)}{3\kappa}\right)^t. \tag{19}$$

Corollary 4.2 shows that when initialized with $B_0 = LI$, BFGS achieves a linear rate of $\mathcal{O}((1 - \frac{1}{\kappa})^t)$ from the first iteration, matching the rate of gradient descent. It also indicates that initializing with $B_0 = \mu I$ achieves a similar rate but after $d\log\kappa$ iterations. While this suggests a preference for initializing with $B_0 = LI$, subsequent analysis reveals that with enough iterations, BFGS with either initialization can attain a faster linear rate independent of $\kappa$. In some cases, starting with $B_0 = \mu I$ may lead to fewer total iterations to achieve this faster rate. We will explore this trade-off later.

# 5 Condition Number Independent Linear Convergence Rates

In this section, we improve the previous results and establish a non-asymptotic, condition number-free global linear convergence rate for BFGS with the Armijo-Wolfe line search. This requires the additional assumption that the Hessian is Lipschitz continuous. Our analysis builds on the previous methodology but uses $P = \nabla^2 f(x_*)$ instead of $P = LI$ to prove the condition number-independent global linear rate. Thus, the weighted initial matrix $\tilde{B}_0$ is $\nabla^2 f(x_*)^{-\frac{1}{2}} B_0 \nabla^2 f(x_*)^{-\frac{1}{2}}$. Next, we present a general global convergence bound for any initial Hessian approximation $B_0 \in \mathbb{S}_{++}^d$.

**Proposition 5.1.** *Suppose Assumptions 2.1, 2.2 and 2.3 hold. Let $\{x_t\}_{t\geq 0}$ be the iterates generated by BFGS with the step size satisfying the Armijo-Wolfe conditions in (5) and (6). Recall the definition of $C_t$ in (9) and $\Psi(\cdot)$ in (10). For any initial point $x_0 \in \mathbb{R}^d$ and any initial Hessian approximation matrix $B_0 \in \mathbb{S}_{++}^d$, the following result holds:*

$$\frac{f(x_t) - f(x_*)}{f(x_0) - f(x_*)} \leq \left(1 - 2\alpha(1-\beta)e^{-\frac{\Psi(\tilde{B}_0) + 3\sum_{i=0}^{t-1} C_i}{t}}\right)^t, \quad \forall t \geq 1.$$

Proposition 5.1 demonstrates that the convergence rate of BFGS with the Armijo-Wolfe line search is influenced by $\Psi(\tilde{B}_0)$ and the sum $\sum_{i=0}^{t-1} C_i$. The first term $\Psi(\tilde{B}_0)$ is a constant that depends on our choice of the initial Hessian approximation matrix $B_0$. The second term $\sum_{i=0}^{t-1} C_i$ can also be upper bounded using the non-asymptotic global linear convergence rate provided in Theorem 4.1.

**Theorem 5.2.** *Suppose Assumptions 2.1, 2.2 and 2.3 hold, and let $\{x_t\}_{t\geq 0}$ be the iterates generated by BFGS with the Armijo-Wolfe line search in (5) and (6). Then, for any initial point $x_0 \in \mathbb{R}^d$ and any initial Hessian approximation $B_0 \in \mathbb{S}_{++}^d$, if $t \geq \Psi(\tilde{B}_0) + 3C_0\Psi(\bar{B}_0) + \frac{9}{\alpha(1-\beta)}C_0\kappa$, we have*

$$\frac{f(x_t) - f(x_*)}{f(x_0) - f(x_*)} \leq \left(1 - \frac{2\alpha(1-\beta)}{3}\right)^t. \tag{20}$$

This result shows that when the number of iterations meets $t \geq \Psi(\tilde{B}_0) + 3C_0\Psi(\bar{B}_0) + \frac{9}{\alpha(1-\beta)}C_0\kappa$, BFGS with Armijo-Wolfe conditions achieves a condition number-independent linear rate. The choice of $B_0$ is critical as it influences the required iterations through $\tilde{B}_0 = \nabla^2 f(x_*)^{-\frac{1}{2}} B_0 \nabla^2 f(x_*)^{-\frac{1}{2}}$ and $\bar{B}_0 = \frac{1}{L}B_0$. Different choices of $B_0$ affect $\Psi(\tilde{B}_0) + 3C_0\Psi(\bar{B}_0)$ and thus the number of iterations needed for condition-free linear convergence. While optimizing $B_0$ to minimize $\Psi(\tilde{B}_0) + 3C_0\Psi(\bar{B}_0)$ is possible, we focus on two practical initialization schemes: $B_0 = LI$ and $B_0 = \mu I$.

**Corollary 5.3.** *Suppose that Assumptions 2.1, 2.2 and 2.3 hold. Let $\{x_t\}_{t\geq 0}$ be the iterates generated by the BFGS method, where the step size satisfies the Armijo-Wolfe conditions in (5) and (6), and $x_0 \in \mathbb{R}^d$ as an arbitrary initial point. Then, given the result in Theorem 5.2, we have*

- *If we set $B_0 = LI$, the rate in (20) holds for $t \geq d\kappa + \frac{9}{\alpha(1-\beta)}C_0\kappa$,*

- *If we set $B_0 = \mu I$, the rate in (20) holds for $t \geq (1 + 3C_0)d\log\kappa + \frac{9}{\alpha(1-\beta)}C_0\kappa$.*

Based on Corollary 5.3, if $C_0 \ll \kappa$, or equivalently $f(x_0) - f(x_*) \ll \frac{L^2\mu}{M^2}$, then BFGS with $B_0 = \mu I$ requires less iterations to achieve the condition number-independent linear convergence rate.

# 6 Global Superlinear Convergence Rates

In this section, we present our global superlinear result. Consider the definition $\tilde{B}_0 = \nabla^2 f(x_*)^{-\frac{1}{2}} B_0 \nabla^2 f(x_*)^{-\frac{1}{2}}$ as well as the definition of $\rho_t$ which is given by

$$\rho_t := \frac{-g_t^\top d_t}{\|\tilde{d}_t\|^2}, \qquad \tilde{d}_t := \nabla^2 f(x_*)^{\frac{1}{2}} d_t, \qquad \forall t \geq 0. \tag{21}$$

To motivate, let us briefly discuss why we are only able to show a linear convergence rate instead of a superlinear rate in Theorem 5.2. By inspecting the proof, we observe that the bottleneck is due to the lower bounds on $\hat{p}_t$ and $\hat{n}_t$: we used $\hat{p}_t \geq \alpha$ and $\hat{n}_t \geq 1 - \beta$ from Lemma 2.1, which leads to the constant factor $\alpha(1 - \beta)$ in the final linear rate in Theorem 5.2. Thus, to show a superlinear convergence rate, we need to establish tighter lower bounds for $\hat{p}_t$ and $\hat{n}_t$. In the following lemma, we show that if the step size $\eta_t = 1$, we are able to establish such tighter lower bounds.

**Lemma 6.1.** *Recall $\hat{p}_t = \frac{f(x_t) - f(x_{t+1})}{-\hat{g}_t^\top \hat{s}_t}$ and $\hat{n}_t = \frac{\hat{y}_t^\top \hat{s}_t}{-\hat{g}_t^\top \hat{s}_t}$ defined in (16). If the unit step size $\eta_t = 1$ satisfies the Armijo-Wolfe conditions (5) and (6), then we have*

$$\hat{p}_t \geq 1 - \frac{1 + C_t}{2\rho_t}, \qquad \hat{n}_t \geq \frac{1}{(1 + C_t)\rho_t}. \tag{22}$$

In contrast to the constant lower bounds in Lemma 2.1, the lower bounds in (22) depend on $C_t$ and $\rho_t$. Later, we show $C_t \to 0$ and $\rho_t \to 1$. Hence, the lower bounds in (22) approach 1 as the number of iterations increases, enabling us to prove a superlinear rate. That said, the lower bounds in Lemma 6.1 hold only when $\eta_t = 1$. To complete the picture, we need to quantify when and how often the unit step size is selected during BFGS execution. This is addressed in the next lemmas.

**Lemma 6.2.** *Suppose Assumptions 2.1, 2.2, and 2.3 hold and define the constants*

$$\delta_1 := \min\left\{\frac{1}{6}, \sqrt{2(1 - \alpha)} - 1, \frac{1}{\sqrt{1 - \beta}} - 1\right\}, \ \delta_2 := \max\left\{\frac{7}{8}, \frac{1}{\sqrt{2(1 - \alpha)}}\right\}, \ \delta_3 := \frac{1}{\sqrt{1 - \beta}}, \tag{23}$$

*which satisfy $0 < \delta_1 < \delta_2 < 1 < \delta_3$. If $C_t \leq \delta_1$ and $\delta_2 \leq \rho_t \leq \delta_3$, then $\eta_t = 1$ satisfies the Armijo-Wolfe conditions (5) and (6).*

Lemma 6.2 shows that when $C_t \leq \delta_1$ and $\rho_t$ falls within the interval $[\delta_2, \delta_3]$, the step size $\eta_t = 1$ is admissible and meets the Armijo-Wolfe conditions. Note that by the linear convergence result in Theorem 4.1, the first condition on $C_t$ will be satisfied when $t$ is sufficiently large. Additionally, using Proposition G.2 in the Appendix, we can show that the second condition on $\rho_t$ is violated only for a finite number of iterations. These observations are formally presented in the following lemma.

**Lemma 6.3.** *Suppose Assumptions 2.1, 2.2, and 2.3 hold and the iterates $\{x_t\}_{t\geq 0}$ are generated by the BFGS method with step size satisfying the Armijo-Wolfe conditions in (5) and (6). Recall $C_t$ defined in (9), $\Psi(\cdot)$ defined in (10), $\{\delta_i\}_{i=1}^3$ defined in (23) and $\bar{B}_0 = \frac{1}{L} B_0$. We have $C_t \leq \delta_1$ when*

$$t \geq t_0 := \max\left\{\Psi(\bar{B}_0), \frac{3\kappa}{\alpha(1 - \beta)} \log \frac{C_0}{\delta_1}\right\}. \tag{24}$$

*Moreover, if we define $\omega(x) = x - \log(1 + x)$, the size of the set $I = \{t : \rho_t \notin [\delta_2, \delta_3]\}$ is at most*

$$|I| \leq \delta_4\left(\Psi(\tilde{B}_0) + 2C_0\Psi(\bar{B}_0) + \frac{6C_0\kappa}{\alpha(1 - \beta)}\right), \quad where \ \delta_4 := \frac{1}{\min\{\omega(\delta_2 - 1), \omega(\delta_3 - 1)\}}. \tag{25}$$

Lemma 6.3 implies that conditions $C_t \leq \delta_1$ and $\rho_t \in [\delta_2, \delta_3]$ will be satisfied for all but a finite number of iterations. Thus, if the line search always starts by testing the unit step size (as shown in Section 7), we will choose $\eta_t = 1$, and accordingly, the tighter lower bound in Lemma 6.1 will apply for all but a finite number of iterations. By applying these lower bounds along with (15) from Proposition 3.1, we can prove a global superlinear convergence rate, as presented next.

**Remark 6.1.** *Lemmas 6.2 and 6.3 are inspired by the analysis in [40]. Specifically, Lemma 5.10 of [40] characterized the conditions on $C_t$ and $\rho_t$ under which $\eta = 1$ satisfies the Armijo condition (5), and further bounded the number of iterations where these conditions are violated. However, our Lemma 6.2 addresses both the Armijo condition in (5) and the curvature condition in (6), and the arguments appear simpler. Additionally, our proof for the superlinear convergence rate differs from [40]. Their approach analyzed the Dennis-Moré ratio and measured "local" superlinear convergence using the distance $\|\nabla f(x_*)^{\frac{1}{2}}(x_t - x_*)\|$. In contrast, our "global" result is based on the unified framework in Proposition 3.1 and uses the function value gap as a measure of convergence.*

**Theorem 6.4.** *Suppose Assumptions 2.1, 2.2, and 2.3 hold and the iterates $\{x_t\}_{t\geq 0}$ are generated by BFGS with step size satisfying the Armijo-Wolfe conditions in (5) and (6). Recall the definition of $C_t$ in (9), $\Psi(\cdot)$ in (10), $\bar{B}_0 := \frac{1}{L}B_0$, $\tilde{B}_0 := \nabla^2 f(x_*)^{-\frac{1}{2}} B_0 \nabla^2 f(x_*)^{-\frac{1}{2}}$, and $\delta_1, \delta_2, \delta_3, \delta_4$ in (23) and (25). Then, for any $x_0 \in \mathbb{R}^d$ and any $B_0 \in \mathbb{S}_{++}^d$, the following global superlinear result holds:*

$$\frac{f(x_t) - f(x_*)}{f(x_0) - f(x_*)} \leq \left( \frac{\delta_7 \Psi(\tilde{B}_0) + (\delta_6 + \delta_8 C_0)\Psi(\bar{B}_0) + (\frac{3\delta_6}{\alpha(1-\beta)} \log \frac{C_0}{\delta_1} + \frac{3\delta_8}{\alpha(1-\beta)} C_0)\kappa}{t} \right)^t, \quad (26)$$

*where $\{\delta_i\}_{i=5}^8$ defined below are constants that only depend on line search parameters $\alpha$ and $\beta$,*

$$\delta_5 := \frac{\max\{2 + \frac{2}{\delta_2}, 4\delta_3\}}{2\delta_2 - 1 - \delta_1}, \ \delta_6 := \log \frac{1}{2\alpha(1-\beta)}, \ \delta_7 := 1 + \delta_4\delta_6 + \delta_5, \ \delta_8 := 1 + 2\delta_7 + \frac{2\delta_2 - \delta_1 - \log \delta_2}{2\delta_2 - 1 - \delta_1}.$$

The above result shows a global superlinear convergence rate of the form $\mathcal{O}((\frac{C'}{t})^t)$, where $C'$ depends on the condition number $\kappa$, the initial weighted distance $C_0$, and the initial Hessian approximation matrix $B_0$. To simplify the expression, we report the above bound for $B_0 = LI$ and $B_0 = \mu I$.

**Corollary 6.5.** *Suppose Assumptions 2.1, 2.2, and 2.3 hold and the iterates $\{x_t\}_{t\geq 0}$ are generated by the BFGS method with step size satisfying the Armijo-Wolfe conditions in (5) and (6), and $x_0 \in \mathbb{R}^d$ as an arbitrary initial point. Then, given the result in Theorem 6.4, the following results hold:*

- *If we set $B_0 = LI$, then we have*

$$\frac{f(x_t) - f(x_*)}{f(x_0) - f(x_*)} \leq \left( \frac{\delta_7 d\kappa + (\frac{3\delta_6}{\alpha(1-\beta)} \log \frac{C_0}{\delta_1} + \frac{3\delta_8}{\alpha(1-\beta)} C_0)\kappa}{t} \right)^t. \quad (27)$$

- *If we set $B_0 = \mu I$, then we have*

$$\frac{f(x_t) - f(x_*)}{f(x_0) - f(x_*)} \leq \left( \frac{(\delta_6 + \delta_7 + \delta_8 C_0)d\log\kappa + (\frac{3\delta_6}{\alpha(1-\beta)} \log \frac{C_0}{\delta_1} + \frac{3\delta_8}{\alpha(1-\beta)} C_0)\kappa}{t} \right)^t. \quad (28)$$

This result shows that BFGS with $B_0 = LI$ achieves a global superlinear rate of $\mathcal{O}((\frac{d\kappa + C_0\kappa}{t})^t)$, while BFGS with the initialization $B_0 = \mu I$ converges at a global superlinear rate of $\mathcal{O}((\frac{C_0 d\log\kappa + C_0\kappa}{t})^t)$. Hence, the superlinear result for $B_0 = \mu I$ outperforms the rate for $B_0 = LI$ when $C_0 \log \kappa \ll \kappa$.

**Remark 6.2.** *We chose $B_0 = LI$ and $B_0 = \mu I$ as two specific cases since they lead to explicit upper bounds in terms of the dimension $d$ and the condition number $\kappa$ in various theorems, simplifying the interpretation of our results. In practice, however, we often set $B_0 = cI$, where $c = \frac{s^\top y}{\|s\|^2}$, with $s = x_2 - x_1$, $y = \nabla f(x_2) - \nabla f(x_1)$, and $x_1, x_2$ as two randomly selected vectors. This choice ensures $c \in [\mu, L]$, and in the following numerical experiments, the performance of $B_0 = cI$ is similar to that of $B_0 = \mu I$. The complexity of BFGS with this initialization is reported in Appendix H.*

## 7 Complexity Analysis

**Discussions on the iteration complexity.** Using the three established convergence results in Theorems 4.1, 5.2 and 6.4, we can characterize the total number of iterations required for the BFGS method with the Armijo-Wolfe line search to find a solution with function suboptimality less than $\epsilon$. However, as discussed above, the choice of the initial Hessian approximation $B_0$ heavily influences the number of iterations required to observe these rates. To simplify our discussion, we focus on two specific initializations: $B_0 = LI$ and $B_0 = \mu I$.

**The case of $B_0 = LI$:** The overall iteration complexity of BFGS with $B_0 = LI$ is given by

$$\mathcal{O}\left( \min \left\{ \kappa \log \frac{1}{\epsilon}, (d + C_0)\kappa + \log \frac{1}{\epsilon}, \frac{\log \frac{1}{\epsilon}}{\log \left( \frac{1}{2} + \sqrt{\frac{1}{4} + \frac{1}{d\kappa + C_0\kappa} \log \frac{1}{\epsilon}} \right)} \right\} \right).$$

**The case of $B_0 = \mu I$:** The overall iteration complexity of BFGS with $B_0 = \mu I$ is given by

$$\mathcal{O}\left( \min \left\{ d\log\kappa + \kappa \log \frac{1}{\epsilon}, C_0(d\log\kappa + \kappa) + \log \frac{1}{\epsilon}, \frac{\log \frac{1}{\epsilon}}{\log \left( \frac{1}{2} + \sqrt{\frac{1}{4} + \frac{1}{C_0(d\log\kappa + \kappa)} \log \frac{1}{\epsilon}} \right)} \right\} \right).$$

We remark that the comparison between these two complexity bounds depends on the relative values of $\kappa$, $d$, $C_0$, and $\epsilon$, and neither is uniformly better than the other. It is worth noting that for BFGS with $B_0 = LI$, we achieve a complexity that is consistently superior to the $\mathcal{O}\left(\kappa \log \frac{1}{\epsilon}\right)$ complexity of gradient descent. Moreover, in scenarios where $C_0 = \mathcal{O}(1)$ and $d \ll \kappa$, BFGS with $B_0 = \mu I$ could result in an iteration complexity of $\mathcal{O}\left(\kappa + \log \frac{1}{\epsilon}\right)$, which is much more favorable than that of gradient descent. The proof of these complexity bounds can be found in Appendix I.

**Discussions on the line search complexity.** We present the log bisection algorithm to choose the step size $\eta_t$ at iteration $t$ satisfying the Armijo-Wolfe conditions (5) and (6) in Algorithm 1 in Appendix J. We define $\eta_{min}$ and $\eta_{max}$ as the lower and upper bounds of the "slicing window" containing the trial step size $\eta_t$, respectively. We start with the initial trial step size $\eta_t = 1$ and keep enlarging or decreasing it depending on whether the Armijo condition (5) or the curvature condition (6) is satisfied. Then, we dynamically update $\eta_{min}, \eta_{max}$ and shrink the size of this "slicing window" $(\eta_{min}, \eta_{max})$. We pick the trial step size $\eta$ as the geometric mean of $\eta_{min}$ and $\eta_{max}$, i.e., $\log \eta = (\log \eta_{max} + \log \eta_{max})/2$, which is the reason why we call this algorithm "log bisection". Note that in each loop of Algorithm 1, we query the function value and gradient at most once to check the Armijo-Wolfe conditions at Lines 2 and 9. The next theorem characterizes the average number of function value and gradient evaluations per iteration in Algorithm 1 after $t$ iterations, denoted by $\Lambda_t$, which is equivalent to the average number of loops per iterations.

**Theorem 7.1.** *Suppose Assumptions 2.1, 2.2 and 2.3 hold. Let $\{x_t\}_{t\geq 0}$ be generated by BFGS with step size satisfying the Armijo-Wolfe conditions in (5) and (6) and is chosen by Algorithm 1. If we define $\sigma := \left(\Psi(\bar{B}_0) + \frac{3}{\alpha(1-\beta)}\kappa\right)C_0$, then for any initial point $x_0 \in \mathbb{R}^d$ and initial Hessian approximation $B_0 \in \mathbb{S}_{++}^d$, the average number of the function value and gradient evaluations per iteration in Algorithm 1 after $t$ iterations satisfies*

$$\Lambda_t \leq 2 + \log_2\left(1 + \frac{1-\beta}{\beta-\alpha} + \frac{2(1-\beta)}{\beta-\alpha}\frac{\sigma}{t}\right) + 2\log_2\left(\log_2 16(1-\alpha) + \log_2\left(1 + \frac{\sigma}{t}\right) + \frac{6\Psi(\tilde{B}_0) + 12\sigma}{t}\right).$$

The above result shows that when we run BFGS for $N$ iterations, the total number of function and gradient evaluations is $\mathcal{O}\left(N + N\log(1 + \frac{\sigma}{N}) + N\log(1 + \frac{\Psi(\tilde{B}_0)+\sigma}{N})\right)$. Thus, the total line search complexity can always be bounded by $\mathcal{O}(N\log(\Psi(\tilde{B}_0) + \sigma)) = \mathcal{O}(N \max\{\log d, \log \kappa, \log C_0\})$. Furthermore, notice that when $N$ is sufficiently large such that we reach the superlinear convergence stage, i.e., $N = \Omega(\Psi(\tilde{B}_0) + \sigma)$, the total line search complexity becomes $\mathcal{O}(N)$, which means the average number of function and gradient evaluations per iteration is a constant $\mathcal{O}(1)$. We report the line search complexity results of different $B_0 = LI$ and $B_0 = \mu I$ in Appendix K.4.

## 8 Numerical Experiments

We conduct numerical experiments on a cubic objective function defined as

$$f(x) = \frac{\alpha}{12}\left(\sum_{i=1}^{d-1} g(v_i^\top x - v_{i+1}^\top x) - \beta v_1^\top x\right) + \frac{\lambda}{2}\|x\|^2, \tag{29}$$

and $g : \mathbb{R} \to \mathbb{R}$ is defined as

$$g(w) = \begin{cases} \frac{1}{3}|w|^3 & |w| \leq \Delta, \\ \Delta w^2 - \Delta^2 |w| + \frac{1}{3}\Delta^3 & |w| > \Delta, \end{cases} \tag{30}$$

where $\alpha, \beta, \lambda, \Delta \in \mathbb{R}$ are hyper-parameters and $\{v_i\}_{i=1}^n$ are standard orthogonal unit vectors in $\mathbb{R}^d$. We focus on this objective function because it is used in [26] to establish a tight lower bound for second-order methods. We compare the convergence paths of BFGS with an inexact line search step size $\eta_t$ that satisfies the Armijo-Wolfe conditions (5) and (6) for various initialization matrices $B_0$: specifically, $B_0 = LI$, $B_0 = \mu I$, $B_0 = I$, and $B_0 = cI$ where $c$ is defined in Remark 6.2. It is easily verified that $c \in [\mu, L]$. We also compare the performance of BFGS methods to the gradient descent (GD) method with backtracking line search, using $\alpha = 0.1$ in condition (5) and $\beta = 0.9$ in condition (6). Step size $\eta_t$ is chosen at each iteration via log bisection in Algorithm 1. Empirical results are compared across various dimensions $d$ and condition numbers $\kappa$, with the x-axis representing the number of iterations $t$ and the y-axis showing the ratio $\frac{f(x_t)-f(x_*)}{f(x_0)-f(x_*)}$.

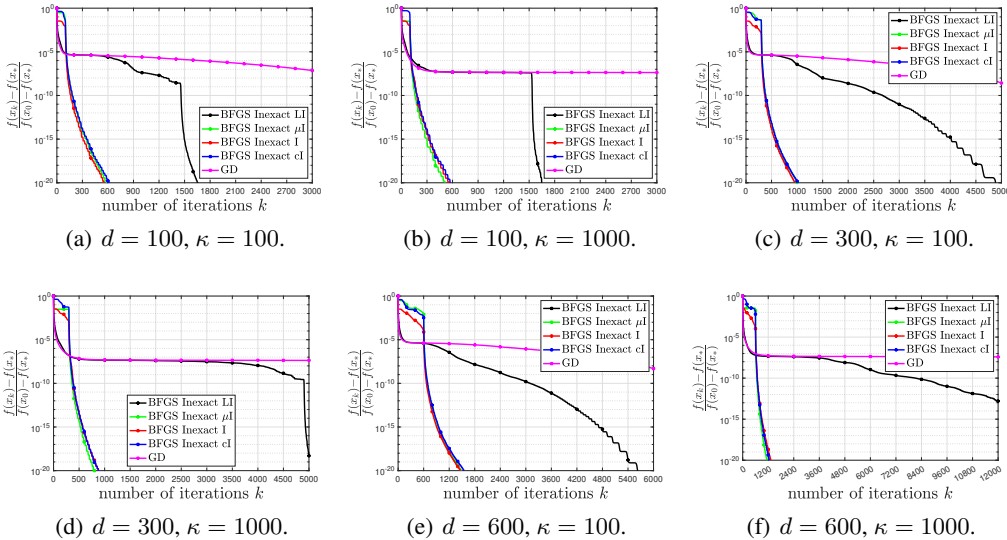

Figure 1: Convergence curves of BFGS with inexact line search of different $B_0$ and gradeint descent with backtracking line search.

First, we observe that BFGS with $B_0 = LI$ initially converges faster than BFGS with $B_0 = \mu I$ in most plots, aligning with our theoretical findings that the linear convergence rate of BFGS with $B_0 = LI$ surpasses that of $B_0 = \mu I$ in Corollary 4.2. In Corollary 4.2, we show that BFGS with $B_0 = LI$ could achieve the linear rate of $(1 - 1/\kappa)$ from the first iteration while BFGS with $B_0 = \mu I$ needs to run $d \log \kappa$ to reach the same linear rate. Second, the transition to superlinear convergence for BFGS with $B_0 = \mu I$ typically occurs around $t \approx d$, as predicted by our theoretical analysis. Although BFGS with $B_0 = LI$ initially converges faster, its transition to superlinear convergence consistently occurs later than for $B_0 = \mu I$. Notably, for a fixed dimension $d = 600$, the transition to superlinear convergence for $B_0 = LI$ occurs increasingly later as the problem condition number rises, an effect not observed for $B_0 = \mu I$. This phenomenon indicates that the superlinear rate for $B_0 = LI$ is more sensitive to the condition number $\kappa$, which corroborates our results in Corollary 6.5. In Corollary 6.5, we present that BFGS with $B_0 = LI$ needs $d\kappa$ steps to reach the superlinear convergence stage while this is improved to $d \log \kappa$ for BFGS with $B_0 = \mu I$. Moreover, the performance of BFGS with $B_0 = I$ and $B_0 = cI$ is similar to BFGS with $B_0 = \mu I$. Notice that the initializations of $B_0 = I$ and $B_0 = cI$ are two commonly-used practical choices of the initial Hessian approximation matrix $B_0$.

## 9 Conclusions, Limitations, and Future Directions

In this paper, we analyzed the global non-asymptotic convergence rates of BFGS with Armijo-Wolfe line search. We showed for an objective function that is $\mu$-strongly convex with an $L$-Lipschitz gradient, BFGS achieves a global convergence rate of $(1 - 1/\kappa)^t$, where $\kappa = L/\mu$. Additionally, assuming the Hessian is $M$-Lipschitz, we showed BFGS achieves a linear convergence rate determined solely by the line search parameters, independent of the condition number. Under similar assumptions, we also established a global superlinear convergence rate. Given these bounds, we determined the overall iteration complexity of BFGS with the Armijo-Wolfe line search and specified this complexity for initial Hessian approximations $B_0 = LI$ and $B_0 = \mu I$.

One limitation of this paper is that the analysis only applies to strongly convex functions. Developing an analysis for the general convex setting is still unsolved. Another drawback is that we focus solely on the BFGS method. Extending our theoretical results to the entire convex Broyden's class of quasi-Newton methods, including both BFGS and DFP, is a natural next step.

## Acknowledgments

The research of Q. Jin, R. Jiang, and A. Mokhtari is supported in part by NSF Award 2007668 and the NSF AI Institute for Foundations of Machine Learning (IFML).

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

# Appendix

## A Some Results on the Connections between Different Hessian Matrices

**Lemma A.1.** *Suppose Assumptions 2.1, 2.2, and 2.3 hold, and recall the definitions of the matrices $J_t$ and $G_t$ in (8), and the quantity $C_t$ in (9). Then, the following statements hold:*

*(a) Suppose that $f(x_{t+1}) \leq f(x_t)$ for any $t \geq 0$, we have that*

$$\frac{1}{1+C_t} \nabla^2 f(x_*) \preceq J_t \preceq (1+C_t)\nabla^2 f(x_*). \tag{31}$$

*(b) Suppose that $f(x_{t+1}) \leq f(x_t)$ for any $t \geq 0$ and $\hat{\tau} \in [0,1]$, we have that*

$$\frac{1}{1+C_t} \nabla^2 f(x_*) \preceq \nabla^2 f(x_t + \hat{\tau}(x_{t+1} - x_t)) \preceq (1+C_t)\nabla^2 f(x_*). \tag{32}$$

*(c) For any $t \geq 0$, we have that*

$$\frac{1}{1+C_t} \nabla^2 f(x_*) \preceq \nabla^2 f(x_t) \preceq (1+C_t)\nabla^2 f(x_*). \tag{33}$$

*(d) For any $t \geq 0$, we have that*

$$\frac{1}{1+C_t} \nabla^2 f(x_*) \preceq G_t \preceq (1+C_t)\nabla^2 f(x_*). \tag{34}$$

*(e) For any $t \geq 0$ and $\tilde{\tau} \in [0,1]$, we have that*

$$\frac{1}{1+C_t} G_t \preceq \nabla^2 f(x_t + \tilde{\tau}(x_* - x_t)) \preceq (1+C_t)G_t. \tag{35}$$

*(f) For any $t \geq 0$ and $\tilde{\tau}, \hat{\tau} \in [0,1]$, suppose that $f(x_{t+1}) \leq f(x_t)$. Then, we have that*

$$\frac{1}{1+2C_t} \nabla^2 f(x_t + \hat{\tau}s_t) \preceq \nabla^2 f(x_t + \tilde{\tau}s_t) \preceq (1+2C_t)\nabla^2 f(x_t + \hat{\tau}s_t). \tag{36}$$

*Proof.* (a) Recall the definition of $J_t$ in (8). Using the triangle inequality, we have that

$$\|\nabla^2 f(x_*) - J_t\| = \left\| \int_0^1 \left( \nabla^2 f(x_*) - \nabla^2 f(x_t + \tau(x_{t+1} - x_t)) \right) d\tau \right\|$$

$$\leq \int_0^1 \|\nabla^2 f(x_*) - \nabla^2 f(x_t + \tau(x_{t+1} - x_t))\| d\tau.$$

Moreover, it follows from Assumption 2.3 that $\|\nabla^2 f(x_*) - \nabla^2 f(x_t + \tau(x_{t+1} - x_t))\| \leq M\|(1-\tau)(x_* - x_t) + \tau(x_* - x_{t+1})\|$ for any $\tau \in [0,1]$. Thus, we can further apply the triangle inequality to obtain

$$\|\nabla^2 f(x_*) - J_t\| \leq \int_0^1 M\|(1-\tau)(x_* - x_t) + \tau(x_* - x_{t+1})\| d\tau$$

$$\leq M\|x_t - x_*\| \int_0^1 (1-\tau)d\tau + M\|x_{t+1} - x_*\| \int_0^1 \tau d\tau$$

$$= \frac{M}{2}(\|x_t - x_*\| + \|x_{t+1} - x_*\|).$$

Since $f$ is strongly convex, by Assumption 2.1 and $f(x_{t+1}) \leq f(x_t)$, we have $\frac{\mu}{2}\|x_t - x_*\|^2 \leq f(x_t) - f(x_*)$, which implies that $\|x_t - x_*\| \leq \sqrt{2(f(x_t) - f(x_*))/\mu}$. Similarly, since $f(x_{t+1}) \leq f(x_t)$, it also holds that $\|x_{t+1} - x_*\| \leq \sqrt{2(f(x_{t+1}) - f(x_*))/\mu} \leq \sqrt{2(f(x_t) - f(x_*))/\mu}$. Hence, we obtain that

$$\|\nabla^2 f(x_*) - J_t\| \leq \frac{M}{\sqrt{\mu}}\sqrt{2(f(x_t) - f(x_*))}. \tag{37}$$

Moreover, notice that by Assumption 2.1, we also have $J_t \succeq \mu I$ and $\nabla^2 f(x_*) \succeq \mu I$. Hence, (37) implies that

$$\nabla^2 f(x_*) - J_t \preceq \|\nabla^2 f(x_*) - J_t\| I \preceq \frac{M}{\mu^{\frac{3}{2}}} \sqrt{2(f(x_t) - f(x_*))} J_t = C_t J_t,$$

$$J_t - \nabla^2 f(x_*) \preceq \|J_t - \nabla^2 f(x_*)\| I \preceq \frac{M}{\mu^{\frac{3}{2}}} \sqrt{2(f(x_t) - f(x_*))} \nabla^2 f(x_*) = C_t \nabla^2 f(x_*).$$

where we used the definition of $C_t$ in (9). By rearranging the terms, we obtain (31).

(b) Similar to the arguments in (a), for any $\hat{\tau} \in [0, 1]$, we have that

$$\begin{aligned}
&\left\|\nabla^2 f(x_t + \hat{\tau}(x_{t+1} - x_t)) - \nabla^2 f(x_*)\right\| \\
&\leq M\|(1 - \hat{\tau})(x_t - x_*) + \hat{\tau}(x_{t+1} - x_*)\| \\
&\leq M\Big((1 - \hat{\tau})\|x_t - x_*\| + \hat{\tau}\|x_{t+1} - x_*\|\Big) \\
&\leq M\Big((1 - \hat{\tau})\sqrt{\frac{2}{\mu}(f(x_t) - f(x_*))} + \hat{\tau}\sqrt{\frac{2}{\mu}(f(x_{t+1}) - f(x_*))}\Big) \\
&\leq M\sqrt{\frac{2}{\mu}(f(x_t) - f(x_*))}
\end{aligned}$$

Moreover, notice that by Assumption 2.1, we also have $\nabla^2 f(x_t + \hat{\tau}(x_{t+1} - x_t)) \succeq \mu I$ and $\nabla^2 f(x_*) \succeq \mu I$. The rest follows similarly as in the proof of (a) and we prove (32).

(c) Similar to the arguments in (a), we have that

$$\left\|\nabla^2 f(x_*) - \nabla^2 f(x_t)\right\| \leq M\|x_t - x_*\| \leq \frac{M}{\sqrt{\mu}}\sqrt{2(f(x_t) - f(x_*))}.$$

Moreover, notice that by Assumption 2.1 we also have $\nabla^2 f(x_t) \succeq \mu I$ and $\nabla^2 f(x_*) \succeq \mu I$. The rest follows similarly as in the proof of (a) and we prove (33).

(d) Recall the definition of $G_t$ in (8). Similar to the arguments in (a), we have that

$$\begin{aligned}
\|\nabla^2 f(x_*) - G_t\| &= \left\|\int_0^1 \left(\nabla^2 f(x_*) - \nabla^2 f(x_t + \tau(x_* - x_t))\right) d\tau\right\| \\
&\leq \int_0^1 \|\nabla^2 f(x_*) - \nabla^2 f(x_t + \tau(x_* - x_t))\| d\tau \\
&\leq M \int_0^1 \|(1 - \tau)(x_* - x_t)\| d\tau = M\|x_t - x_*\| \int_0^1 (1 - \tau) d\tau \\
&= \frac{M}{2}\|x_t - x_*\| \leq \frac{M}{\sqrt{\mu}}\sqrt{2(f(x_t) - f(x_*))}.
\end{aligned}$$

Moreover, notice that by Assumption 2.1 we also have $G_t \succeq \mu I$ and $\nabla^2 f(x_*) \succeq \mu I$. The rest follows similarly as in the proof of (a) and we prove (34).

(e) Recall the definition of $g_t$ in (8). Similar to the arguments in (a), for any $\tilde{\tau} \in [0, 1]$, we have that

$$\begin{aligned}
&\left\|\nabla^2 f(x_t + \tilde{\tau}(x_* - x_t)) - G_t\right\| \\
&= \left\|\int_0^1 \left(\nabla^2 f(x_t + \tilde{\tau}(x_* - x_t)) - \nabla^2 f(x_t + \tau(x_* - x_t))\right) d\tau\right\| \\
&\leq \int_0^1 \left\|\nabla^2 f(x_t + \tilde{\tau}(x_* - x_t)) - \nabla^2 f(x_t + \tau(x_* - x_t))\right\| d\tau \\
&\leq \int_0^1 M|\tilde{\tau} - \tau|\|x_t - x_*\| d\tau \leq \frac{1}{2}M\|x_t - x_*\| \leq \frac{M}{\sqrt{\mu}}\sqrt{2(f(x_t) - f(x_*))}.
\end{aligned}$$

Moreover, notice that by Assumption 2.1, we also have $\nabla^2 f(x_t + \tilde{\tau}(x_* - x_t)) \succeq \mu I$ and $G_t \succeq \mu I$. The rest follows similarly as in the proof of (a) and we prove (35).

(f) Similar to the arguments in (a), for any $\tilde{\tau}, \hat{\tau} \in [0,1]$, we have that

$$
\begin{aligned}
&\left\| \nabla^2 f(x_t + \tilde{\tau} s_t) - \nabla^2 f(x_t + \hat{\tau} s_t) \right\| \\
&\leq M|\tilde{\tau} - \hat{\tau}| \|s_t\| \leq M \|s_t\| \leq M(\|x_{t+1} - x_*\| + \|x_t - x_*\|) \\
&\leq M\left( \sqrt{\frac{2}{\mu}(f(x_t) - f(x_*))} + \sqrt{\frac{2}{\mu}(f(x_{t+1}) - f(x_*))} \right) \\
&\leq 2M \sqrt{\frac{2}{\mu}(f(x_t) - f(x_*))}
\end{aligned}
$$

Moreover, notice that by Assumption 2.1, we also have $\nabla^2 f(x_t + \tilde{\tau} s_t) \succeq \mu I$ and $\nabla^2 f(x_t + \hat{\tau} s_t) \succeq \mu I$. The rest follows similarly as in the proof of (a) and we prove (36).

$\square$

# B  Proof of Lemma 2.1

Recall that $g_t = \nabla f(x_t)$. Given the condition in (5) and the fact that $s_t = \eta_t d_t$, we have

$$
f(x_{t+1}) \leq f(x_t) + \alpha g_t^\top s_t.
$$

Moreover, since $B_t$ is symmetric positive definite, we have $-g_t^\top s_t = \eta_t g_t^\top B_t^{-1} g_t > 0$ (unless $g_t = 0$ and we are at the optimal solution). This further leads to the first claim, which is

$$
\frac{f(x_t) - f(x_{t+1})}{-g_t^\top s_t} \geq \alpha.
$$

Similarly, the above argument implies that $\alpha g_t^\top s_t < 0$ and as a result $f(x_{t+1}) \leq f(x_t)$ and the last claim also follows.

To prove the second claim, we leverage the condition in (6). Specifically, if we subtract $g_t^\top d_t$ from both sides of that condition, we obtain that

$$
(g_{t+1} - g_t)^\top d_t \geq (\beta - 1) g_t^\top d_t
$$

Next, using the fact that $s_t = \eta_t d_t$, by multiplying both sides by $\eta_t$ and use the simplification $y_t = g_{t+1} - g_t$ we obtain that

$$
y_t^\top s_t \geq (\beta - 1) g_t^\top s_t = -g_t^\top s_t (1 - \beta).
$$

Again using the argument that $-g_t^\top s_t$ is positive (if we are not at the optimal solution), we can divide both sides of the above inequality by $-g_t^\top s_t$, leading to the second claim.

# C  Proof of Proposition 3.1

First, we note that $\hat{g}_t^\top \hat{s}_t = g_t^\top s_t$ and $\hat{y}_t^\top \hat{s}_t = y_t^\top s_t$. Using the definition of $\hat{p}_t$ in (16), we have that

$$
f(x_t) - f(x_{t+1}) = \hat{p}_t \frac{-\hat{g}_t^\top \hat{s}_t}{\|\hat{g}_t\|^2} \|\hat{g}_t\|^2. \tag{38}
$$

Hence, using the definition of $\hat{\theta}_t$ in (14) and the definition of $\hat{m}_t, \hat{n}_t$ in (16), it follows that

$$
\frac{-\hat{g}_t^\top \hat{s}_t}{\|\hat{g}_t\|^2} = \frac{(\hat{g}_t^\top \hat{s}_t)^2}{\|\hat{g}_t\|^2 \|\hat{s}_t\|^2} \frac{\|\hat{s}_t\|^2}{-\hat{g}_t^\top \hat{s}_t} = \frac{(\hat{g}_t^\top \hat{s}_t)^2}{\|\hat{g}_t\|^2 \|\hat{s}_t\|^2} \frac{\|\hat{s}_t\|^2}{\hat{y}_t^\top \hat{s}_t} \frac{\hat{y}_t^\top \hat{s}_t}{-\hat{g}_t^\top \hat{s}_t} = \hat{n}_t \frac{\cos^2(\hat{\theta}_t)}{\hat{m}_t}.
$$

Furthermore, we have $\|\hat{g}_t\|^2 = \hat{q}_t(f(x_t) - f(x_*))$ from the definition of $\hat{q}_t$ in (16). Thus, the equality in (38) can be rewritten as

$$
f(x_t) - f(x_{t+1}) = \hat{p}_t \hat{q}_t \hat{n}_t \frac{\cos^2(\hat{\theta}_t)}{\hat{m}_t} (f(x_t) - f(x_*)).
$$

By rearranging the term in the above equality, we obtain

$$f(x_{t+1}) - f(x_*) = \left(1 - \hat{p}_t\hat{q}_t\hat{n}_t\frac{\cos^2(\hat{\theta}_t)}{\hat{m}_t}\right)(f(x_t) - f(x_*)), \tag{39}$$

To prove the inequality in (15), note that for any $t \geq 1$, we have

$$\frac{f(x_t) - f(x_*)}{f(x_0) - f(x_*)} = \prod_{i=0}^{t-1}\frac{f(x_{i+1}) - f(x_*)}{f(x_i) - f(x_*)} = \prod_{i=0}^{t-1}\left(1 - \hat{p}_i\hat{q}_i\hat{n}_i\frac{\cos^2(\hat{\theta}_i)}{\hat{m}_i}\right),$$

where the last equality is due to (39). Note that all the terms of the form $1 - \hat{p}_i\hat{q}_i\hat{n}_i\frac{\cos^2(\hat{\theta}_i)}{\hat{m}_i}$ are non-negative, for any $i \geq 0$. Thus, by applying the inequality of arithmetic and geometric means twice, we obtain

$$\prod_{i=0}^{t-1}\left(1 - \hat{p}_i\hat{q}_i\hat{n}_i\frac{\cos^2(\hat{\theta}_i)}{\hat{m}_i}\right) \leq \left[\frac{1}{t}\sum_{i=0}^{t-1}\left(1 - \hat{p}_i\hat{q}_i\hat{n}_i\frac{\cos^2(\hat{\theta}_i)}{\hat{m}_i}\right)\right]^t$$

$$= \left[1 - \frac{1}{t}\sum_{i=0}^{t-1}\hat{p}_i\hat{q}_i\hat{n}_i\frac{\cos^2(\hat{\theta}_i)}{\hat{m}_i}\right]^t \leq \left[1 - \left(\prod_{i=0}^{t-1}\hat{p}_i\hat{q}_i\hat{n}_i\frac{\cos^2(\hat{\theta}_i)}{\hat{m}_i}\right)^{\frac{1}{t}}\right]^t.$$

This completes the proof.

# D  Results from [38]

In this section, we summarize some results that we use from [38] to establish a lower bound on $\prod_{i=0}^{t-1}\frac{\cos^2(\hat{\theta}_i)}{\hat{m}_i}$ and $\hat{q}_t$.

**Proposition D.1** ([38, Proposition 2]). *Let $\{B_t\}_{t\geq 0}$ be the Hessian approximation matrices generated by the BFGS update in (3). For a given weight matrix $P \in \mathbb{S}_{++}^d$, recall the weighted vectors defined in (11) and the weighted matrix in (12). Then, we have*

$$\Psi(\hat{B}_{t+1}) \leq \Psi(\hat{B}_t) + \frac{\|\hat{y}_t\|^2}{\hat{y}_t^\top\hat{s}_t} - 1 + \log\frac{\cos^2\hat{\theta}_t}{\hat{m}_t}, \qquad \forall t \geq 0,$$

*where $\hat{m}_t$ is defined in (16) and $\cos(\hat{\theta}_t)$ is defined in (14). As a corollary, we have,*

$$\sum_{i=0}^{t-1}\log\frac{\cos^2(\hat{\theta}_i)}{\hat{m}_i} \geq -\Psi(\hat{B}_0) + \sum_{i=0}^{t-1}\left(1 - \frac{\|\hat{y}_i\|^2}{\hat{y}_i^\top\hat{s}_i}\right), \qquad \forall t \geq 1. \tag{40}$$

If we take exponentiation on both sides of the above inequality (40) in Proposition D.1, we can obtain a lower bound for the product $\prod_{i=0}^{t-1}\frac{\cos^2(\hat{\theta}_i)}{\hat{m}_i}$ with the sum $\sum_{i=0}^{t-1}\frac{\|\hat{y}_i\|^2}{\hat{s}_i^\top\hat{y}_i}$ and $\Psi(\hat{B}_0)$. This classical inequality describing the relationship between the ratio $\frac{\cos^2(\hat{\theta}_t)}{\hat{m}_t}$ and the potential function $\Psi(.)$ plays a critical role in the following convergence analysis.

In the following two lemmas, we provide bounds on the quantities $\hat{q}_t$ and $\|\hat{y}_t\|^2/\hat{s}_t^\top\hat{y}_t$ respectively by directly citing results from Lemma 4 and Lemma 5 in [38] again. Notice that both $\hat{q}_t$ and $\|\hat{y}_t\|^2/\hat{s}_t^\top\hat{y}_t$ depend on different choices of the weight matrix $P$.

**Lemma D.2** ([38, Lemma 4]). *Recall the definition $\hat{q}_t = \frac{\|\hat{g}_t\|^2}{f(x_t) - f(x_*)}$ in (16). Suppose Assumptions 2.1, 2.2, and 2.3 hold. Then we have the following results:*

(a) *If we choose $P = LI$, then $\hat{q}_t \geq 2/\kappa$.*

(b) *If we choose $P = \nabla^2 f(x_*)$, then $\hat{q}_t \geq 2/(1 + C_t)^2$.*

**Lemma D.3** ([38, Lemma 5]). *Let $\{x_t\}_{t\geq 0}$ be the iterates generated by the BFGS algorithm with inexact line search satisfying (5) and (6). Suppose Assumptions 2.1, 2.2, and 2.3 hold. Then we have the following results:*

(a) *If we choose $P = LI$, then $\frac{\|\hat{y}_t\|^2}{\hat{s}_t^\top\hat{y}_t} \leq 1$.*

(b) *If we choose $P = \nabla^2 f(x_*)$, then $\frac{\|\hat{y}_t\|^2}{\hat{s}_t^\top\hat{y}_t} \leq 1 + C_t$.*

# E  Proofs in Section 4

## E.1  Proof of Theorem 4.1

Recall that we choose $P = LI$ throughout the proof. Note that given this weight matrix, it can be easily verified that $\frac{\|\hat{y}_t\|^2}{\hat{s}_t^\top \hat{y}_t} \leq 1$ for any $t \geq 0$ by using Lemma D.3 (a). Hence, we use (40) in Proposition D.1 to obtain

$$\sum_{i=0}^{t-1} \log \frac{\cos^2(\hat{\theta}_i)}{\hat{m}_i} \geq -\Psi(\bar{B}_0) + \sum_{i=0}^{t-1} \left( 1 - \frac{\|\hat{y}_i\|^2}{\hat{s}_i^\top \hat{y}_i} \right) \geq -\Psi(\bar{B}_0),$$

which further implies that

$$\prod_{i=0}^{t-1} \frac{\cos^2(\hat{\theta}_i)}{\hat{m}_i} \geq e^{-\Psi(\bar{B}_0)}.$$

Moreover, for the choice $P = LI$, it can be shown that $\hat{q}_t = \frac{\|g_t\|^2}{L(f(x_t) - f(x_*))} \geq \frac{2}{\kappa}$ by using Lemma D.2 (a). From Lemma 2.1, we know $\hat{p}_t \geq \alpha$ and $\hat{n}_t \geq 1 - \beta$, which lead to

$$\prod_{i=0}^{t-1} \frac{\hat{p}_i \hat{n}_i \hat{q}_i}{\hat{m}_i} \cos^2(\hat{\theta}_i) \geq \prod_{i=0}^{t-1} \hat{p}_i \prod_{i=0}^{t-1} \hat{q}_i \prod_{i=0}^{t-1} \hat{n}_i \prod_{i=0}^{t-1} \frac{\cos^2(\hat{\theta}_i)}{\hat{m}_i} \geq \left( \frac{2\alpha(1-\beta)}{\kappa} \right)^t e^{-\Psi(\bar{B}_0)}.$$

Thus, it follows from Proposition 3.1 that

$$\frac{f(x_t) - f(x_*)}{f(x_0) - f(x_*)} \leq \left[ 1 - \left( \prod_{i=0}^{t-1} \frac{\hat{p}_i \hat{q}_i \hat{n}_i}{\hat{m}_i} \cos^2(\hat{\theta}_i) \right)^{\frac{1}{t}} \right]^t \leq \left( 1 - e^{-\frac{\Psi(\bar{B}_0)}{t}} \frac{2\alpha(1-\beta)}{\kappa} \right)^t.$$

This completes the proof.

## E.2  Proof of Corollary 4.2

Notice that in the first case where $B_0 = LI$, we have $\Psi(\bar{B}_0) = 0$ and thus it achieves the best linear convergence results according to Theorem 4.1. On the other hand, for $B_0 = \mu I$, we have $\Psi(\bar{B}_0) = \Psi(\frac{\mu}{L}I) = d(\frac{1}{\kappa} - 1 + \log \kappa) \leq d \log \kappa$. We complete the proof by combining these conditions with the inequality (17) in Theorem 4.1. Notice that $e^{-x} \geq e^{-1} \geq \frac{1}{3}$ for $x \leq 1$.

# F  Proofs in Section 5

## F.1  Proof of Proposition 5.1

Recall that we choose the weight matrix as $P = \nabla^2 f(x_*)$ throughout the proof. Similar to the proof of Theorem 4.1, we start from the key inequality in (15), but we apply different bounds on the $\hat{q}_t$ and $\frac{\cos^2(\hat{\theta}_t)}{\hat{m}_t}$. Specifically, by using Lemma D.3 (b), we have $\frac{\|\hat{y}_t\|^2}{\hat{s}_t^\top \hat{y}_t} \leq 1 + C_t$ for any $t \geq 0$. Hence, we use (40) in Proposition D.1 to obtain

$$\sum_{i=0}^{t-1} \log \frac{\cos^2(\hat{\theta}_i)}{\hat{m}_i} \geq -\Psi(\tilde{B}_0) + \sum_{i=0}^{t-1} \left( 1 - \frac{\|\hat{y}_i\|^2}{\hat{s}_i^\top \hat{y}_i} \right) \geq -\Psi(\tilde{B}_0) - \sum_{i=0}^{t-1} C_i,$$

which further implies that

$$\prod_{i=0}^{t-1} \frac{\cos^2(\hat{\theta}_i)}{\hat{m}_i} \geq e^{-\Psi(\tilde{B}_0) - \sum_{i=0}^{t-1} C_i}. \tag{41}$$

Moreover, since $\hat{q}_t \geq \frac{2}{(1+C_t)^2}$ for any $t \geq 0$ by using Lemma D.2 (b), we get

$$\prod_{i=0}^{t-1} \hat{q}_i \geq \prod_{i=0}^{t-1} \frac{2}{(1+C_i)^2} \geq 2^t \prod_{i=0}^{t-1} e^{-2C_i} = 2^t e^{-2\sum_{i=0}^{t-1} C_i}, \tag{42}$$

where we use the inequality $1 + x \le e^x$ for any $x \in \mathbb{R}$. From Lemma 2.1, we know $\hat{p}_t \ge \alpha$ and $\hat{n}_t \ge 1 - \beta$, which lead to

$$\prod_{i=0}^{t-1} \hat{p}_i \hat{n}_i \ge \alpha^t (1-\beta)^t. \tag{43}$$

Combining (41), (42), (43) and (15) from Proposition 3.1, we prove that

$$\frac{f(x_t) - f(x_*)}{f(x_0) - f(x_*)} \le \left[ 1 - \left( \prod_{i=0}^{t-1} \frac{\hat{p}_i \hat{q}_i \hat{n}_i}{\hat{m}_i} \cos^2(\hat{\theta}_i) \right)^{\frac{1}{t}} \right]^t \le \left( 1 - 2\alpha(1-\beta) e^{-\frac{\Psi(\tilde{B}_0) + 3\sum_{i=0}^{t-1} C_i}{t}} \right)^t.$$

This completes the proof.

## F.2   Proof of Theorem 5.2

When we have $t \ge \Psi(\tilde{B}_0) + 3\sum_{i=0}^{t-1} C_i$, Proposition 5.1 implies the condition that $\frac{f(x_t) - f(x_*)}{f(x_0) - f(x_*)} \le \left( 1 - \frac{2\alpha(1-\beta)}{e} \right)^t \le \left( 1 - \frac{2\alpha(1-\beta)}{3} \right)^t$, which leads to the linear rate in (20). Hence, it is sufficient to establish an upper bound on $\sum_{i=0}^{t-1} C_i$. Recall that $C_i = \frac{M}{\mu^{\frac{3}{2}}} \sqrt{2(f(x_i) - f(x_*))}$ defined in (9). We decompose the sum into two parts: $\sum_{i=0}^{\lceil \Psi(\bar{B}_0)\rceil - 1} C_i$ and $\sum_{i=\lceil \Psi(\bar{B}_0)\rceil}^{t} C_i$. For the first part, note that since $f(x_{i+1}) \le f(x_i)$ by Lemma 2.1, we also have $C_{i+1} \le C_i$ for $i \ge 0$. Hence, we have $\sum_{i=0}^{\lceil \Psi(\bar{B}_0)\rceil - 1} C_i \le C_0 \lceil \Psi(\bar{B}_0) \rceil \le C_0(\Psi(\bar{B}_0) + 1)$. Moreover, by Theorem 4.1, when $t \ge \Psi(\bar{B}_0)$ we have

$$\frac{f(x_t) - f(x_*)}{f(x_0) - f(x_*)} \le \left( 1 - e^{-\frac{\Psi(\bar{B}_0)}{t}} \frac{2\alpha(1-\beta)}{\kappa} \right)^t \le \left( 1 - \frac{2\alpha(1-\beta)}{e\kappa} \right)^t \le \left( 1 - \frac{2\alpha(1-\beta)}{3\kappa} \right)^t.$$

Hence, this further implies that

$$\sum_{i=\lceil \Psi(\bar{B}_0)\rceil}^{t} C_i = C_0 \sum_{i=\lceil \Psi(\bar{B}_0)\rceil}^{t} \sqrt{\frac{f(x_i) - f(x_*)}{f(x_0) - f(x_*)}} \le C_0 \sum_{i=\lceil \Psi(\bar{B}_0)\rceil}^{t} \left( 1 - \frac{2\alpha(1-\beta)}{3\kappa} \right)^{\frac{i}{2}}$$

$$\le C_0 \sum_{i=1}^{\infty} \left( 1 - \frac{2\alpha(1-\beta)}{3\kappa} \right)^{\frac{i}{2}} \le C_0 \left( \frac{3\kappa}{\alpha(1-\beta)} - 1 \right),$$

where we used the fact that $\sum_{i=1}^{\infty} (1-\rho)^{\frac{i}{2}} = \frac{\sqrt{1-\rho}}{1-\sqrt{1-\rho}} = \frac{\sqrt{1-\rho}+1-\rho}{\rho} \le \frac{2}{\rho} - 1$ for any $\rho \in (0,1)$. Hence, by combining both inequalities, we have

$$\sum_{i=0}^{t-1} C_i = \sum_{i=0}^{\lceil \Psi(\bar{B}_0)\rceil - 1} C_i + \sum_{i=\lceil \Psi(\bar{B}_0)\rceil}^{t} C_i \le C_0 \Psi(\bar{B}_0) + \frac{3 C_0 \kappa}{\alpha(1-\beta)}. \tag{44}$$

Hence, this proves that (20) is satisfied when $t \ge \Psi(\tilde{B}_0) + 3 C_0 \Psi(\bar{B}_0) + \frac{9 C_0 \kappa}{\alpha(1-\beta)}$.

## F.3   Proof of Corollary 5.3

For $B_0 = LI$, we have $\bar{B}_0 = \frac{1}{L} B_0 = I$ and $\tilde{B}_0 = \nabla^2 f(x_*)^{-\frac{1}{2}} B_0 \nabla^2 f(x_*)^{-\frac{1}{2}} = L \nabla^2 f(x_*)^{-1}$. Thus, it holds that $\Psi(\bar{B}_0) = \Psi(I) = 0$. Moreover, by Assumptions 2.1 and 2.2, we have $\frac{1}{L} I \preceq \nabla^2 f(x_*)^{-1} \preceq \frac{1}{\mu} I$, which implies that $I \preceq \tilde{B}_0 \preceq \kappa I$. Thus, we further have

$$\Psi(\tilde{B}_0) \le \mathbf{Tr}(\kappa I) - d - \log \mathbf{Det}(I) = d\kappa - d \le d\kappa.$$

Combining these two results, the threshold for transition time can be bounded by $\Psi(\tilde{B}_0) + 3 C_0 \Psi(\bar{B}_0) + \frac{9}{\alpha(1-\beta)} C_0 \kappa \le d\kappa + \frac{9}{\alpha(1-\beta)} C_0 \kappa$. Hence, by Theorem 5.2, the linear rate in (20) is achieved when $t \ge d\kappa + \frac{9}{\alpha(1-\beta)} C_0 \kappa$.

For $B_0 = \mu I$, we have $\bar{B}_0 = \frac{1}{L}B_0 = \frac{1}{\kappa}I$ and $\tilde{B}_0 = \nabla^2 f(x_*)^{-\frac{1}{2}} B_0 \nabla^2 f(x_*)^{-\frac{1}{2}} = \mu \nabla^2 f(x_*)^{-1}$. Thus, it holds that $\Psi(\bar{B}_0) = \Psi(\frac{1}{\kappa}I) = \frac{d}{\kappa} - d + d\log\kappa \le d\log\kappa$. Moreover, by Assumptions 2.1 and 2.2, we have $\frac{1}{\kappa}I \preceq \tilde{B}_0 \preceq I$. This implies that

$$\Psi(\tilde{B}_0) = \mathbf{Tr}(\tilde{B}_0) - d - \log\mathbf{Det}(\tilde{B}_0) \le \mathbf{Tr}(I) - d - \log\mathbf{Det}(\frac{1}{\kappa}I) = d\log\kappa.$$

Combining these two results, the threshold for the transition tume can be bounded by $\Psi(\tilde{B}_0) + 3C_0\Psi(\bar{B}_0) + \frac{9}{\alpha(1-\beta)}C_0\kappa \le (1 + 3C_0)d\log\kappa + \frac{9}{\alpha(1-\beta)}C_0\kappa$. Hence, by Theorem 5.2, the linear rate in (20) is satisfied when $t \ge (1 + 3C_0)d\log\kappa + \frac{9}{\alpha(1-\beta)}C_0\kappa$.

# G  Intermediate Results and Proofs in Section 6

## G.1  Intermediate Results

To present our result we first introduce the following function

$$\omega(x) := x - \log(x + 1), \tag{45}$$

which is defined for $x > -1$. Further In the next result, we present some basic properties of the function $\omega(x)$ defined in (45).

**Lemma G.1.** *Recall the definition of function $\omega(x)$ in (45), we have that*

(a) *$\omega(x)$ is increasing function for $x > 0$ and decreasing function for $-1 < x < 0$. Moreover, $\omega(x) \ge 0$ for all $x > -1$.*

(b) *When $x \ge 0$, we have that $\omega(x) \ge \frac{x^2}{2(1+x)}$.*

(c) *When $-1 < x \le 0$, we have that $\omega(x) \ge \frac{x^2}{2+x}$.*

*Proof.* Notice that $\omega'(x) = \frac{x}{1+x}$, we know that when $x > 0$, $\omega'(x) > 0$ and when $-1 < x < 0$, $\omega'(x) < 0$, $\omega'(x) < 0$. Therefore, $\omega(x)$ is increasing function for $x > 0$ and $\omega(x)$ is decreasing function for $-1 < x < 0$. Hence, $\omega(x) \ge \omega(0) = 0$ for all $x > -1$.

$\omega(x) \ge \frac{x^2}{2(1+x)}$ is equivalent to $\omega_1(x) := 2(1 + x)\omega(x) - x^2 \ge 0$. Since $\omega_1'(x) = 2x - 2\log(1 + x) = 2\omega(x) \ge 0$ for all $x > -1$, we know that $\omega_1(x)$ is increasing function for $x > -1$ and hence, $\omega_1(x) \ge \omega_1(0) = 0$ for $x \ge 0$.

$\omega(x) \ge \frac{x^2}{2+x}$ is equivalent to $\omega_2(x) := (2+x)\omega(x) - x^2 \ge 0$. Since $\omega_2'(x) = \frac{x}{1+x} - \log(1 + x) \le 0$ for all $x > -1$, we know that $\omega_2(x)$ is decreasing function for $x > -1$ and hence, $\omega_2(x) \ge \omega_2(0) = 0$ for $x \le 0$. $\qquad\square$

**Proposition G.2.** *Let $\{B_t\}_{t \ge 0}$ be the Hessian approximation matrices generated by the BFGS update in (3). Suppose Assumptions 2.1, 2.2, and 2.3 hold and $f(x_{t+1}) \le f(x_t)$ for any $t \ge 0$. Recall the definition of $\Psi(.)$ in (10) and $C_t$ in (9), we have that*

$$\sum_{i=0}^{t-1} \omega(\rho_i - 1) \le \Psi(\tilde{B}_0) + 2\sum_{i=0}^{t-1} C_i, \qquad \forall t \ge 1, \tag{46}$$

*Proof.* First, taking the trace and determinant on both sides of the equation (13) for any weight matrix $P \in \mathbb{S}_{++}^d$ and using results from Lemma 6.2 of [34], we show that

$$\mathbf{Tr}(\hat{B}_{t+1}) = \mathbf{Tr}(\hat{B}_t) - \frac{\|\hat{B}_t \hat{s}_t\|^2}{\hat{s}_t^\top \hat{B}_t \hat{s}_t} + \frac{\|\hat{y}_t\|^2}{\hat{s}_t^\top \hat{y}_t}, \qquad \mathbf{Det}(\hat{B}_{t+1}) = \mathbf{Det}(\hat{B}_t)\frac{\hat{s}_t^\top \hat{y}_t}{\hat{s}_t^\top \hat{B}_t \hat{s}_t}.$$

Taking the logarithm on both sides of the second equation, we obtain that

$$\log\frac{\hat{s}_t^\top \hat{y}_t}{\hat{s}_t^\top \hat{B}_t \hat{s}_t} = \log\mathbf{Det}(\hat{B}_{t+1}) - \log\mathbf{Det}(\hat{B}_t).$$

Thus, we obtain that

$$\Psi(\hat{B}_{t+1}) - \Psi(\hat{B}_t) = \mathbf{Tr}(\hat{B}_{t+1}) - \mathbf{Tr}(\hat{B}_t) + \log \mathbf{Det}(\hat{B}_t) - \log \mathbf{Det}(\hat{B}_{t+1})$$

$$= \frac{\|\hat{y}_t\|^2}{\hat{s}_t^\top \hat{y}_t} - \frac{\|\hat{B}_t \hat{s}_t\|^2}{\hat{s}_t^\top \hat{B}_t \hat{s}_t} - \log \frac{\hat{s}_t^\top \hat{y}_t}{\hat{s}_t^\top \hat{B}_t \hat{s}_t} = \frac{\|\hat{y}_t\|^2}{\hat{s}_t^\top \hat{y}_t} - \frac{\|\hat{B}_t \hat{s}_t\|^2}{\hat{s}_t^\top \hat{B}_t \hat{s}_t} - \log \frac{\hat{s}_t^\top \hat{y}_t}{\|\hat{s}_t\|^2} - \log \frac{\|\hat{s}_t\|^2}{\hat{s}_t^\top \hat{B}_t \hat{s}_t},$$

which leads to

$$\frac{\|\hat{B}_t \hat{s}_t\|^2}{\hat{s}_t^\top \hat{B}_t \hat{s}_t} - \log \frac{\hat{s}_t^\top \hat{B}_t \hat{s}_t}{\|\hat{s}_t\|^2} - 1 = \Psi(\hat{B}_t) - \Psi(\hat{B}_{t+1}) + \frac{\|\hat{y}_t\|^2}{\hat{s}_t^\top \hat{y}_t} - 1 + \log \frac{\|\hat{s}_t\|^2}{\hat{s}_t^\top \hat{y}_t}.$$

Notice that $\hat{B}_t \hat{s}_t = -\eta_t \hat{g}_t$, $\hat{s}_t^\top \hat{B}_t \hat{s}_t = -\eta_t^2 \hat{g}_t^\top \hat{d}_t$ and $\|\hat{s}_t\|^2 = \eta_t^2 \|\hat{d}_t\|^2$, we have that

$$\frac{\|\hat{g}_t\|^2}{-\hat{g}_t^\top \hat{d}_t} - \log \frac{-\hat{g}_t^\top \hat{d}_t}{\|\hat{d}_t\|^2} - 1 = \Psi(\hat{B}_t) - \Psi(\hat{B}_{t+1}) + \frac{\|\hat{y}_t\|^2}{\hat{s}_t^\top \hat{y}_t} - 1 + \log \frac{\|\hat{s}_t\|^2}{\hat{s}_t^\top \hat{y}_t}.$$

Note that given the fact that $-\hat{g}_t^\top \hat{d}_t = \hat{g}_t^\top \hat{B}_t^{-1} \hat{g}_t > 0$, by using the Cauchy–Schwarz inequality we obtain $\frac{\|\hat{g}_t\|^2}{-\hat{g}_t^\top \hat{d}_t} \geq \frac{-\hat{g}_t^\top \hat{d}_t}{\|\hat{d}_t\|^2}$. Hence, we can write

$$\frac{-\hat{g}_t^\top \hat{d}_t}{\|\hat{d}_t\|^2} - \log \frac{-\hat{g}_t^\top \hat{d}_t}{\|\hat{d}_t\|^2} - 1 \leq \Psi(\hat{B}_t) - \Psi(\hat{B}_{t+1}) + \frac{\|\hat{y}_t\|^2}{\hat{s}_t^\top \hat{y}_t} - 1 + \log \frac{\|\hat{s}_t\|^2}{\hat{s}_t^\top \hat{y}_t}.$$

Now, by selecting the weight matrix as $P = \nabla^2 f(x_*)$, many expressions get simplified and we have $\frac{-\hat{g}_t^\top \hat{d}_t}{\|\hat{d}_t\|^2} = \frac{-g_t^\top d_t}{\|\tilde{d}_t\|^2} = \rho_t$, $\rho_t - \log \rho_t - 1 = \omega(\rho_t - 1)$, and $\hat{B}_t = \tilde{B}_t = \nabla^2 f(x_*)^{-\frac{1}{2}} B_t \nabla^2 f(x_*)^{-\frac{1}{2}}$. Hence, we have

$$\omega(\rho_t - 1) \leq \Psi(\tilde{B}_t) - \Psi(\tilde{B}_{t+1}) + \frac{\|\hat{y}_t\|^2}{\hat{s}_t^\top \hat{y}_t} - 1 + \log \frac{\|\hat{s}_t\|^2}{\hat{s}_t^\top \hat{y}_t}. \qquad (47)$$

Notice that $\frac{\|\hat{y}_t\|^2}{\hat{s}_t^\top \hat{y}_t} \leq 1 + C_t$ for any $t \geq 0$ by using Lemma D.3 (b) with $P = \nabla^2 f(x_*)$ and $\log \frac{\|\hat{s}_t\|^2}{\hat{s}_t^\top \hat{y}_t} = \log \frac{\|\hat{s}_t\|^2}{\hat{s}_t^\top \hat{J}_t \hat{s}_t} \leq \log(1 + C_t) \leq C_t$ for any $t \geq 0$ by using (31) from Lemma A.1. Leveraging these conditions with the inequality (47), we obtain that

$$\omega(\rho_t - 1) \leq \Psi(\tilde{B}_t) - \Psi(\tilde{B}_{t+1}) + 2C_t.$$

Summing both sides of the above inequality from $i = 0$ to $t - 1$, we prove the conclusion

$$\sum_{i=0}^{t-1} \omega(\rho_i - 1) \leq \Psi(\tilde{B}_0) - \Psi(\tilde{B}_t) + 2 \sum_{i=0}^{t-1} C_i \leq \Psi(\tilde{B}_0) + 2 \sum_{i=0}^{t-1} C_i,$$

where the last inequality holds since $\Psi(\tilde{B}_t) \geq 0$. $\qquad \square$

**Lemma G.3.** *Suppose Assumptions 2.1, 2.2, and 2.3 hold and $C_t \leq \frac{1}{6}$ and $\rho_t \geq \frac{7}{8}$ at iteration t, then we have*

$$f(x_t + d_t) \leq f(x_t). \qquad (48)$$

*Proof.* Since assumption 2.3 hold, using Lemma 1.2.4 in [44], we have that

$$|f(y) - f(x) - \nabla f(x)^\top (y - x) - \frac{1}{2}(y - x)^\top \nabla^2 f(x)(y - x)| \leq \frac{M}{6}\|y - x\|^3, \qquad \forall x, y \in \mathbb{R}^d.$$

Setting $x = x_t$ and $y = x_t + d_t$, we have that

$$f(x_t + d_t) - f(x_t) \leq g_t^\top d_t + \frac{1}{2} d_t^\top \nabla^2 f(x_t) d_t + \frac{M}{6}\|d_t\|^3. \qquad (49)$$

Notice that using (33) from Lemma A.1 and the definition of $\rho_t$ in (21), we have that

$$d_t^\top \nabla^2 f(x_t) d_t \leq (1 + C_t) d_t^\top \nabla^2 f(x_*) d_t = -g_t^\top d_t (1 + C_t) \frac{\|\tilde{d}_t\|^2}{-g_t^\top d_t} = -g_t^\top d_t \frac{1 + C_t}{\rho_t}. \qquad (50)$$

Applying Assumption 2.1 with the definition $\tilde{d}_t = \nabla^2 f(x_*)^{\frac{1}{2}} d_t$, we obtain that

$$\|d_t\|^3 \leq \frac{1}{\mu^{\frac{3}{2}}} \|\tilde{d}_t\|^3 = \frac{-g_t^\top d_t}{\mu^{\frac{3}{2}}} \frac{\|\tilde{d}_t\|^2}{-g_t^\top d_t} \|\tilde{d}_t\| = \frac{-g_t^\top d_t}{\mu^{\frac{3}{2}}} \frac{1}{\rho_t} \|\tilde{d}_t\|.$$

Since $-\tilde{g}_t^\top \tilde{d}_t \leq \|\tilde{g}_t\| \|\tilde{d}_t\|$ by Cauchy–Schwarz inequality where $\tilde{g}_t = \nabla^2 f(x_*)^{-\frac{1}{2}} g_t$, we obtain

$$\|\tilde{d}_t\| = \|\tilde{g}_t\| \frac{\|\tilde{d}_t\|}{\|\tilde{g}_t\|} \leq \|\tilde{g}_t\| \frac{\|\tilde{d}_t\|^2}{-\tilde{g}_t^\top \tilde{d}_t} = \frac{1}{\rho_t} \|\tilde{g}_t\|,$$

which leads to

$$\|d_t\|^3 \leq \frac{-g_t^\top d_t}{\mu^{\frac{3}{2}}} \frac{1}{\rho_t} \|\tilde{d}_t\| \leq \frac{-g_t^\top d_t}{\mu^{\frac{3}{2}}} \frac{1}{\rho_t^2} \|\tilde{g}_k\|. \tag{51}$$

By applying Taylor's theorem with Lagrange remainder, there exists $\tilde{\tau}_t \in [0, 1]$ such that

$$\begin{aligned}
f(x_t) &= f(x_*) + \nabla f(x_*)^\top (x_t - x_*) + \frac{1}{2}(x_t - x_*)^\top \nabla^2 f(x_t + \tilde{\tau}_t(x_* - x_t))(x_t - x_*) \\
&= f(x_*) + \frac{1}{2}(x_t - x_*)^\top \nabla^2 f(x_t + \tilde{\tau}_t(x_* - x_t))(x_t - x_*),
\end{aligned} \tag{52}$$

where we used the fact that $\nabla f(x_*) = 0$ in the last equality. Moreover, by the fundamental theorem of calculus, we have

$$\nabla f(x_t) - \nabla f(x_*) = \int_0^1 \nabla^2 f(x_t + \tau(x_* - x_t))(x_t - x_*) \, d\tau = G_t(x_t - x^*),$$

where we use the definition of $G_t$ in (8). Since $\nabla f(x_*) = 0$ and we denote $g_t = \nabla f(x_t)$, this further implies that

$$x_t - x_* = G_t^{-1}(\nabla f(x_t) - \nabla f(x_*)) = G_t^{-1} g_t. \tag{53}$$

Combining (52) and (53) leads to

$$f(x_t) - f(x_*) = \frac{1}{2} g_t^\top G_t^{-1} \nabla^2 f(x_t + \tilde{\tau}_t(x_* - x_t)) G_t^{-1} g_t. \tag{54}$$

Based on (35) in Lemma A.1, we have $\nabla^2 f(x_t + \tilde{\tau}_t(x_* - x_t)) \succeq \frac{1}{1+C_t} G_t$, which implies that

$$G_t^{-1} \nabla^2 f(x_t + \tilde{\tau}_t(x_* - x_t)) G_t^{-1} \succeq \frac{1}{1 + C_t} G_t^{-1}.$$

Moreover, it follows from (34) in Lemma A.1 that $G_t \preceq (1 + C_t) \nabla^2 f(x_*)$, which implies that

$$G_t^{-1} \succeq \frac{1}{1 + C_t} (\nabla^2 f(x_*))^{-1}.$$

Combining the above two conditions, we obtain that

$$G_t^{-1} \nabla^2 f(x_t + \tilde{\tau}_t(x_* - x_t)) G_t^{-1} \succeq \frac{1}{(1 + C_t)^2} (\nabla^2 f(x_*))^{-1},$$

and hence

$$g_t^\top G_t^{-1} \nabla^2 f(x_t + \tilde{\tau}_t(x_* - x_t)) G_t^{-1} g_t \geq \frac{1}{(1 + C_t)^2} g_t^\top (\nabla^2 f(x_*))^{-1} g_t = \frac{1}{(1 + C_t)^2} \|\tilde{g}_t\|^2. \tag{55}$$

Combining (54) and (55) leads to

$$\|\tilde{g}_k\| \leq (1 + C_t) \sqrt{2(f(x_t) - f(x_*))}. \tag{56}$$

Combining (51) and (56) leads to

$$\|d_t\|^3 \leq \frac{-g_t^\top d_t}{\mu^{\frac{3}{2}}} \frac{1}{\rho_t^2} \|\tilde{g}_k\| \leq \frac{-g_t^\top d_t}{\mu^{\frac{3}{2}}} \frac{1}{\rho_t^2} (1 + C_t) \sqrt{2(f(x_t) - f(x_*))}. \tag{57}$$

Leveraging (49), (50) and (57) with the definition of $C_t$ in (9), we have that

$$f(x_t + d_t) - f(x_t) \leq g_t^\top d_t + \frac{1}{2} d_t^\top \nabla^2 f(x_t) d_t + \frac{M}{6} \|d_t\|^3$$

$$= -g_t^\top d_t \left(-1 + \frac{1 + C_t}{2\rho_t} + \frac{M}{6} \frac{1}{\mu^{\frac{3}{2}}} \frac{1}{\rho_t^2} (1 + C_t) \sqrt{2(f(x_t) - f(x_*))}\right) \qquad (58)$$

$$= -g_t^\top d_t \left(-1 + \frac{1 + C_t}{2\rho_t} + \frac{C_t(1 + C_t)}{6\rho_t^2}\right).$$

Notice that $-g_t^\top d_t = -g_t^\top B_t^{-1} g_t > 0$ and when $C_t \leq \frac{1}{6}$ and $\rho_t \geq \frac{7}{8}$, we can verify that

$$\frac{1 + C_t}{2\rho_t} + \frac{C_t(1 + C_t)}{6\rho_t^2} < 1.$$

Therefore, (58) implies the conclusion that

$$f(x_t + d_t) - f(x_t) \leq 0.$$

$\square$

## G.2 Proof of Lemma 6.1

Since $\eta_t = 1$ satisfies Armijo-Wolfe conditions, we know that $\eta_t$ is chosen to be one at iteration $t$ and $x_{t+1} = x_t + d_t$. We have $f(x_{t+1}) \leq f(x_t)$ from Lemma 2.1. Using Taylor's expansion, we have that $f(x_{t+1}) = f(x_t) + g_t^\top d_t + \frac{1}{2} d_t^\top \nabla^2 f(x_t + \hat{\tau}(x_{t+1} - x_t)) d_t$, where $\hat{\tau} \in [0, 1]$. Hence, we have that

$$\hat{p}_t = \frac{f(x_t) - f(x_{t+1})}{-g_t^\top d_t} = \frac{-g_t^\top d_t - \frac{1}{2} d_t^\top \nabla^2 f(x_t + \hat{\tau}(x_{t+1} - x_t)) d_t}{-g_t^\top d_t}$$

$$= 1 - \frac{1}{2} \frac{d_t^\top \nabla^2 f(x_t + \hat{\tau}(x_{t+1} - x_t)) d_t}{-g_t^\top d_t} \geq 1 - \frac{1 + C_t}{2} \frac{d_t^\top \nabla^2 f(x_*) d_t}{-g_t^\top d_t} = 1 - \frac{1 + C_t}{2\rho_t},$$

where we apply the (32) from Lemma A.1 since $f(x_{t+1}) \leq f(x_t)$ and recall the definition of $\rho_t$ in (21). Similarly, using (31) from Lemma A.1 since $f(x_{t+1}) \leq f(x_t)$, we have that

$$\hat{n}_t = \frac{y_t^\top s_t}{-g_t^\top s_t} = \frac{s_t^\top J_t s_t}{-g_t^\top s_t} = \frac{d_t^\top J_t d_t}{-g_t^\top d_t} \geq \frac{1}{1 + C_t} \frac{d_t^\top \nabla^2 f(x_*) d_t}{-g_t^\top d_t} = \frac{1}{(1 + C_t)\rho_t},$$

where we use the fact that $y_t = J_t s_t$ with $J_t$ defined in (8) and $s_t = x_{t+1} - x_t = d_t$. Therefore, we prove the conclusions.

## G.3 Proof of Lemma 6.2

Denote $\bar{x}_{t+1} = x_t + d_t$ and $\bar{s}_t = \bar{x}_{t+1} - x_t = d_t$. Since $\delta_1 \leq \frac{1}{6}$ and $\delta_2 \geq \frac{7}{8}$, we have $f(\bar{x}_{t+1}) \leq f(x_t)$ from Lemma G.3. Using Taylor's expansion, we have that $f(\bar{x}_{t+1}) = f(x_t) + g_t^\top d_t + \frac{1}{2} d_t^\top \nabla^2 f(x_t + \hat{\tau}(\bar{x}_{t+1} - x_t)) d_t$, where $\hat{\tau} \in [0, 1]$. Hence, we have

$$\frac{f(x_t) - f(\bar{x}_{k+1})}{-g_t^\top d_t} = \frac{-g_t^\top d_t - \frac{1}{2} d_t^\top \nabla^2 f(x_t + \hat{\tau}(\bar{x}_{t+1} - x_t)) d_t}{-g_t^\top d_t}$$

$$= 1 - \frac{1}{2} \frac{d_t^\top \nabla^2 f(x_t + \hat{\tau}(\bar{x}_{t+1} - x_t)) d_t}{-g_t^\top d_t} \geq 1 - \frac{1 + C_t}{2} \frac{d_t^\top \nabla^2 f(x_*) d_t}{-g_t^\top d_t} = 1 - \frac{1 + C_t}{2\rho_t},$$

where we apply the (32) from Lemma A.1 since $f(\bar{x}_{t+1}) \leq f(x_t)$. Therefore, when $C_t \leq \delta_1 \leq \sqrt{2(1 - \alpha)} - 1$ and $\rho_t \geq \delta_2 \geq \frac{1}{\sqrt{2(1 - \alpha)}}$, we obtain that $\frac{f(x_t) - f(\bar{x}_{k+1})}{-g_t^\top d_t} \geq 1 - \frac{1 + C_t}{2\rho_t} \geq \alpha$ and unit step size $\eta_t = 1$ satisfies the sufficient condition (5).

Similarly, using (31) from Lemma A.1 since $f(\bar{x}_{t+1}) \leq f(x_t)$ and denote $\bar{g}_{k+1} = \nabla f(\bar{x}_{t+1})$, $\bar{y}_t = \bar{g}_{k+1} - g_t$, we have that

$$\frac{\bar{y}_t^\top \bar{s}_t}{-g_t^\top \bar{s}_t} = \frac{\bar{s}_t^\top J_t \bar{s}_t}{-g_t^\top \bar{s}_t} = \frac{d_t^\top J_t d_t}{-g_t^\top d_t} \geq \frac{1}{1 + C_t} \frac{d_t^\top \nabla^2 f(x_*) d_t}{-g_t^\top d_t} = \frac{1}{(1 + C_t)\rho_t}.$$

Therefore, when $C_t \leq \delta_1 \leq \frac{1}{\sqrt{1-\beta}} - 1$ and $\rho_t \leq \delta_3 = \frac{1}{\sqrt{1-\beta}}$, we obtain that $\frac{\bar{y}_t^\top \bar{s}_t}{-g_t^\top \bar{s}_t} \geq \frac{1}{(1+C_t)\rho_t} \geq 1 - \beta$, which indicates that $\bar{g}_{t+1}^\top d_t = \bar{g}_{t+1}^\top \bar{s}_t = \bar{y}_t^\top \bar{s}_t + g_t^\top \bar{s}_t \geq -g_t^\top \bar{s}_t(1 - \beta) + g_t^\top \bar{s}_t = \beta g_t^\top \bar{s}_t = \beta g_t^\top d_t$. Hence, unit step size $\eta_t = 1$ satisfies the curvature condition (6). Therefore, we prove that when $C_t \leq \delta_1$ and $\delta_2 \leq \rho_t \leq \delta_3$, step size $\eta_t = 1$ satisfies the Armijo-Wolfe conditions (5) and (6).

## G.4  Proof of Lemma 6.3

Since in Theorem 4.1, we already prove that

$$\frac{f(x_t) - f(x_*)}{f(x_0) - f(x_*)} \leq \left(1 - e^{-\frac{\Psi(\bar{B}_0)}{t}} \frac{2\alpha(1-\beta)}{\kappa}\right)^t.$$

This implies that

$$C_t \leq \left(1 - e^{-\frac{\Psi(\bar{B}_0)}{t}} \frac{2\alpha(1-\beta)}{\kappa}\right)^{\frac{t}{2}} C_0.$$

When $t \geq \Psi(\bar{B}_0)$, we obtain that

$$C_t \leq \left(1 - \frac{2\alpha(1-\beta)}{3\kappa}\right)^{\frac{t}{2}} C_0.$$

When $t \geq \frac{3\kappa}{\alpha(1-\beta)} \log \frac{C_0}{\delta_1}$, we obtain that

$$C_t \leq \left(1 - \frac{2\alpha(1-\beta)}{3\kappa}\right)^{\frac{t}{2}} C_0 \leq \delta_1.$$

Therefore, the first claim in (24) follows.

Now define $I_1 = \{t : \rho_t < \delta_2\}$ and $I_2 = \{t : \rho_t > \delta_3\}$, we know that $|I| = |I_1| + |I_2|$. Notice that for $t \in I_1$, we have that $\rho_t - 1 < \delta_2 - 1 < 0$ since $\delta_2 < 1$ and the function $\omega(x)$ defined in (45) is decreasing for $-1 < x < 0$ from (a) in Lemma G.1. Hence, we have that $\sum_{i \in I_1} \omega(\rho_i - 1) \geq \sum_{i \in I_1} \omega(\delta_2 - 1) = \omega(\delta_2 - 1)|I_1|$. Similarly, we have that for $t \in I_2$, we have that $\rho_i - 1 > \delta_3 - 1 > 0$ since $\delta_3 > 1$ and the function $\omega(x)$ is increasing for $x > 0$ from (a) in Lemma G.1. Hence, we have that $\sum_{i \in I_2} \omega(\rho_i - 1) \geq \sum_{i \in I_2} \omega(\delta_3 - 1) = \omega(\delta_3 - 1)|I_2|$. Using (46) from Proposition G.2, we have that $\sum_{i=0}^{t-1} \omega(\rho_i - 1) \leq \Psi(\tilde{B}_0) + 2\sum_{i=0}^{t-1} C_i \leq \Psi(\tilde{B}_0) + 2\sum_{i=0}^{+\infty} C_i$ for any $t \geq 1$. Therefore, we obtain that

$$\Psi(\tilde{B}_0) + 2\sum_{i=0}^{+\infty} C_i \geq \sum_{i=0}^{+\infty} \omega(\rho_i - 1) \geq \sum_{i \in I_1} \omega(\beta_i - 1) + \sum_{i \in I_2} \omega(\beta_i - 1)$$

$$\geq \omega(\delta_2 - 1)|I_1| + \omega(\delta_3 - 1)|I_2| \geq \min\{\omega(\delta_2 - 1), \omega(\delta_3 - 1)\}(|I_1| + |I_2|),$$

which leads to the result

$$|I| = |I_1| + |I_2| \leq \frac{\Psi(\tilde{B}_0) + 2\sum_{i=0}^{+\infty} C_i}{\min\{\omega(\delta_2 - 1), \omega(\delta_3 - 1)\}} = \delta_4 \left(\Psi(\tilde{B}_0) + 2\sum_{i=0}^{+\infty} C_i\right), \qquad (59)$$

where $\delta_4 := \frac{1}{\min\{\omega(\delta_2-1), \omega(\delta_3-1)\}}$. Using the upper bound of $\sum_{i=0}^{+\infty} C_i \leq C_0 \Psi(\bar{B}_0) + \frac{3C_0\kappa}{\alpha(1-\beta)}$ in (44), we prove the second claim in (25).

## G.5  Proof of Theorem 6.4

First, we prove that for any initial point $x_0 \in \mathbb{R}^d$ and any initial Hessian approximation matrix $B_0 \in \mathbb{S}_{++}^d$, the following result holds:

$$\frac{f(x_t) - f(x_*)}{f(x_0) - f(x_*)} \leq \left(\frac{\delta_6 t_0 + \delta_7 \Psi(\tilde{B}_0) + \delta_8 \sum_{i=0}^{+\infty} C_i}{t}\right)^t, \qquad \forall t > t_0,$$

where $t_0$ is defined in (24). We choose the weight matrix as $P = \nabla^2 f(x_*)$ throughout the proof. Using results (41) and (42) from the proof of Proposition 5.1, we obtain that

$$\prod_{i=0}^{t-1} \frac{\cos^2(\hat{\theta}_i)}{\hat{m}_i} \geq e^{-\Psi(\tilde{B}_0) - \sum_{i=0}^{t-1} C_i} \geq e^{-\Psi(\tilde{B}_0) - \sum_{i=0}^{+\infty} C_i}. \tag{60}$$

$$\prod_{i=0}^{t-1} \hat{q}_i \geq 2^t e^{-2\sum_{i=0}^{t-1} C_i} \geq 2^t e^{-2\sum_{i=0}^{+\infty} C_i}. \tag{61}$$

Recall the definition of the set $I = \{t : \rho_t \notin [\delta_2, \delta_3]\}$. Notice that for $t \geq t_0$, define $I_3 = \{t : t \geq t_0, \rho_t \notin [\delta_2, \delta_3]\}$ and $I_4 = \{t : t \geq t_0, \rho_t \in [\delta_2, \delta_3]\}$. Then, we have that

$$\prod_{i=0}^{t-1} \hat{p}_i \hat{n}_i = \prod_{i=0}^{t_0-1} \hat{p}_i \hat{n}_i \prod_{i=t_0}^{t-1} \hat{p}_i \hat{n}_i = \prod_{i=0}^{t_0-1} \hat{p}_i \hat{n}_i \prod_{i\in I_3} \hat{p}_i \hat{n}_i \prod_{i\in I_4} \hat{p}_i \hat{n}_i. \tag{62}$$

From Lemma 2.1, we know $\hat{p}_t \geq \alpha$ and $\hat{n}_t \geq 1 - \beta$ for any $t \geq 0$, which lead to

$$\prod_{i=0}^{t_0-1} \hat{p}_i \hat{n}_i \geq \alpha^{t_0} (1-\beta)^{t_0} = \frac{1}{2^{t_0}} e^{-t_0 \log \frac{1}{2\alpha(1-\beta)}}. \tag{63}$$

$$\prod_{i\in I_3} \hat{p}_i \hat{n}_i \geq \prod_{i\in I_3} \alpha(1-\beta) = \frac{1}{2^{|I_3|}} e^{-|I_3| \log \frac{1}{2\alpha(1-\beta)}} \geq \frac{1}{2^{|I_3|}} e^{-|I| \log \frac{1}{2\alpha(1-\beta)}}$$

$$\geq \frac{1}{2^{|I_3|}} e^{-\delta_4 \left( \Psi(\tilde{B}_0) + 2\sum_{i=0}^{+\infty} C_i \right) \log \frac{1}{2\alpha(1-\beta)}}, \tag{64}$$

where the second inequality holds since $|I_3| \leq |I|$, $\log \frac{1}{2\alpha(1-\beta)} > 0$ and the last inequality holds since (59) from the proof of Lemma 6.3 in Appendix G.4. Notice that when index $i \in I_4$, we have $C_i \leq \delta_1$ from Lemma 6.3 and $\rho_i \in [\delta_2, \delta_3]$. Applying Lemma 6.1 and Lemma 6.2, we know that for $i \in I_4$, $\eta_i = 1$ satisfies the Armijo-Wolfe conditions (5), (6) and we have $\hat{p}_i \geq 1 - \frac{1+C_i}{2\rho_i} > 0$ (since $C_i \leq \delta_1 \leq \frac{1}{6}$, $\rho_i \geq \delta_2 \geq \frac{7}{8}$) and $\hat{n}_i \geq \frac{1}{(1+C_i)\rho_i}$ from (22). Hence, we obtain that

$$\prod_{i\in I_4} \hat{p}_i \hat{n}_i \geq \frac{1}{2^{|I_4|}} \prod_{i\in I_4} (2 - \frac{1+C_i}{\rho_i}) \frac{1}{(1+C_i)\rho_i} \geq \frac{1}{2^{|I_4|}} e^{-\sum_{i\in I_4} C_i} \prod_{i\in I_4} (2 - \frac{1+C_i}{\rho_i}) \frac{1}{\rho_i}, \tag{65}$$

where the last inequality holds since $\frac{1}{1+C_i} \geq e^{-C_i}$. Using the fact that $\log x \geq 1 - \frac{1}{x}$, we obtain

$$\prod_{i\in I_4} (2 - \frac{1+C_i}{\rho_i}) \frac{1}{\rho_i} = \prod_{i\in I_4} e^{\log(2 - \frac{1+C_i}{\rho_i}) - \log \rho_i} \geq \prod_{i\in I_4} e^{1 - \frac{1}{2 - \frac{1+C_i}{\rho_i}} - \log \rho_i}$$

$$= \prod_{i\in I_4} e^{\frac{\rho_i - 1 - C_i}{2\rho_i - 1 - C_i} - \log \rho_i} = \prod_{i\in I_4} e^{\frac{\rho_i - 1 - \log \rho_i + 2(1-\rho_i)\log \rho_i - (1-\log \rho_i)C_i}{2\rho_i - 1 - C_i}}$$

$$= \prod_{i\in I_4} e^{\frac{\omega(\rho_i - 1) + 2(1-\rho_i)\log \rho_i - (1-\log \rho_i)C_i}{2\rho_i - 1 - C_i}} \geq \prod_{i\in I_4} e^{\frac{-2(\rho_i - 1)\log \rho_i - (1-\log \rho_i)C_i}{2\rho_i - 1 - C_i}} \tag{66}$$

$$= \prod_{i\in I_4} e^{-\frac{2(\rho_i - 1)\log \rho_i + (1-\log \rho_i)C_i}{2\rho_i - 1 - C_i}} \geq \prod_{i\in I_4} e^{-\frac{2(\rho_i - 1)\log \rho_i + (1-\log \delta_2)C_i}{2\delta_2 - 1 - \delta_1}},$$

where the second inequality holds since $\omega(\rho_i - 1) \geq 0$ and the third inequality holds since $\rho_i \geq \delta_2$ due to $i \in I_4$ and $C_i \leq \delta_1$ due to $i \geq t_0$ and Lemma 6.3. Notice that $2\rho_i - 1 - C_i \geq 2\delta_2 - 1 - \delta_1 > 0$ for all $i \in I_4$ since $C_i \leq \delta_1 \leq \frac{1}{6}$ and $\rho_i \geq \delta_2 \geq \frac{7}{8}$.

When $\rho_i \geq 1$, using $\log \rho_i \leq \rho_i - 1$, (b) in Lemma G.1 and $\rho_i \leq \delta_3$ due to $i \in I_4$, we have that

$$(\rho_i - 1) \log \rho_i \leq (\rho_i - 1)^2 \leq 2\rho_i \omega(\rho_i - 1) \leq 2\delta_3 \omega(\rho_i - 1). \tag{67}$$

Similarly, when $\rho_i < 1$, using $\log \rho_i \geq 1 - \frac{1}{\rho_i}$, (c) in Lemma G.1 and $\rho_i \geq \delta_2$ due to $i \in I_4$, we have

$$(\rho_i - 1) \log \rho_i \leq \frac{(\rho_i - 1)^2}{\rho_i} \leq \frac{\rho_i + 1}{\rho_i} \omega(\rho_i - 1) \leq (1 + \frac{1}{\delta_2}) \omega(\rho_i - 1). \tag{68}$$

Combining (66), (67) and (68), we obtain that

$$
\prod_{i\in I_4}(2-\frac{1+C_i}{\rho_i})\frac{1}{\rho_i}
$$

$$
\geq \prod_{i\in I_4} e^{-\frac{2(\rho_i-1)\log\rho_i+(1-\log\delta_2)C_i}{2\delta_2-1-\delta_1}} = \prod_{i\in I_4} e^{-\frac{2(\rho_i-1)\log\rho_i}{2\delta_2-1-\delta_1}} \prod_{i\in I_4} e^{-\frac{(1-\log\delta_2)C_i}{2\delta_2-1-\delta_1}}
$$

$$
= \prod_{i\in I_4,\rho_i<1} e^{-\frac{2(\rho_i-1)\log\rho_i}{2\delta_2-1-\delta_1}} \prod_{i\in I_4,\rho_i\geq1} e^{-\frac{2(\rho_i-1)\log\rho_i}{2\delta_2-1-\delta_1}} \prod_{i\in I_4} e^{-\frac{(1-\log\delta_2)C_i}{2\delta_2-1-\delta_1}}
$$

$$
\geq \prod_{i\in I_4,\rho_i<1} e^{-\frac{2(1+\frac{1}{\delta_2})\omega(\rho_i-1)}{2\delta_2-1-\delta_1}} \prod_{i\in I_4,\rho_i\geq1} e^{-\frac{4\delta_3\omega(\rho_i-1)}{2\delta_2-1-\delta_1}} \prod_{i\in I_4} e^{-\frac{(1-\log\delta_2)C_i}{2\delta_2-1-\delta_1}} \tag{69}
$$

$$
= e^{-\frac{2+\frac{2}{\delta_4}}{2\delta_2-1-\delta_1}\sum_{i\in I_2,\rho_i<1}\omega(\rho_i-1)-\frac{4\delta_3}{2\delta_2-1-\delta_1}\sum_{i\in I_4,\rho_i\geq1}\omega(\rho_i-1)-\frac{1-\log\delta_2}{2\delta_2-1-\delta_1}\sum_{i\in I_4}C_i}
$$

$$
\geq e^{-\delta_5\left(\sum_{i\in I_4,\rho_i<1}\omega(\rho_i-1)+\sum_{i\in I_4,\rho_i\geq1}\omega(\rho_i-1)\right)-\frac{1-\log\delta_2}{2\delta_2-1-\delta_1}\sum_{i\in I_4}C_i}
$$

$$
= e^{-\delta_5\sum_{i\in I_4}\omega(\rho_i-1)-\frac{1-\log\delta_2}{2\delta_2-1-\delta_1}\sum_{i\in I_4}C_i}
$$

where $\delta_5 = \max\{\frac{2+\frac{2}{\delta_2}}{2\delta_2-1-\delta_1}, \frac{4\delta_3}{2\delta_2-1-\delta_1}\}$. Combining (65) and (69), we obtain that

$$
\prod_{i\in I_4}\hat{p}_i\hat{n}_i \geq \frac{1}{2^{|I_4|}}e^{-\sum_{i\in I_4}C_i}\prod_{i\in I_4}(2-\frac{1+C_i}{\rho_i})\frac{1}{\rho_i}
$$

$$
\geq \frac{1}{2^{|I_4|}}e^{-\delta_5\sum_{i\in I_4}\omega(\rho_i-1)-(1+\frac{1-\log\delta_2}{2\delta_2-1-\delta_1})\sum_{i\in I_4}C_i}
$$

$$
\geq \frac{1}{2^{|I_4|}}e^{-\delta_5\sum_{i=0}^{+\infty}\omega(\rho_i-1)-\frac{2\delta_2-\delta_1-\log\delta_2}{2\delta_2-1-\delta_1}\sum_{i=0}^{+\infty}C_i} \tag{70}
$$

$$
\geq \frac{1}{2^{|I_4|}}e^{-\delta_5\left(\Psi(\tilde{B}_0)+2\sum_{i=0}^{+\infty}C_i\right)-\frac{2\delta_2-\delta_1-\log\delta_2}{2\delta_2-1-\delta_1}\sum_{i=0}^{+\infty}C_i},
$$

where the last inequality is due to (46) from Lemma G.1. Combining (62), (63), (64) and (70), we obtain that

$$
\prod_{i=0}^{t-1}\hat{p}_i\hat{n}_i = \prod_{i=0}^{t_0-1}\hat{p}_i\hat{n}_i \prod_{i\in I_3}\hat{p}_i\hat{n}_i \prod_{i\in I_4}\hat{p}_i\hat{n}_i \tag{71}
$$

$$
\geq \frac{1}{2^t}e^{-\left(t_0\log\frac{1}{2\alpha(1-\beta)}+(\delta_4\log\frac{1}{2\alpha(1-\beta)}+\delta_5)\Psi(\tilde{B}_0)+(2\delta_4\log\frac{1}{2\alpha(1-\beta)}+2\delta_5+\frac{2\delta_2-\delta_1-\log\delta_2}{2\delta_2-1-\delta_1})\sum_{i=0}^{+\infty}C_i\right)}.
$$

Leveraging (60), (61), (71) with (15) from Proposition 3.1, we prove that

$$
\frac{f(x_t)-f(x_*)}{f(x_0)-f(x_*)} \leq \left[1-\left(\prod_{i=0}^{t-1}\hat{p}_i\hat{q}_i\hat{n}_i\frac{\cos^2(\hat{\theta}_i)}{\hat{m}_i}\right)^{\frac{1}{t}}\right]^t = \left[1-\left(\prod_{i=0}^{t-1}\hat{p}_i\hat{n}_i\prod_{i=0}^{t-1}\hat{q}_i\prod_{i=0}^{t-1}\frac{\cos^2(\hat{\theta}_i)}{\hat{m}_i}\right)^{\frac{1}{t}}\right]^t
$$

$$
\leq \left(1-e^{-\frac{t_0\log\frac{1}{2\alpha(1-\beta)}+(1+\delta_4\log\frac{1}{2\alpha(1-\beta)}+\delta_5)\Psi(\tilde{B}_0)+(3+2\delta_4\log\frac{1}{2\alpha(1-\beta)}+2\delta_5+\frac{2\delta_2-\delta_1-\log\delta_2}{2\delta_2-1-\delta_1})\sum_{i=0}^{+\infty}C_i}{t}}\right)^t
$$

$$
= \left(1-e^{-\frac{\delta_6 t_0+\delta_7\Psi(\tilde{B}_0)+\delta_8\sum_{i=0}^{+\infty}C_i}{t}}\right)^t \leq \left(\frac{\delta_6 t_0+\delta_7\Psi(\tilde{B}_0)+\delta_8\sum_{i=0}^{+\infty}C_i}{t}\right)^t,
$$

where the inequality is due to the fact that $1-e^{-x}\leq x$ for any $x\in\mathbb{R}$ and $\delta_6,\delta_7,\delta_8$ are defined in Theorem 6.4. Hence, we prove that

$$
\frac{f(x_t)-f(x_*)}{f(x_0)-f(x_*)} \leq \left(\frac{\delta_6 t_0+\delta_7\Psi(\tilde{B}_0)+\delta_8\sum_{i=0}^{+\infty}C_i}{t}\right)^t, \qquad \forall t > t_0. \tag{72}
$$

Using (44) from the proof of Theorem 5.2 in Appendix F.2, we have that

$$\sum_{i=0}^{+\infty} C_i \leq C_0 \Psi(\bar{B}_0) + \frac{3C_0 \kappa}{\alpha(1-\beta)}. \tag{73}$$

Notice that from (24) in Lemma 6.3, we have that

$$t_0 = \max\{\Psi(\bar{B}_0), \frac{3\kappa}{\alpha(1-\beta)} \log \frac{C_0}{\delta_1}\} \leq \Psi(\bar{B}_0) + \frac{3\kappa}{\alpha(1-\beta)} \log \frac{C_0}{\delta_1}. \tag{74}$$

Leveraging (72), (73) and (74), we prove the conclusion.

### G.6  Proof of Corollary 6.5

Using the fact that for $B_0 = LI$, we have $\Psi(\bar{B}_0) = 0$ and $\Psi(\tilde{B}_0) \leq d\kappa$, and for the case that $B_0 = \mu I$, we have $\Psi(\bar{B}_0) \leq d \log \kappa$, and $\Psi(\tilde{B}_0) \leq d \log \kappa$, we obtain the corresponding superlinear results for these two conditions.

### G.7  Specific Values of $\{\delta_i\}_{i=1}^{8}$

As we stated before, all the $\{\delta_i\}_{i=1}^{8}$ are universal constants that only depend on line search parameters $\alpha$ and $\beta$. We can choose specific values of $\alpha$ and $\beta$ to make definitions of $\{\delta_i\}_{i=1}^{8}$ more clear. If we pick $\alpha = \frac{1}{4}$ and $\beta = \frac{3}{4}$, we have that

$$\delta_1 = \frac{1}{6}, \quad \delta_2 = \frac{7}{8}, \quad \delta_3 = 2, \quad \delta_4 = 118, \quad \delta_5 = 14, \quad \delta_6 = \log 8, \quad \delta_7 = 260, \quad \delta_8 = 524.$$

## H  Complexity of BFGS with the Initialization $B_0 = cI$

Recall that $c \in [\mu, L]$ by our choice of $c$ in Remark 6.2. If we choose $B_0 = cI$, then $\Psi(\bar{B}_0) = \Psi(\frac{c}{L}I) = \frac{c}{L}d - d + d \log \frac{L}{c}$. Moreover, we have $\Psi(\tilde{B}_0) = \Psi(c\nabla^2 f(x_*)^{-1}) = c\mathbf{Tr}(\nabla^2 f(x_*)^{-1}) - d - \log \mathbf{Det}(c\nabla^2 f(x_*)^{-1})$, which is determined by the Hessian matrix $\nabla^2 f(x_*)^{-1}$. In this case, one can use the upper bounds $\Psi(\bar{B}_0) = d(\frac{c}{L} - 1 + \log \frac{L}{c})$ and $\Psi(\tilde{B}_0) = \mathbf{Tr}(c\nabla^2 f(x_*)^{-1}) - d - \log \mathbf{Det}(c\nabla^2 f(x_*)^{-1}) \leq d(\frac{c}{\mu} - 1 + \log \frac{L}{c})$ to simplify the expressions.

Applying these values of $\Psi(\bar{B}_0)$ and $\Psi(\tilde{B}_0)$ to our linear convergence result in Theorem 4.1 and the superlinear convergence result in Theorem 6.4, we can obtain the following convergence guarantees for $B_0 = cI$:

- For $t \geq d(\frac{c}{L} - 1 + \log \frac{L}{c})$, we have $\frac{f(x_t) - f(x_*)}{f(x_0) - f(x_*)} \leq \left(1 - \frac{2\alpha(1-\beta)}{3\kappa}\right)^t$;

- For $t = \Omega(d(\frac{c}{\mu} - 1 + \log \frac{L}{c}) + C_0 d(\frac{c}{L} - 1 + \log \frac{L}{c}) + C_0\kappa)$, we have $\frac{f(x_t) - f(x_*)}{f(x_0) - f(x_*)} \leq \left(\mathcal{O}\left(\frac{d(\frac{c}{\mu} - 1 + \log \frac{L}{c}) + C_0 d(\frac{c}{L} - 1 + \log \frac{L}{c}) + C_0\kappa}{t}\right)\right)^t$.

Moreover, we can derive similar iteration complexity bounds following the same arguments as in Section I. We also include the performance of BFGS with $B_0 = cI$ in our numerical experiments as presented in Figure 1. We observe that the performance of BFGS with $B_0 = cI$ is very similar to the convergence curve of BFGS with $B_0 = \mu I$ in our numerical experiments.

## I  Proof of Iteration Complexity

When $B_0 = LI$, if we regard the line search parameters $\alpha$ and $\beta$ as absolute constants, the first result established in Corollary 4.2 leads to a global complexity of $\mathcal{O}(\kappa \log \frac{1}{\epsilon})$, which is on par with gradient descent. Moreover, the first result in Corollary 5.3 implies a complexity of $\mathcal{O}\left((d + C_0)\kappa + \log \frac{1}{\epsilon}\right)$, where the first term represents the number of iterations required to attain the linear rate in (20), and the second term represents the additional number of iterations needed to achieve the desired accuracy

$\epsilon$ from the condition number-independent linear rate. For the analysis of the superlinear convergence rate, we denote that $\Omega_L = d\kappa + C_0\kappa$. From the first result in Corollary 6.5, we have that

$$\frac{f(x_t) - f(x_*)}{f(x_0) - f(x_*)} \leq (\frac{\Omega_L}{t})^t$$

Let $T_*$ be the number such that the inequality $(\frac{\Omega_L}{t})^t \leq \epsilon$ above becomes equality. we have

$$\log \frac{1}{\epsilon} = T_* \log \frac{T_*}{\Omega_L} \leq T_*(\frac{T_*}{\Omega_L} - 1),$$

$$T_* \geq \frac{\Omega_L + \sqrt{\Omega_L^2 + 4\Omega_L \log \frac{1}{\epsilon}}}{2}.$$

Hence, we have that

$$\log \frac{1}{\epsilon} = T_* \log \frac{T_*}{\Omega_L} \geq T_* \log \frac{\Omega_L + \sqrt{\Omega_L^2 + 4\Omega_L \log \frac{1}{\epsilon}}}{2\Omega_L} \geq T_* \log \left( \frac{1}{2} + \sqrt{\frac{1}{4} + \frac{\log \frac{1}{\epsilon}}{\Omega_L}} \right),$$

$$T_* \leq \frac{\log \frac{1}{\epsilon}}{\log \left( \frac{1}{2} + \sqrt{\frac{1}{4} + \frac{\log \frac{1}{\epsilon}}{\Omega_L}} \right)}.$$

Hence, to reach the accuracy of $\epsilon$, we need the number of iterations $t$ to be at least

$$t \geq \frac{\log \frac{1}{\epsilon}}{\log \left( \frac{1}{2} + \sqrt{\frac{1}{4} + \frac{1}{\Omega_L} \log \frac{1}{\epsilon}} \right)}.$$

Therefore, the iteration complexity for the case of $B_0 = LI$ is

$$\mathcal{O}\left( \min \left\{ \kappa \log \frac{1}{\epsilon}, (d + C_0)\kappa + \log \frac{1}{\epsilon}, \frac{\log \frac{1}{\epsilon}}{\log \left( \frac{1}{2} + \sqrt{\frac{1}{4} + \frac{1}{d\kappa + C_0\kappa} \log \frac{1}{\epsilon}} \right)} \right\} \right).$$

Similarly, in this case of $B_0 = \mu I$, the second result in Corollary 4.2 establishes a global complexity of $\mathcal{O}\left( d \log \kappa + \kappa \log \frac{1}{\epsilon} \right)$, where the first term represents the number of iterations before the linear convergence rate in (19) begins, and the second term arises from the linear rate itself. Additionally, following the same argument, the second result in Corollary 5.3 indicates a complexity of $\mathcal{O}(C_0 d \log \kappa + C_0\kappa + \log \frac{1}{\epsilon})$. Here, the first term accounts for the wait time until the convergence rate takes effect, and the second term is associated with the condition number-independent linear rate. For the superlinear convergence rate, when $B_0 = \mu I$, to reach the accuracy of $\epsilon$, we need the number of iterations $t$ to be at least

$$t \geq \frac{\log \frac{1}{\epsilon}}{\log \left( \frac{1}{2} + \sqrt{\frac{1}{4} + \frac{1}{\Omega_\mu} \log \frac{1}{\epsilon}} \right)},$$

where $\Omega_\mu = C_0 d \log \kappa + C_0\kappa$. The proof is the same as the proof for the case of $B_0 = LI$. Therefore, the iteration complexity for the case of $B_0 = \mu I$ is

$$\mathcal{O}\left( \min \left\{ d \log \kappa + \kappa \log \frac{1}{\epsilon}, C_0(d \log \kappa + \kappa) + \log \frac{1}{\epsilon}, \frac{\log \frac{1}{\epsilon}}{\log \left( \frac{1}{2} + \sqrt{\frac{1}{4} + \frac{1}{C_0(d \log \kappa + \kappa)} \log \frac{1}{\epsilon}} \right)} \right\} \right).$$

## J  Log Bisection Algorithm for Weak Wolfe Conditions

**Algorithm 1** Log Bisection Algorithm for Weak Wolfe Conditions

---

**Require:** Initial step size $\eta^{(0)} = 1$, $\eta_{min}^{(0)} = 0$, $\eta_{max}^{(0)} = +\infty$

1: **for** $i = 0, 1, 2, \ldots$ **do**
2:    **if** $f(x_t + \eta^{(i)} d_t) > f(x_t) + \alpha \eta^{(i)} \nabla f(x_t)^\top d_t$ **then**
3:       Set $\eta_{max}^{(i+1)} = \eta^{(i)}$ and $\eta_{min}^{(i+1)} = \eta_{min}^{(i)}$
4:       **if** $\eta_{min}^{(i)} = 0$ **then**
5:          $\eta^{(i+1)} = (\frac{1}{2})^{2^{i+1}-1}$
6:       **else**
7:          $\eta^{(i+1)} = \sqrt{\eta_{max}^{(i+1)} \eta_{min}^{(i+1)}}$
8:       **end if**
9:    **else if** $\nabla f(x_t + \eta^{(i)} d_t)^\top d_t < \beta \nabla f(x_t)^\top d_t$ **then**
10:      Set $\eta_{max}^{(i+1)} = \eta_{max}^{(i)}$ and $\eta_{min}^{(i+1)} = \eta^{(i)}$
11:      **if** $\eta_{max}^{(i)} = +\infty$ **then**
12:         $\eta^{(i+1)} = 2^{2^{i+1}-1}$
13:      **else**
14:         $\eta^{(i+1)} = \sqrt{\eta_{max}^{(i+1)} \eta_{min}^{(i+1)}}$
15:      **end if**
16:    **else**
17:      Return $\eta^{(i)}$
18:    **end if**
19: **end for**

---

# K   Results and Discussion on the Bisection Scheme for Line Search in Section 7

## K.1   Proof of Lemma K.1

First, we present major results concerning the complexity of the bisection method, which specifies a range of values that meet the conditions in (5) and (6).

**Lemma K.1.** *Suppose that Assumptions 2.1, 2.2 and 2.3 hold. Recall the definition of $\rho_t$ in (21) and $C_t$ in (9). At iteration t, there is unique $\eta_r > 0$ such that the sufficient decrease condition (5) is equity for $\eta_r$, i.e.,*

$$f(x_t + \eta_r d_t) = f(x_t) + \alpha \eta_r \nabla f(x_t)^\top d_t. \tag{75}$$

*Then, $\eta_t$ satisfies the sufficient decrease condition (5) if and only if $\eta_t \leq \eta_r$. We also have that*

$$\frac{2(1-\alpha)}{1+C_t}\rho_t \leq \eta_r \leq 2(1-\alpha)(1+C_t)\rho_t. \tag{76}$$

*Similarly, there is also unique $\eta_l > 0$ such that the curvature condition (6) is equity for $\eta_l$, i.e.,*

$$\nabla f(x_t + \eta_l d_t)^\top d_t = \beta \nabla f(x_t)^\top d_t. \tag{77}$$

*Then, $\eta_t$ satisfies the curvature condition (6) if and only if $\eta_t \geq \eta_l$. Moreover, we have that*

$$\frac{\eta_r}{\eta_l} \geq 1 + \frac{\beta - \alpha}{(1-\beta)(1+2C_t)} > 1. \tag{78}$$

*Proof.* Notice that Assumption 2.1 indicates that the objective function $f(x)$ is strongly convex. Consider function $h_1(\eta) = f(x_t + \eta d_t) - \alpha \eta \nabla f(x_t)^\top d_t$. We observe that this function $h_1(\eta)$ is strongly convex and $h_1(0) = f(x_t)$, $h_1'(0) < 0$. Hence, there is unique $\eta_r > 0$ such that $h_1(\eta_r) = f(x_t)$ and $\eta_t \leq \eta_r$ if and only if $f(x_t + \eta_t d_t) \leq f(x_t) + \alpha \eta_t \nabla f(x_t)^\top d_t$.

Denote that $\bar{x}_{t+1} = x_t + \eta_r d_t$. We know that $f(\bar{x}_{t+1}) - f(x_t) = \alpha \eta_r g_t^\top d_t$. Since $f(\bar{x}_{t+1}) - f(x_t) = \eta_r g_t^\top d_t + \frac{1}{2}\eta_r^2 d_t^\top \nabla^2 f(x_t + \tau(\bar{x}_{t+1} - x_t))d_t$ for $\tau \in (0, 1)$, we have that

$$\eta_r g_t^\top d_t + \frac{1}{2}\eta_r^2 d_t^\top \nabla^2 f(x_t + \tau(\bar{x}_{t+1} - x_t))d_t = \alpha \eta_r g_t^\top d_t,$$

$$\eta_r = 2(1-\alpha)\frac{-g_t^\top d_t}{d_t^\top \nabla^2 f(x_t + \tau(\bar{x}_{t+1} - x_t))d_t}.$$

which leads to

$$\eta_r = 2(1-\alpha)\frac{-g_t^\top d_t}{d_t^\top \nabla^2 f(x_t + \tau(\bar{x}_{t+1} - x_t))d_t} \leq 2(1-\alpha)(1+C_t)\frac{-g_t^\top d_t}{d_t^\top \nabla^2 f(x_*)d_t}$$

$$= 2(1-\alpha)(1+C_t)\frac{-g_t^\top d_t}{\|\tilde{d}_t\|^2} = 2(1-\alpha)(1+C_t)\rho_t.$$

$$\eta_r = 2(1-\alpha)\frac{-g_t^\top d_t}{d_t^\top \nabla^2 f(x_t + \tau(\bar{x}_{t+1} - x_t))d_t} \geq \frac{2(1-\alpha)}{1+C_t}\frac{-g_t^\top d_t}{d_t^\top \nabla^2 f(x_*)d_t} = \frac{2(1-\alpha)}{1+C_t}\rho_t.$$

where we use the (32) from Lemma A.1 and the fact that $f(\bar{x}_{t+1}) = f(x_t) + \alpha\eta_r g_t^\top d_t \leq f(x_t)$. Hence, we prove the results in (76).

Similarly, consider function $h_2(\eta) = \nabla f(x_t + \eta d_t)^\top d_t$. We observe that this function $h_2(\eta)$ is strictly increasing function for $\eta \geq 0$ and $h_2(0) = \nabla f(x_t)^\top d_t < \beta\nabla f(x_t)^\top d_t$, $h_2(\eta_{exact}) = \nabla f(x_t + \eta_{exact}d_t)^\top d_t = 0 > \beta\nabla f(x_t)^\top d_t$ where $\eta_{exact} := \arg\min_{\eta>0} f(x_t + \eta d_t)$ is the exact line search step size satisfying $\nabla f(x_t + \eta_{exact}d_t)^\top d_t = 0$. Hence, there is unique $\eta_l \in (0, \eta_{exact})$ such that $h_2(\eta_l) = \beta\nabla f(x_t)^\top d_t$ and $\eta_t \geq \eta_l$ if and only if $\nabla f(x_t + \eta_t d_t)^\top d_t \geq \beta\nabla f(x_t)^\top d_t$.

Notice that

$$f(x_t + \eta_r d_t) = f(x_t) + \alpha\eta_r \nabla f(x_t)^\top d_t.$$

Using mean value theorem, we know there exists $\bar{\eta} \in (0, \eta_r)$ such that

$$f(x_t + \eta_r d_t) = f(x_t) + \eta_r \nabla f(x_t + \bar{\eta}d_t)^\top d_t.$$

The above two equities indicates that

$$\nabla f(x_t + \bar{\eta}d_t)^\top d_t = \alpha\nabla f(x_t)^\top d_t.$$

Recall that

$$\nabla f(x_t + \eta_l d_t)^\top d_t = \beta\nabla f(x_t)^\top d_t.$$

Combing the above two equities, we obtain that

$$(\nabla f(x_t + \bar{\eta}d_t) - \nabla f(x_t + \eta_l d_t))^\top d_t = -\nabla f(x_t)^\top d_t(\beta - \alpha).$$

Using mean value theorem again, we know there exists $\tilde{\eta} \in (\eta_l, \bar{\eta})$ such that

$$(\nabla f(x_t + \bar{\eta}d_t) - \nabla f(x_t + \eta_l d_t))^\top d_t = (\bar{\eta} - \eta_l)d_t^\top \nabla^2 f(x_t + \tilde{\eta}d_t)d_t.$$

Leveraging the above two equities, we obtain that

$$\bar{\eta} - \eta_l = (\beta - \alpha)\frac{-\nabla f(x_t)^\top d_t}{d_t^\top \nabla^2 f(x_t + \tilde{\eta}d_t)d_t}.$$

Notice that $\bar{\eta} \leq \eta_r$, we have that

$$\eta_r - \eta_l \geq \bar{\eta} - \eta_l = (\beta - \alpha)\frac{-\nabla f(x_t)^\top d_t}{d_t^\top \nabla^2 f(x_t + \tilde{\eta}d_t)d_t}. \tag{79}$$

Recall the definition of $\eta_l$ in (77), we have that

$$(\nabla f(x_t + \eta_l d_t) - \nabla f(x_t))^\top d_t = -(1-\beta)\nabla f(x_t)^\top d_t.$$

Notice that there exists $\hat{\eta} \in (0, \eta_l)$, such that

$$(\nabla f(x_t + \eta_l d_t) - \nabla f(x_t))^\top d_t = \eta_l d_t^\top \nabla^2 f(x_t + \hat{\eta}d_t)d_t.$$

Combing the above two equities, we obtain that

$$\eta_l = \frac{-(1-\beta)\nabla f(x_t)^\top d_t}{d_t^\top \nabla^2 f(x_t + \hat{\eta}d_t)d_t}. \tag{80}$$

Leveraging (79) and (80), we have that

$$\frac{\eta_r}{\eta_l} = 1 + \frac{\eta_r - \eta_l}{\eta_l} \geq 1 + \frac{(\beta - \alpha)d_t^\top \nabla^2 f(x_t + \hat{\eta}d_t)d_t}{(1 - \beta)d_t^\top \nabla^2 f(x_t + \tilde{\eta}d_t)d_t}.$$

Recall that $\bar{x}_{t+1} = x_t + \eta_r d_t$ and notice that $\hat{\eta} \leq \eta_r$, $\tilde{\eta} \leq \eta_r$. We have that $x_t + \hat{\eta}d_t = x_t + \hat{\tau}(\bar{x}_{t+1} - x_t)$ and $x_t + \tilde{\eta}d_t = x_t + \tilde{\tau}(\bar{x}_{t+1} - x_t)$ with $\hat{\tau} = \frac{\hat{\eta}}{\eta_r} \in (0, 1)$ and $\tilde{\tau} = \frac{\tilde{\eta}}{\eta_r} \in (0, 1)$. Since $f(\bar{x}_{t+1}) = f(x_t + \eta_r d_t) = f(x_t) + \alpha\eta_r \nabla f(x_t)^\top d_t \leq f(x_t)$, applying (36) in Lemma A.1, we prove the conclusion that

$$\frac{\eta_r}{\eta_l} \geq 1 + \frac{(\beta - \alpha)d_t^\top \nabla^2 f(x_t + \hat{\eta}d_t)d_t}{(1 - \beta)d_t^\top \nabla^2 f(x_t + \tilde{\eta}d_t)d_t}$$

$$= 1 + \frac{(\beta - \alpha)d_t^\top \nabla^2 f(x_t + \hat{\tau}(\bar{x}_{t+1} - x_t))d_t}{(1 - \beta)d_t^\top \nabla^2 f(x_t + \tilde{\tau}(\bar{x}_{t+1} - x_t))d_t} \geq 1 + \frac{\beta - \alpha}{(1 - \beta)(1 + 2C_t)}.$$

$\square$

### K.2 Bound on the Number of Inner Loops

**Proposition K.2.** *Suppose that Assumptions 2.1, 2.2 and 2.3 hold. Consider the BFGS method with inexact line search defined in (5) and (6) and we choose the step size $\eta_t$ according to Algorithm 1. At iteration $t$, denote $\lambda_t$ as the number of loops in Algorithm 1 to terminate and return the $\eta_t$ satisfying the Wolfe conditions (5) and (6). Then $\lambda_t$ is finite and upper bounded by*

$$\lambda_t \leq 2 + \log_2\left(1 + \frac{(1 - \beta)(1 + 2C_t)}{\beta - \alpha}\right)$$
$$+ 2\log_2\left(1 + \log_2\left(2(1 - \alpha)(1 + C_t)\right) + \max\{\log_2 \rho_t, \log_2 \frac{1}{\rho_t}\}\right). \tag{81}$$

*Proof.* At the first iteration, if $\eta^{(0)} = 1$ satisfies the weak Wolfe conditions (5) and (6), the algorithm terminates and returns the unit step size $\eta_t = 1$. In this case, we have that $\lambda_t = 1$.

Suppose that at the first iteration, $\eta^{(0)} = 1$ doesn't satisfy the sufficient decrease condition (5) but satisfies the curvature condition (6), we have that $\eta_{max}^{(1)} = +\infty$, $\eta_{min}^{(1)} = 1$ and $\eta^{(1)} = 2$. Assume that in the Algorithm 1, $\eta_{max}^{(i)}$ is never set to a finite value and the algorithm never returns. This means that the condition in line 2 is never satisfied, and as a result, we keep repeating steps in line 12. Thus, $\eta^{(i)} = 2^{2^i - 1}$ and since the condition in line 2 is never satisfied, we always have that $f(x_t + \eta^{(i)}d_t) \leq f(x_t) + \alpha\eta^{(i)}\nabla f(x_t)^\top d_t$. Notice that $\lim_{i \to \infty} \eta^{(i)} \to +\infty$ and $\nabla f(x_t)^\top d_t < 0$. We obtain that $\lim_{i \to \infty} f(x_t + \eta^{(i)}d_t) \to -\infty$, which is a contradiction since $f$ is strongly convex.

Hence, at some point, either the algorithm finds an admissible step size and returns, or $\eta_{max}^{(i)}$ must become finite. Suppose that this happens at iteration $K_1 \geq 1$ of the loop in Algorithm 1. Then, we know that $\eta^{(K_1)} = 2^{2^{K_1} - 1}$. In the first case that the algorithm finds an admissible step size and returns $\eta^{K_1}$, $\eta^{K_1}$ satisfies the Armijo-Wolfe conditions and therefore $\eta^{K_1} \leq \eta_r$. Using the upper bound result in (76) from Lemma K.1, we obtain that $\eta^{(K_1)} = 2^{2^{K_1} - 1} \leq \eta_r \leq 2(1 - \alpha)(1 + C_t)\rho_t$, which leads to

$$\lambda_t = K_1 \leq \log_2\left(1 + \log_2\left(2(1 - \alpha)(1 + C_t)\rho_t\right)\right). \tag{82}$$

In the second case that $\eta_{max}^{(i)}$ becomes finite but the algorithm does not terminate, we have that $\eta^{(K_1 - 1)}$ satisfies the sufficient condition (5) and $\eta^{(K_1 - 1)} \leq \eta_r$. Similarly, this implies that

$$K_1 \leq 1 + \log_2\left(1 + \log_2\left(2(1 - \alpha)(1 + C_t)\rho_t\right)\right). \tag{83}$$

Then, we further go through the log bisection process. Notice that for any iteration $i > K_1$, the sequence $\eta_{max}^{(i)}$ is finite and non-increasing and the sequence $\eta_{min}^{(i)} \geq 1$ and non-decreasing. The log bisection process indicates that

$$\log_2 \frac{\eta_{max}^{(i+1)}}{\eta_{min}^{(i+1)}} = \frac{1}{2}\log_2 \frac{\eta_{max}^{(i)}}{\eta_{min}^{(i)}}, \qquad \forall i > K_1. \tag{84}$$

The Algorithm 1 implies that for any $i > K_1$, we have that

$$f(x_t + \eta_{max}^{(i)} d_t) > f(x_t) + \alpha \eta_{max}^{(i)} \nabla f(x_t)^\top d_t, \qquad \nabla f(x_t + \eta_{min}^{(i)} d_t)^\top d_t < \beta \nabla f(x_t)^\top d_t.$$

Hence, we know that for any $i > K_1$, $\eta_{max}^{(i)} \geq \eta_r$ and $\eta_{min}^{(i)} \leq \eta_l$ where $\eta_r, \eta_l$ are defined in (75), (77) from Lemma K.1. Therefore, using result (78) from Lemma K.1, we have that for any $j \geq 1$,

$$\log_2 \frac{\eta_{max}^{(K_1+j)}}{\eta_{min}^{(K_1+j)}} \geq \log_2 \frac{\eta_r}{\eta_l} > 0. \tag{85}$$

Notice that (84) implies that

$$\log_2 \frac{\eta_{max}^{(K_1+j)}}{\eta_{min}^{(K_1+j)}} = \frac{1}{2^{j-1}} \log_2 \frac{\eta_{max}^{(K_1+1)}}{\eta_{min}^{(K_1+1)}}, \tag{86}$$

which leads to $0 = \lim_{j \to +\infty} \frac{1}{2^{j-1}} \log_2 \frac{\eta_{max}^{(K_1+1)}}{\eta_{min}^{(K_1+1)}} = \lim_{j \to +\infty} \log_2 \frac{\eta_{max}^{(K_1+j)}}{\eta_{min}^{(K_1+j)}} \geq \log_2 \frac{\eta_r}{\eta_l} > 0$. This is a contradiction. Hence, Algorithm 1 must terminate after finite number of loops. Now suppose that Algorithm 1 terminates after $K_1 + \Gamma_1$ iterations, (85) and (86) indicate that when $\Gamma_1 \geq 1$, we have

$$\frac{1}{2^{\Gamma_1-1}} \log_2 \frac{\eta_{max}^{(K_1+1)}}{\eta_{min}^{(K_1+1)}} = \log_2 \frac{\eta_{max}^{(K_1+\Gamma_1)}}{\eta_{min}^{(K_1+\Gamma_1)}} \geq \log_2 \frac{\eta_r}{\eta_l} > \log_2 \left( 1 + \frac{\beta - \alpha}{(1-\beta)(1+2C_t)} \right) \tag{87}$$

where the last inequality holds since (78) in Lemma K.1. Notice that $\eta_{max}^{(K_1+1)} = 2^{2^{K_1}-1}$ and $\eta_{min}^{K_1+1} = 2^{2^{K_1-1}-1}$. Hence, we obtain that

$$\log_2 \frac{\eta_{max}^{(K_1+1)}}{\eta_{min}^{(K_1+1)}} = 2^{K_1-1} \leq 1 + \log_2 \left( 2(1-\alpha)(1+C_t)\rho_t \right). \tag{88}$$

Combing (87), (88) and using $\log x \geq 1 - \frac{1}{x}$, we have that

$$\begin{aligned}
\Gamma_1 &\leq 1 + \log_2 \left( 1 + \log_2 \left( 2(1-\alpha)(1+C_t)\rho_t \right) \right) - \log_2 \log_2 \left( 1 + \frac{\beta - \alpha}{(1-\beta)(1+2C_t)} \right) \\
&\leq 1 + \log_2 \left( 1 + \log_2 \left( 2(1-\alpha)(1+C_t)\rho_t \right) \right) - \log_2 \log \left( 1 + \frac{\beta - \alpha}{(1-\beta)(1+2C_t)} \right) \\
&\leq 1 + \log_2 \left( 1 + \log_2 \left( 2(1-\alpha)(1+C_t)\rho_t \right) \right) - \log_2 \left( 1 - \frac{1}{1 + \frac{\beta-\alpha}{(1-\beta)(1+2C_t)}} \right) \\
&= 1 + \log_2 \left( 1 + \log_2 \left( 2(1-\alpha)(1+C_t)\rho_t \right) \right) + \log_2 \left( 1 + \frac{(1-\beta)(1+2C_t)}{\beta - \alpha} \right).
\end{aligned} \tag{89}$$

Leveraging (83) and (89), we prove that

$$\begin{aligned}
\lambda_t &= K_1 + \Gamma_1 \\
&\leq 2 + 2 \log_2 \left( 1 + \log_2 \left( 2(1-\alpha)(1+C_t)\rho_t \right) \right) + \log_2 \left( 1 + \frac{(1-\beta)(1+2C_t)}{\beta - \alpha} \right).
\end{aligned} \tag{90}$$

Similarly, suppose that at the first iteration, $\eta^{(0)} = 1$ satisfies the sufficient decrease condition (5) but doesn't satisfy the curvature condition (6), we have that $\eta_{max}^{(1)} = 1$, $\eta_{min}^{(1)} = 0$ and $\eta^{(1)} = \frac{1}{2}$. Assume that in the Algorithm 1, $\eta_{min}^{(i)}$ is never set to a positive value and the algorithm never returns. This means that the condition in line 2 is always satisfied, and as a result, we keep repeating steps in line 5. Thus, $\eta^{(i)} = (\frac{1}{2})^{2^i-1}$ and since the condition in line 2 is always satisfied, we have that $f(x_t + \eta^{(i)} d_t) > f(x_t) + \alpha \eta^{(i)} \nabla f(x_t)^\top d_t$. Therefore, we know that $\eta^{(i)} \geq \eta_r$ where $\eta_r > 0$ is defined in (75) from Lemma K.1. Notice that $\eta^{(i)} \geq \eta_r > 0$ for any $i$ and $\lim_{i \to \infty} \eta^{(i)} = 0$, this leads to a contradiction.

Hence, at some point either the algorithm returns a step size satisfying the weak Wolfe conditions or $\eta_{min}^{(i)}$ must become positive. Suppose that this happens at iteration $K_2 \geq 1$ of the loop in Algorithm 1. Then, we know that $\eta^{(K_2)} = (\frac{1}{2})^{2^{K_2}-1}$.

In the first case that the algorithm finds an admissible step size and returns $\eta^{K_2}$, $\eta^{K_2}$ satisfies the Armijo-Wolfe conditions and therefore $\eta^{K_2} \le \eta_r$. Using the upper bound result in (76) from Lemma K.1, we obtain that $\eta^{(K_2)} = 2^{2^{K_2}-1} \le \eta_r \le 2(1-\alpha)(1+C_t)\rho_t$, which leads to

$$\lambda_t = K_2 \le \log_2\Big(1 + \log_2\big(2(1-\alpha)(1+C_t)\rho_t\big)\Big). \tag{91}$$

In the second case that $\eta_{min}^{(i)}$ becomes positive but the algorithm does not terminate, we have that $\eta^{(K_2-1)}$ doesn't satisfy the sufficient condition (5) and $\eta^{(K_2-1)} \ge \eta_r$. Using the lower bound result in (76) from Lemma K.1, we obtain that $\eta^{(K_2-1)} = (\frac{1}{2})^{2^{K_2-1}-1} \ge \eta_r \ge \frac{2(1-\alpha)}{1+C_t}\rho_t$, which leads to

$$K_2 \le 1 + \log_2\Big(1 + \log_2 \frac{1+C_t}{2(1-\alpha)\rho_t}\Big). \tag{92}$$

Then, we further go through the log bisection process. Using the same techniques, we can assume that Algorithm 1 terminates after $K_2 + \Gamma_2$ iterations, where $\Gamma_2 \ge 1$ satisfies that

$$\frac{1}{2^{\Gamma_2-1}}\log_2 \frac{\eta_{max}^{(K_2+1)}}{\eta_{min}^{(K_2+1)}} = \log_2 \frac{\eta_{max}^{(K_2+\Gamma_2)}}{\eta_{min}^{(K_2+\Gamma_2)}} \ge \log_2 \frac{\eta_r}{\eta_l} > \log_2\Big(1 + \frac{\beta-\alpha}{(1-\beta)(1+2C_t)}\Big) \tag{93}$$

where the last inequality holds since (78) in Lemma K.1. Notice that $\eta_{max}^{(K_2+1)} = (\frac{1}{2})^{2^{K_2-1}-1}$ and $\eta_{min}^{K_2+1} = (\frac{1}{2})^{2^{K_2-1}}$. Hence, we obtain that

$$\log_2 \frac{\eta_{max}^{(K_2+1)}}{\eta_{min}^{(K_2+1)}} = 2^{K_2-1} \le 1 + \log_2 \frac{1+C_t}{2(1-\alpha)\rho_t}. \tag{94}$$

Combing (93), (94) and using $\log x \ge 1 - \frac{1}{x}$, we have that

$$\begin{aligned}
\Gamma_2 &\le 1 + \log_2\Big(1 + \log_2 \frac{1+C_t}{2(1-\alpha)\rho_t}\Big) - \log_2\log_2\Big(1 + \frac{\beta-\alpha}{(1-\beta)(1+2C_t)}\Big) \\
&\le 1 + \log_2\Big(1 + \log_2 \frac{1+C_t}{2(1-\alpha)\rho_t}\Big) - \log_2\log\Big(1 + \frac{\beta-\alpha}{(1-\beta)(1+2C_t)}\Big) \\
&\le 1 + \log_2\Big(1 + \log_2 \frac{1+C_t}{2(1-\alpha)\rho_t}\Big) - \log_2\Big(1 - \frac{1}{1 + \frac{\beta-\alpha}{(1-\beta)(1+2C_t)}}\Big) \\
&= 1 + \log_2\Big(1 + \log_2 \frac{1+C_t}{2(1-\alpha)\rho_t}\Big) + \log_2\Big(1 + \frac{(1-\beta)(1+2C_t)}{\beta-\alpha}\Big).
\end{aligned} \tag{95}$$

Leveraging (92) and (95), we prove that

$$\begin{aligned}
\lambda_t &= K_2 + \Gamma_2 \\
&\le 2 + 2\log_2\Big(1 + \log_2 \frac{1+C_t}{2(1-\alpha)\rho_t}\Big) + \log_2\Big(1 + \frac{(1-\beta)(1+2C_t)}{\beta-\alpha}\Big).
\end{aligned} \tag{96}$$

Notice that $\alpha < \frac{1}{2}$ and thus $\frac{1}{2(1-\alpha)} < 2(1-\alpha)$, combining (82), (90), (91) and (96), we prove the final conclusion

$$\begin{aligned}
\lambda_t &\le 2 + \log_2\Big(1 + \frac{(1-\beta)(1+2C_t)}{\beta-\alpha}\Big) \\
&\quad + 2\log_2\Big(1 + \log_2\big(2(1-\alpha)(1+C_t)\big) + \max\{\log_2\rho_t, \log_2 \frac{1}{\rho_t}\}\Big).
\end{aligned}$$

$\square$

### K.3 Proof of Theorem 7.1

Using result from Proposition K.2, we have that

$$\begin{aligned}
\Lambda_t &= \frac{1}{t}\sum_{i=0}^{t-1}\lambda_i \le 2 + \frac{1}{t}\sum_{i=0}^{t-1}\log_2\Big(1 + \frac{(1-\beta)(1+2C_i)}{\beta-\alpha}\Big) \\
&\quad + \frac{2}{t}\sum_{i=0}^{t-1}\log_2\Big(1 + \log_2\big(2(1-\alpha)(1+C_i)\big) + \max\{\log_2\rho_i, \log_2 \frac{1}{\rho_i}\}\Big).
\end{aligned} \tag{97}$$

Using Jensen's inequality, we have that

$$\frac{1}{t}\sum_{i=0}^{t-1}\log_2\left(1+\frac{(1-\beta)(1+2C_i)}{\beta-\alpha}\right)\leq\log_2\left(1+\frac{1-\beta}{\beta-\alpha}+\frac{2(1-\beta)}{\beta-\alpha}\frac{\sum_{i=0}^{t-1}C_i}{t}\right). \tag{98}$$

$$\frac{1}{t}\sum_{i=0}^{t-1}\log_2\left(1+\log_2\left(2(1-\alpha)(1+C_i)\right)+\max\{\log_2\rho_i,\log_2\frac{1}{\rho_i}\}\right)$$

$$\leq\log_2\left(1+\log_2 2(1-\alpha)+\frac{1}{t}\sum_{i=0}^{t-1}\log_2(1+C_i)+\frac{1}{t}\sum_{i=0}^{t-1}\max\{\log_2\rho_i,\log_2\frac{1}{\rho_i}\}\right) \tag{99}$$

$$\leq\log_2\left(1+\log_2 2(1-\alpha)+\log_2\left(1+\frac{\sum_{i=0}^{t-1}C_i}{t}\right)+\frac{1}{t}\sum_{i=0}^{t-1}\max\{\log_2\rho_i,\log_2\frac{1}{\rho_i}\}\right).$$

We also have that

$$\frac{1}{t}\sum_{i=0}^{t-1}\max\{\log_2\rho_i,\log_2\frac{1}{\rho_i}\}=\frac{1}{t}\sum_{i=0,\rho_i\geq 1}^{t-1}\log_2\rho_i+\frac{1}{t}\sum_{i=0,0\leq\rho_i<1}^{t-1}\log_2\frac{1}{\rho_i}$$

$$=\frac{1}{t}\sum_{i=0,\rho_i\geq 2}^{t-1}\log_2\rho_i+\frac{1}{t}\sum_{i=0,1\leq\rho_i<2}^{t-1}\log_2\rho_i+\frac{1}{t}\sum_{i=0,\frac{1}{2}<\rho_i<1}^{t-1}\log_2\frac{1}{\rho_i}+\frac{1}{t}\sum_{i=0,\rho_i\leq\frac{1}{2}}^{t-1}\log_2\frac{1}{\rho_i} \tag{100}$$

$$\leq 2+\frac{1}{t}\sum_{i=0,\rho_i\geq 2}^{t-1}\log_2\rho_i+\frac{1}{t}\sum_{i=0,\rho_i\leq\frac{1}{2}}^{t-1}\log_2\frac{1}{\rho_i},$$

where the inequality is due to $\log_2\rho_i\leq 1$ for $\rho_i<2$ and $\log_2\frac{1}{\rho_i}\leq 1$ for $\rho_i>\frac{1}{2}$. Using the definition of $\omega$ and (b) in Lemma G.1, we obtain that

$$\frac{1}{t}\sum_{i=0,\rho_i\geq 2}^{t-1}\log_2\rho_i=\frac{\log_2 e}{t}\sum_{i=0,\rho_i\geq 2}^{t-1}\log\rho_i=\frac{\log_2 e}{t}\sum_{i=0,\rho_i\geq 2}^{t-1}(\rho_i-1-\omega(\rho_i-1))$$

$$\leq\frac{\log_2 e}{t}\sum_{i=0,\rho_i\geq 2}^{t-1}(\frac{2\rho_i}{\rho_i-1}\omega(\rho_i-1)-\omega(\rho_i-1)) \tag{101}$$

$$=\frac{\log_2 e}{t}\sum_{i=0,\rho_i\geq 2}^{t-1}\frac{\rho_i+1}{\rho_i-1}\omega(\rho_i-1)\leq\frac{3\log_2 e}{t}\sum_{i=0,\rho_i\geq 2}^{t-1}\omega(\rho_i-1).$$

Similarly, using (c) in Lemma G.1, we obtain that

$$\frac{1}{t}\sum_{i=0,\rho_i\leq\frac{1}{2}}^{t-1}\log_2\frac{1}{\rho_i}=\frac{\log_2 e}{t}\sum_{i=0,\rho_i\leq\frac{1}{2}}^{t-1}\log\frac{1}{\rho_i}=\frac{\log_2 e}{t}\sum_{i=0,\rho_i\leq\frac{1}{2}}^{t-1}(\omega(\rho_i-1)+1-\rho_i)$$

$$\leq\frac{\log_2 e}{t}\sum_{i=0,\rho_i\leq\frac{1}{2}}^{t-1}(\omega(\rho_i-1)+\frac{1+\rho_i}{1-\rho_i}\omega(\rho_i-1)) \tag{102}$$

$$=\frac{\log_2 e}{t}\sum_{i=0,\rho_i\leq\frac{1}{2}}^{t-1}\frac{2}{1-\rho_i}\omega(\rho_i-1)\leq\frac{4\log_2 e}{t}\sum_{i=0,\rho_i\leq\frac{1}{2}}^{t-1}\omega(\rho_i-1).$$

Combining (100), (101) and (102), we prove that

$$\frac{1}{t}\sum_{i=0}^{t-1}\max\{\log_2\rho_i,\log_2\frac{1}{\rho_i}\}\leq 2+\frac{1}{t}\sum_{i=0,\rho_i\geq 2}^{t-1}\log_2\rho_i+\frac{1}{t}\sum_{i=0,\rho_i\leq\frac{1}{2}}^{t-1}\log_2\frac{1}{\rho_i}$$

$$\leq 2+\frac{4\log_2 e}{t}\sum_{i=0}^{t-1}\omega(\rho_i-1)\leq 2+\frac{6}{t}\left(\Psi(\tilde{B}_0)+2\sum_{i=0}^{t-1}C_i\right). \tag{103}$$

where we use the fact that $\omega(\rho_i - 1) \geq 0$ for any $i \geq 0$ and the last inequality is due to (46) in Proposition G.2. Leveraging (97), (98), (99) and (103), we have that

$$\Lambda_t \leq 2 + \log_2 \left(1 + \frac{1-\beta}{\beta - \alpha} + \frac{2(1-\beta)}{\beta - \alpha} \frac{\sum_{i=0}^{t-1} C_i}{t}\right)$$

$$+ 2\log_2 \left(3 + \log_2 2(1-\alpha) + \log_2 \left(1 + \frac{\sum_{i=0}^{t-1} C_i}{t}\right) + \frac{6}{t}\left(\Psi(\tilde{B}_0) + 2\sum_{i=0}^{t-1} C_i\right)\right)$$

$$\leq 2 + \log_2 \left(1 + \frac{1-\beta}{\beta - \alpha} + \frac{2(1-\beta)}{\beta - \alpha} \frac{\sum_{i=0}^{t-1} C_i}{t}\right)$$

$$+ 2\log_2 \left(\log_2 16(1-\alpha) + \log_2 \left(1 + \frac{\sum_{i=0}^{t-1} C_i}{t}\right) + \frac{6\Psi(\tilde{B}_0) + 12\sum_{i=0}^{t-1} C_i}{t}\right).$$

We prove the final conclusion using (44) from the proof of Theorem 5.2 in Appendix F.2, i.e.,

$$\sum_{i=0}^{t-1} C_i \leq C_0 \Psi(\bar{B}_0) + \frac{3C_0\kappa}{\alpha(1-\beta)}.$$

## K.4 Corollaries of Theorem 7.1 for $B_0 = LI$ and $B_0 = \mu I$

**Corollary K.3** ($B_0 = LI$). *Suppose that Assumptions 2.1, 2.2 and 2.3 hold. Let $\{x_t\}_{t\geq 0}$ be the iterates generated by the BFGS method, where the step size satisfies the Armijo-Wolfe conditions in (5) and (6). For any initial point $x_0 \in \mathbb{R}^d$ and the initial Hessian approximation matrix $B_0 = LI$, the average complexity of line search Algorithm 1 $T_k$ is upper bounded by*

$$\Lambda_t \leq 2 + \log_2 \left(1 + \frac{1-\beta}{\beta - \alpha} + \frac{2(1-\beta)}{\beta - \alpha} \frac{3C_0\kappa}{\alpha(1-\beta)t}\right)$$

$$+ 2\log_2 \left(\log_2 16(1-\alpha) + \log_2 \left(1 + \frac{3C_0\kappa}{\alpha(1-\beta)t}\right) + \frac{6d\kappa + \frac{36C_0\kappa}{\alpha(1-\beta)}}{t}\right).$$

*Moreover, when $t \geq 6d\kappa + \frac{36}{\alpha(1-\beta)}C_0\kappa$, we have that*

$$\Lambda_t \leq 2 + \log_2 \left(1 + \frac{3(1-\beta)}{\beta - \alpha}\right) + 2\log_2(5 + \log_2 2(1-\alpha)). \tag{104}$$

*Proof.* Since $B_0 = LI$, we have $\bar{B}_0 = \frac{1}{L}B_0 = I$ and $\tilde{B}_0 = \nabla^2 f(x_*)^{-\frac{1}{2}} B_0 \nabla^2 f(x_*)^{-\frac{1}{2}} = L\nabla^2 f(x_*)^{-1}$. Using results in the proof of Corollary 5.3, we have

$$\Psi(\bar{B}_0) = 0, \qquad \Psi(\tilde{B}_0) \leq d\kappa.$$

Combining these two results with the result in Theorem 7.1, we prove the conclusion. $\square$

**Corollary K.4** ($B_0 = \mu I$). *Let $\{x_t\}_{t\geq 0}$ be the iterates generated by the BFGS method with inexact line search (5), (6) and suppose that Assumptions 2.1, 2.2 and 2.3 hold. For any initial point $x_0 \in \mathbb{R}^d$ and the initial Hessian approximation matrix $B_0 = \mu I$, the average complexity of line search Algorithm 1 $T_k$ is upper bounded by*

$$\Lambda_t \leq 2 + \log_2 \left(1 + \frac{1-\beta}{\beta - \alpha} + \frac{2(1-\beta)}{\beta - \alpha} \frac{C_0 d\log\kappa + \frac{3C_0\kappa}{\alpha(1-\beta)}}{t}\right)$$

$$+ 2\log_2 \left(\log_2 16(1-\alpha) + \log_2 \left(1 + \frac{C_0 d\log\kappa + \frac{3C_0\kappa}{\alpha(1-\beta)}}{t}\right) + \frac{6(1+2C_0)d\log\kappa + \frac{36C_0\kappa}{\alpha(1-\beta)}}{t}\right).$$

*Moreover, when $t \geq 6(1+2C_0)d\log\kappa + \frac{36C_0\kappa}{\alpha(1-\beta)}$, we have that*

$$\Lambda_t \leq 2 + \log_2 \left(1 + \frac{3(1-\beta)}{\beta - \alpha}\right) + 2\log_2(5 + \log_2 2(1-\alpha)). \tag{105}$$

*Proof.* Since $B_0 = \mu I$, we have $\bar{B}_0 = \frac{1}{\kappa} B_0 = I$ and $\tilde{B}_0 = \nabla^2 f(x_*)^{-\frac{1}{2}} B_0 \nabla^2 f(x_*)^{-\frac{1}{2}} = \mu \nabla^2 f(x_*)^{-1}$. Using results in the proof of Corollary 4.2, we have

$$\Psi(\bar{B}_0) \leq d \log \kappa, \qquad \Psi(\tilde{B}_0) \leq d \log \kappa.$$

Combining these two results with (26) in Theorem 6.4, we prove the conclusion. $\qquad \square$

