# OpenReview forum: "Non-asymptotic Global Convergence Analysis of BFGS with the Armijo-Wolfe Line Search"
_NeurIPS.cc/2024/Conference — NeurIPS 2024 spotlight_

### Official Review · Reviewer_EiJP · 2024-07-08

**Soundness:** 3
**Presentation:** 2
**Contribution:** 3
**Rating:** 6
**Confidence:** 4

**Summary:**

This paper establishes a unified non-asymptotic convergence analysis of the BFGS method with Armijo-Wolfe line search. It shows that BFGS exhibits two converging stages: (1) a linear global convergence rate, and the rate is  independent of the condition number when Hessian is Lipschitz continuous. (2) a superlinear convergence as the unit step length satisfies the Armijo-Wolfe line search condition. The convergence rate is dependent of initialization of weight matrix, suboptimality of initial iterate, condition number, and parameters in line search condition. The iteration complexity is also explored using a log bisection algorithm.

**Strengths:**

- The non-asymptotic global convergence rate presented in this paper improves the previous result in the literature. The theoretical framework is rigorous, with clearly stated assumptions and detailed proofs supporting the main results. The theorem statements and comments are clear.
- The use of log bisection algorithm is innovative.
- The explanation of theoretical results aid in the reader's understanding of the methodology and results.
- By providing a unified non-asymptotic convergence analysis of global and superlinear convergence result together with the complexity analysis, the paper  in optimization literature.

**Weaknesses:**

- The statement “present the first explicit” in abstract is too strong.
- The author could add “with $B_0 = L I$” or comment that this result needs such condition in Line 4 to be more precise.
- A general comment for abstract: the abstract starts with a global convergence result and then establishes the global superlinear convergence rate. However, this could be confusing to the audiences unfamiliar with the literature. Why would you present linear convergence if you have proved a superlinear convergence? Hence, I would suggest to improve the presentation to make the abstract more clear.
- When [38] is mentioned, the authors could comments about why the convergence rate is worse even with an exact line search. Although a theoretical reason is discussed in remark 4.1, I feel that a more intuitive explanation could be provided in the introduction section.
- In line 62 of the contribution, the author should mention that the convergence rate $(1-\frac{\mu}{L})^t$ is proved under the condition $B_0 = L I$.
- It is stated “we focus on two practical initialization schemes:” in line 217. However, the initialization requires $L$ or $\mu$, which is normally unknown. Hence, they are not practical schemes.
- $\tilde B_0$ is redefined in line 227.
- The $\hat \rho_t$ should converge to 1/2 instead of 1 in line 237.
- make { } in $\delta_2$ larger in eq. 23.
- It would be more interesting to add one more bullet point to Corollary 4.2, for $t \ge T(d, \kappa, B_0)$, you could achieve similar explicit linear rate for any initial weighted matrix $B_0$.
- The authors could comment more about the finite number of iterations. (results in lemma 6.3) For example, how large it is in your specialized two initializations? How will the interation indices in set I be distributed?
- When $t$ is not sufficiently large, the upper bound provided in (26) could be very large compared to the upper bound given in Section 4 and 5.  In terms of the presentation, it might be more transparent to present all the transitions you have in terms of the convergence, from inexplicit linear convergence, to explicit linear convergence, to superlinear convergence.
- $C_0$ Should be suboptimality of initial iterate in line 273.
- The author comments in line 319 that “notice that when N is sufficiently large such that we reach the superlinear convergence stage”. However, there are not really different stages presented in Section 6.
- Log bisection algorithm is not usually used in practice for BFGS. The author should comment about the reason that one should use log bisection rather than other popular line search schemes like backtracking.
- While this is a theoretical paper, authors could still show some numerical experiment results to present different convergence stages of BFGS, performance with different initialized matrix $B_0$, change of step size etc.

**Questions:**

Please see the previous section.

**Limitations:**

The discussed limitations of the work are clear. I see no potential negative societal impact related to this work. I suggest to include empirical results or numerical experiments to validate the theoretical findings.

---

> ### Author Rebuttal · Authors · 2024-08-07
>
> **Response to weakness 1,2,3.** We will soften our claims and add “$B_0 = L I$” in the abstract. Our global superlinear convergence analysis is based on the results from the global linear convergence rates, so we need to present both the linear and superlinear convergence results. Moreover, the linear convergence rates we obtained in this submission are also novel and innovative since there is no such linear ratio of $1 - \frac{1}{\kappa}$ or parameter-free linear rate as presented in Section 5 in previous works. We will revise the abstract accordingly to resolve these ambiguities.
>
> **Response to weakness 4.** The key intuitive explanation is the step size $\eta_t$ satisfying the exact line search may not satisfy the sufficient descent condition (5) for any $\alpha \in (0, \frac{1}{2})$, which is a fundamental condition to prove the convergence bound in our paper. We will add this explanation in our revised version.
>
> **Response to weakness 5.** We will clarify that when $B_0 = L I$, the linear rate of $(1 - \frac{\mu}{L})^t$ is achieved at the first iteration while for $B_0 = \mu I$, the same bound holds for $t \geq d\log{\kappa}$.
>
> **Response to weakness 6.**  We agree with the reviewer that $LI$ and $\mu I$ are not necessarily practical initialization schemes, as we generally may not know the values of $L$ or $\mu$. We will modify the wording in our revised version. It is better to state that we choose these two specific initialization matrices since they can make all the non-asymptotic global linear and superlinear convergence bounds and expressions simpler and clearer. That said, in the attached numerical experiments, we have illustrated the convergence rate of two more practical choices: $B_0 = I$ and $B_0 = cI$ where $c \in [\mu, L]$ and is easy to compute. In the revised paper, we plan to add corollaries for these two special cases after each result presented in the paper.
>
> **Response to weakness 7, 9, 13.**  We will fix all these typos.
>
> **Response to weakness 8.**  Let's clarify why $\hat{\rho_t}$ should converge to 1. This is because The global linear convergence results indicate that $\sum_{i = 0}^{\infty}C_i < +\infty$. Hence, from (44) of Proposition G.2 in the appendix, we know that $\sum_{i = 0}^{\infty}\omega(\rho_i - 1) < +\infty$, which implies that $\lim_{i \to \infty}\omega(\rho_i - 1) = 0$. Notice that the function $\omega$ is continuous and $\omega(x) = 0$ if and only if $x = 0$. Therefore, we have that $\lim_{i \to \infty}\rho_i = 1$.
>
> In equation (22), the lower bound of $\hat{p}_t$ should converge to $\frac{1}{2}$ since $C_t$ converges to 0 and $\rho_t$ converges to 1. We would like to highlight that the fact that $\hat{p}_t$ converges to $\frac{1}{2}$ is not problematic, as the lower bound of $\hat{q}_t$ has a factor of 2 (see line 519, part (b) of Lemma D.2 in the appendix). This factor of 2 of $\hat{q}_t$ times the limit $\frac{1}{2}$ of $\hat{p}_t$ produces the unit factor 1, which leads to the superlinear convergence rates of the BFGS method in section 6.
>
> **Response to weakness 10.**  We will add more lines in Corollary 4.2 for the convergence results of any initial $B_0$ in our revised version.
>
> **Response to weakness 11.** We will add comments to our revised paper. In fact when $B_0 = L I$, we have that $t_0 = \frac{3\kappa}{\alpha(1 - \beta)}\log{\frac{C_0}{\delta_1}}$ and $|I|  \leq \delta_4\Big(d\kappa + \frac{6C_0\kappa}{\alpha(1 - \beta)}\Big)$. When $B_0 = \mu I$, we have that $t_0 = \max\\{d\log{\kappa}, \frac{3\kappa}{\alpha(1 - \beta)}\log{\frac{C_0}{\delta_1}}\\}$ and $|I|  \leq \delta_4\Big(\Psi((1 + 2C_0)d\log{\kappa} + \frac{6C_0\kappa}{\alpha(1 - \beta)}\Big)$.
>
> **Response to weakness 12.**  We have added one table in the attached pdf presenting all the convergence stages with a required number of iterations to transit to each phase.
>
> **Response to weakness 14.**  Thanks for the comments. We did not explicitly present different convergence stages in our submission. To address this issue, in the attached PDF we included a table highlighting all the linear and superlinear convergence stages with the required number of steps to transition to each phase. Please check the table in the uploaded PDF file. We will add this transition table in our revised version.
>
> **Response to weakness 15.**  In general, backtracking line search is used for finding an acceptable step size when we know that the interval of admissible step sizes is bounded below by zero and can be written as $(0, \eta_1]$. In this case, we can start the process of selecting the step size with a large step size $\eta_{initial}$, possibly $\eta = 1$, and then backtrack it by a factor $\beta < 1$. It can be easily verified that after at most $\log_{1/\beta} (\eta_{initial}/\eta_1)$ iterations, we will find an admissible step size. In fact, if we only needed the sufficient decrease condition (5), we could apply the backtracking technique with initial $\eta_1 = 1$.
>
> However, the weak Wolfe conditions require both (5) and (6). Condition (6) indicates that there exists a strictly positive $\eta_2$ such that the admissible step size $\eta_t \geq \eta_2$. Hence, the lower bound of the proper step size satisfying both (5) and (6) is not zero, and we can no longer utilize backtracking to find such a step size. Therefore, we need to find a window $[\eta'_2, \eta'_1]$ such that $\eta'_2 \geq \eta_2$ and $\eta'_1 \leq \eta_1$ so that when $\eta_t$ belongs to this window $[\eta'_2, \eta'_1]$, we have that $\eta_t \in [\eta_2, \eta_1]$ and this $\eta_t$ satisfies conditions (5) and (6). We need to use the bisection method to construct both the upper and lower bounds of this window. We apply the log bisection instead of the vanilla bisection because it helps simplify the theoretical analysis of the line search complexity. We will highlight this in our revised version of the paper.
>
> **Response to weakness 16 and Limitations.** Please check numerical experiments in the attached file.

---

### Official Review · Reviewer_GXyX · 2024-07-12

**Soundness:** 4
**Presentation:** 3
**Contribution:** 4
**Rating:** 8
**Confidence:** 4

**Summary:**

This paper provides the non-asymptotic global linear convergence rate of $O((1-1/\kappa)^t)$ for BFGS method with inexact line search. It also shows the superlinear convergence rate of $O((1/t)^t)$ under the Hessian Lipschitz condition.

**Strengths:**

See questions.

**Weaknesses:**

N/A

**Questions:**

The BFGS method is a classical quasi-Newton method proposed in 1970 and its explicit local superlinear convergence rate has been established until 2021. However, the global convergence of BFGS is an interesting problem which has not been well addressed before. For example, applying traditional analysis of Armijo-Wolfe line search on BFGS to guarantee the global convergence cannot achieve the explicit local superlinear rates.

This paper provides both global and local convergence rates of BFGS methods with inexact line search, matching the rates of gradient descent and BFGS under the first-order and second-order Lipschitz assumptions. Both of these results are reasonable, and the theoretical analysis based on the function $\Phi(A)={\rm Tr}(A)-d-\log\det(A)$ make sense. I believe the theoretical contribution of this paper is valuable to the community of machine learning and optimization.

I have some minor questions/comments:
1. Unifying the global and local convergence rates into one framework of BFGS with line search is nice. We can also address this problem by a simple way, i.e., run (accelerated) gradient descent to enter the local region, then run standard BFGS to achieve the superlinear rate. Can you compare the results in your paper with this heuristic method?
2. Can you provide some result for $B_0=\alpha I$ with $\alpha\in(\mu,L)$ as the extension of Corollary 6.5?
3. The discussion on the initial Hessian estimator after Corollary 6.5 is interesting. It is better to provide some experiments to validate it. Considering the theoretical contribution of this paper is strong, I will still recommend accept even if there is no additional experiment in rebuttal.
4. The following recent work for the explicit superlinear convergence rates of quasi-Newton methods should be involved into the literature review:

         [A] Zhuanghua Liu, Luo Luo, Bryan Kian Hsiang Low. Incremental quasi-Newton methods with faster superlinear convergence rates. AAAI 2024.
         [B] Aakash Lahoti, Spandan Senapati, Ketan Rajawat, Alec Koppel. Sharpened lazy incremental quasi-Newton method. arXiv:2305.17283.
         [C] Chengchang Liu, Cheng Chen, Luo Luo. Symmetric rank-k methods. arXiv:2303.16188, 2023.
         [D] Chengchang Liu, Cheng Chen, Luo Luo, John C.S. Lui. Block Broyden's methods for solving nonlinear equations. NeurIPS 2023.
         [E] Chengchang Liu, Luo Luo. Quasi-Newton methods for saddle point problems. NeurIPS 2022.
         [F] Chengchang Liu, Shuxian Bi, Luo Luo, John C.S. Lui. Partial-quasi-Newton methods: efficient algorithms for minimax optimization problems with unbalanced dimensionality. KDD 2022.
         [G] Haishan Ye, Dachao Lin, Zhihua Zhang. Greedy and random Broyden's methods with explicit superlinear convergence rates in nonlinear equations. arXiv:2110.08572
         [H] Dachao Lin, Haishan Ye, Zhihua Zhang. Explicit superlinear convergence rates of Broyden's methods in nonlinear equations. arXiv:2109.01974

5. In line 49, it is somewhat inappropriate to call reference [40] a draft. It is more appropriate to use "unpublished paper" or "technical report" to describe it.

---

> ### Author Rebuttal · Authors · 2024-08-07
>
> **Question 1.** *Unifying the global and local convergence rates into one framework of BFGS with line search is nice. We can also address this problem by a simple way, i.e., run (accelerated) gradient descent to enter the local region, then run standard BFGS to achieve the superlinear rate. Can you compare the results in your paper with this heuristic method?*
>
> **Response.** This is a great question. We can use the suggested method of the combination of accelerated gradient descent (AGD) with BFGS to achieve a faster convergence rate compared to the vanilla BFGS with line search. However, we would like to highlight the following two points:
>
> First, we would like to mention that the global convergence complexity of the vanilla BFGS quasi-Newton method with line search was an interesting open problem worth exploring. Interestingly, our results show that BFGS with inexact line search is no worse than gradient descent, and will reach the faster superlinear convergence stage after some explicit iterations. Indeed, it would be great if one could use this analysis to design an accelerated version of BFGS that globally outperforms AGD, meaning it achieves a global linear rate that matches the one for AGD and an eventual superlinear rate.
>
> Second, the size of the local neighborhood for BFGS to achieve superlinear convergence depends on some parameters of the objective function, such as the gradient smoothness parameter $L$ or the strong convexity parameter $\mu$. Hence, to follow the reviewer's suggestion and switch from AGD to BFGS in a local neighborhood of the solution, we need to know these parameters to determine if the iteration has reached the local regime. Therefore, this combination of AGD and BFGS requires knowing the values of parameters $L$ or $\mu$. For our vanilla BFGS with line search, we do not require access to any of those parameters in the implementation of the algorithm. Although in Corollary 6.5 we compare the results of different $B_0 = LI$ and $B_0 = \mu I$, we only use these $B_0$ to explicitly obtain the convergence bounds of the BFGS. In practice, $B_0$ could be any symmetric positive definite matrix. We can just choose $B_0 = I$ or $B_0 = cI$ for any positive real number $c$ (please check the presented experiments in the attached PDF). Therefore, to implement the AGD plus BFGS, we need to know some parameters of the objective function. In contrast, to implement the vanilla BFGS with line search, we don't need to know any parameters except the line search parameters $\alpha$ and $\beta$ as specified in (5) and (6). This is one advantage of vanilla global BFGS with line search over the combination of first-order methods with local BFGS.
>
> **Question 2.** *Can you provide some result for $B_0 = \alpha I$ with $\alpha \in (\mu, I)$ as the extension of Corollary 6.5?*
>
> **Response.** All our global linear and superlinear convergence bounds hold for any $B_0$ as long as it is symmetric positive definite. The only difference the choice of initial Hessian approximation could make is in the value of $\Psi(\bar{B_0})$ and $\Psi(\tilde{B_0})$ appearing in our theoretical results. In the corollaries of our paper, we chose $B_0 = LI$ and $B_0 = \mu I$ as two special cases because we can obtain specific and explicit upper bounds in terms of the dimension $d$ and the condition number $\kappa$ in different theorems, making our results easier to parse.
>
> If we choose $B_0 = \alpha I$ in Corollary 6.5, then $\Psi(\bar{B_0}) = \Psi(\frac{\alpha}{L}I) = \frac{\alpha}{L}d - d + d\log{\frac{L}{\alpha}}$. Moreover, we would have $\Psi(\tilde{B_0}) = \Psi(\alpha \nabla^2{f(x_*)^{-1}}) = \alpha\mathbf{Tr}(\nabla^2{f(x_*)^{-1}}) - d - \log{\mathbf{Det}(\alpha\nabla^2{f(x_*)^{-1}})}$ which is totally determined by the Hessian matrix $\nabla^2{f(x_*)^{-1}}$. In this case, one can use the upper bounds $\Psi(\bar{B_0}) = d(\frac{\alpha}{L} - 1 + \log{\frac{L}{\alpha}})$ and $\Psi(\tilde{B_0}) = \mathbf{Tr}(\alpha\nabla^2{f(x_*)}^{-1}) - d - \log{\mathbf{Det}(\alpha\nabla^2{f(x_*)}^{-1})} \leq d(\frac{\alpha}{\mu} - 1 + \log{\frac{L}{\alpha}}) $ to simplify the expressions.
>
> Applying these values of $\Psi(\bar{B_0})$ and $\Psi(\tilde{B_0})$ to the superlinear bound in Theorem 6.4, we can obtain the superlinear convergence rates for $B_0 = \alpha I$ in Corollary 6.5. We also include the performance of BFGS with $B_0 = cI$ where $c \in [\mu, L]$ in our numerical experiments as presented in Figure 1 of the attached pdf. We observe that the performance of BFGS with $B_0 = cI$ is very similar to the convergence curve of BFGS with $B_0 = \mu I$ in our numerical experiments. We will add this dicussion to the revised paper.
>
> **Question 3.** *The discussion on the initial Hessian estimator after Corollary 6.5 is interesting. It is better to provide some experiments to validate it. Considering the theoretical contribution of this paper is strong, I will still recommend accept even if there is no additional experiment in rebuttal.*
>
> **Response.** Thanks for the advice. We have attached the empirical results of our numerical experiments with different $B_0$ in the pdf.
>
> **Question 4.** *The following recent work for the explicit superlinear convergence rates of quasi-Newton methods should be involved into the literature review:*
>
> **Response.** Thanks for pointing out all these papers. Indeed, they are all relevant to our submission since they characterize local non-asymptotic convergence rates of different quasi-Newton methods in various settings. We will cite these papers as previous works in the introduction section of our revised version.
>
> **Question 5.** *In line 49, it is somewhat inappropriate to call reference [40] a draft. It is more appropriate to use "unpublished paper" or "technical report" to describe it.*
>
> **Response.** Thanks for the advice. We will modify our text in the revised version to cite the reference [40] as a "technical report".

---

> > ### Comment · Reviewer_GXyX · 2024-08-08
> >
> > Thanks for your careful rebuttal.  I strongly recommend you incorporate the discussion on AGD+BFGS into revision.
> >
> > Some minor comments on figures in PDF:
> > 1. The initial Hessian estimator in the figure should be displayed in formula font.
> > 2. Different line styles should be used to distinguish the curves in the figures.

---

> > > ### Author Response · Authors · 2024-08-08
> > >
> > > Thank you for reading our rebuttal and providing your follow-up comments.
> > >
> > > We will include the discussion on the AGD+BFGS comparison in our revised paper. Additionally, we will address the comments you made regarding the figures. We appreciate your feedback.

---

> > > > ### Comment · Reviewer_GXyX · 2024-08-09
> > > >
> > > > Thanks for your further response.

---

### Official Review · Reviewer_7mqx · 2024-07-12

**Soundness:** 3
**Presentation:** 3
**Contribution:** 3
**Rating:** 6
**Confidence:** 3

**Summary:**

The paper provides non-asymptotic global convergence for BFGW with Armijo-Wolfe (A-S) line search. It provides three main results:

(a) $O(1- 1/\kappa)^t$ rate globally

(b) with Lipschitz Hessian: $O(1-\alpha(1-\beta))^t$ rate (condition number independent) after iteration t is large enough

(c) with Lipschitz Hessian: $O((1/t)^t)$ superlinear rate

**Strengths:**

1. The convergence is characterized non-asymptotically. I believe such results for BFGS with A-S stepsize are new.

2. The rate provided for BFGS is faster than in previous literature.

**Weaknesses:**

Authors should carefully consider the similarities and differences with the reference [38]. For example, Proposition 3.1 in this manuscript is almost the same as Proposition 1 in [38], which is a critical element of this paper. Additionally, the proof employs some results from [38]. I am concerned about the extent of this overlap. Although authors already commented on the final results difference with [38], there is also much overlap in the main contents. It is essential for the authors to more thoroughly address and discuss these parallels within the manuscript.

**Questions:**

1. I am looking for some comments about the existence of stepsize $\eta_t$ satisfying A-S conditions (5) and (6). Does the exact line-search stepisze in [38] satisfy it?

2. What happens if we simply pick the initial $B_0$ to be the identity matrix$?

---

> ### Author Rebuttal · Authors · 2024-08-07
>
> **Response to Weakness.** Thank you to the reviewer for raising this valid point. We would like to mention that our paper and reference [38] have similar goals, both aiming to establish the global convergence rate of BFGS under some line-search scheme. The primary difference is in the choice of line-search: in our work, the step size is chosen to satisfy the weak Wolfe conditions specified in (5) and (6), while in [38], the step size is determined by exact line search. Given that, it is inevitable to see some similarities and overlaps in the presentation and intermediate results of our submission and [38]. In fact, we have  highlighted in the paper any overlap with [38]. That said, all our theoretical results are different from the ones in [38] and our analysis deviates from [38] in the following multiple levels:
> 1. The global linear convergence rate $1 - \frac{1}{\kappa}$ in Theorem 4.1 of our paper is strictly better than the corresponding global linear rate $1 - \frac{1}{\kappa^{1.5}}$ of the exact line search in [38]. This improvement is due to the fact that we can lower bound the term $\hat{p}_t$ defined in (22) by $\alpha$ as presented in Lemma 2.1 in our submission instead of $\frac{1}{\sqrt{\kappa}}$ in [38]. This difference of the lower bounds of $\hat{p}_t$ leads to the improvement of our results.
> 2. The convergence bounds in Theorem 5.2 are novel and unique to our submission, demonstrating that under the additional assumption that the Hessian is also Lipschitz, BFGS can achieve a linear rate independent of the problem parameters after a certain number of iterations. The analysis in [38] failed to establish similar parameter-free global linear rates, under similar assumptions.
> 3. The most significant distinction between our analysis and the one in [38] is in the global superlinear convergence. Specifically, in our submission, we focus on the "good" event that ensures the unit step size $\eta_t = 1$ is admissible (satisfying the weak Wolfe conditions (5) and (6)) and control the size of the "bad" set of the time indexes where $\eta_t \neq 1$ (see Lemmas 6.1 to 6.3). This technique and idea are independent of the superlinear convergence analysis in [38], where the step size $\eta_t$, determined by the exact line search, can be $1$, larger than $1$, or smaller than $1$.
>
> **Response to question 1.** This is a good point. Note that for any objective function that is bounded below and any descent search direction $d_t$ where $g_t^\top d_t < 0$, there always exists an interval of the form $[\eta_l, \eta_u]$ for which all the real values in this interval satisfy the weak Wolfe conditions specified in (5) and (6). (Please check Lemma 3.1 in Numerical Optimization by Stephen J. Wright and Jorge Nocedal.) Notice that we assume that the objective function is bounded below in our submission and the BFGS search direction $d_t = B_t^{-1} g_t$ is a descent search direction. Hence, there always exists an interval of step size choices for which the conditions (5) and (6) are satisfied. We will highlight this point in the revised paper. While this is only an existence result, in our line search routine, we proposed the log bisection algorithm (Algorithm 1) that chooses the step size $\eta_t$ satisfying the weak Wolfe conditions (5) and (6), and we also characterized the complexity of our log bisection scheme that produces such $\eta_t$. Please check all the proofs in Appendix I. Hence, it is possible to efficiently find such stepsize.
>
> Regarding your question on whether the exact line-search step size satisfies the conditions in (5) and (6), the short answer is no. To be more precise, the step size $\eta_t$ obtained by the exact line-search satisfies the property that $g_{t + 1}^\top d_t = 0$. Hence, it satisfies the curvature condition (6) for any $\beta \in (0, 1)$. However, there is no guarantee that the exact line-search step size could satisfy the sufficient descent condition (5) for any $\alpha \in (0, 1/2)$. Therefore, in general, the exact line-search step size from [38] is not admissible for the weak Wolfe conditions (5) and (6).
>
> **Response to question 2.** This is a good question. All our global linear and superlinear convergence bounds hold for any $B_0$ as long as it is symmetric positive definite. The only difference the choice of initial Hessian approximation could make is in the value of $\Psi(\bar{B_0})$ and $\Psi(\tilde{B_0})$ appearing in our theoretical results. In the corollaries of our paper, we chose $B_0 = LI$ and $B_0 = \mu I$ as two special cases because we can obtain specific and explicit upper bounds in terms of the dimension $d$ and the condition number $\kappa$ in different theorems, making our results easier to parse.
>
> It is possible to establish all results for the special case where $B_0 = I$, and only the values of $\Psi(\bar{B_0})$ in Theorem 4.1 and Theorem 5.2 and $\Psi(\tilde{B_0})$ in Theorem 6.4 will change, where $\bar{B_0} = \frac{1}{L}B_0$ and $\tilde{B_0} = \nabla^2{f(x_*)}^{-\frac{1}{2}}B_0 \nabla^2{f(x_*)}^{-\frac{1}{2}}$. Notice that the transition time required to reach different linear and superlinear convergence stages depends on the values of $\Psi(\bar{B_0})$ and $\Psi(\tilde{B_0})$. Hence, the only impact of choosing $B_0$ as the identity matrix is that it will affect the required number of steps to reach the parameter-free linear convergence stage and the superlinear convergence stage. In fact, if $B_0 = I$, we have that $\Psi(\bar{B_0}) = \frac{1}{L}d - d + d\log{L}$ and $\Psi(\tilde{B_0}) = \mathbf{Tr}(\nabla^2{f(x_*)}^{-1}) - d - \log{\mathbf{Det}(\nabla^2{f(x_*)}^{-1})} \leq d(\frac{1}{\mu} - 1 + \log{L})$. Finally, to better study the special case of $B_0 = I$, we conducted numerical experiments with BFGS using $B_0 = I$, as presented in Figure 1 of the attached PDF. As we observe, the performance of choosing $B_0$ as the identity matrix $I$ is quite good compared to other initializations of $B_0$.

---

> > ### Comment · Reviewer_7mqx · 2024-08-12
> >
> > Thank you for the reply. I don't have further questions.

---

### Author Rebuttal · Authors · 2024-08-07

We thank the reviewers for providing all these valuable advice and constructive feedbacks. Here is the general response.

**Numerical Experiments**

We conducted some numerical experiments and we attached all our empirical results in Figure 1 of the uploaded file. We focus on the hard cubic objective function defined as
\begin{equation*}
    f(x) = \frac{\alpha}{12}\left(\sum_{i = 1}^{d - 1}g(v_i^\top x - v_{i + 1}^\top x) - \beta v_1^\top x\right) + \frac{\lambda}{2}\\|x\\|^2,
\end{equation*}
and $g: \mathbb{R} \to \mathbb{R}$ is defined as
\begin{equation*}
    g(w) =
    \begin{cases}
        \frac{1}{3}|w|^3 & |w| \leq \Delta, \\
        \Delta w^2 - \Delta^2 |w| + \frac{1}{3}\Delta^3  & |w| > \Delta,
    \end{cases}
\end{equation*}
where $\alpha, \beta, \lambda, \Delta \in \mathbb{R}$ are hyper-parameters and $\{v_i\}_{i = 1}^{n}$ are standard orthogonal unit vectors in $\mathbb{R}^{d}$. This hard cubic function is used to establish a lower bound for second-order methods.

We compared the convergence curves of the BFGS method with inexact line search step size $\eta_t$ satisfying weak Wolfe conditions (5) and (6) of different initialization matrices $B_0$: $B_0 = L I$, $B_0 = \mu I$, $B_0 = I$ and $B_0 = c I$ where $c = \frac{s^\top y}{\|s\|^2}$, $s = x_2 - x_1$, $y = \nabla{f(x_2)} - \nabla{f(x_1)}$ and $x_1, x_2$ are two randomly generated vectors. It can be easily verified that $c \in [\mu, L]$. We choose $\alpha = 0.1$ in condition (5) and $\beta = 0.9$ in condition (6). We apply the log bisection in Algorithm 1 to choose the step size $\eta_t$ at each iteration. We compare the empirical results with various dimensions $d$ and condition numbers $\kappa$. The x-axis is the number of iterations $t$ and the y-axis is the ratio $\frac{f(x_t) - f(x_*)}{f(x_0) - f(x_*)}$.

First, we observe that BFGS with $B_0 = L I$ initially converges faster than BFGS with $B_0 = \mu I$ in most plots, aligning with our theoretical findings that the linear convergence rate of BFGS with $B_0 = L I$ surpasses that of $B_0 = \mu I$ in Corollary 4.2. In Corollary 4.2, we show that BFGS with $B_0 = L I$ could achieve the linear rate of $1 - 1/\kappa$ from the first iteration while BFGS with $B_0 = \mu I$ needs to run $d\log{\kappa}$ to reach the same linear rate.

Second, the transition to superlinear convergence for BFGS with $B_0 = \mu I$ typically occurs around $t \approx d$, as predicted by our theoretical analysis. Although BFGS with $B_0 = L I$ initially converges faster, its transition to superlinear convergence consistently occurs later than for $B_0 = \mu I$. Notably, for a fixed dimension $d=600$, the transition to superlinear convergence for $B_0 = L I$ occurs increasingly later as the problem condition number rises, an effect not observed for $B_0 = \mu I$. This phenomenon indicates that the superlinear rate for $B_0 = L I$ is more sensitive to the condition number $\kappa$, which corroborates our results in Corollary 6.5. In Corollary 6.5, we present that BFGS with $B_0 = L I$ needs $d\kappa$ steps to reach the superlinear convergence stage while this is improved to $d\log{\kappa}$ for BFGS with $B_0 = \mu I$.

Moreover, the performance of BFGS for $B_0 = I$ and $B_0 = c I$ are similar to the performance of BFGS with $B_0 = \mu I$. Notice that the initializations of $B_0 = I$ and $B_0 = c I$ are two practical choices of the initial Hessian approximation matrix $B_0$ that are commonly used.

**Different Convergence Stages**

We also attach table 1 in the uploaded file with summary of all our convergence results for (i) an arbitrary positive definite $B_0$, (ii) $B_0=L I$, and (iii)$B_0=\mu I$. The table contains all the convergence rates of three different convergence phases: (i) linear convergence phase with $1 - 1/\kappa$. (ii) parameter-free linear convergence phase and (iii) superlinear convergence phase with number of iterations required to achieve the corresponding stages.

---

### Decision · Program_Chairs · 2024-09-25

**Decision:**

Accept (spotlight)

**Comment:**

The paper studies BFGS with an inexact line search scheme satisfying the Armijo-Wolfe conditions.
1. On smooth strongly convex functions they show the sub-optimality is bounded by O( (1 - 1/\kappa)^t ) matching gradient descent. This improves on best-known the global convergence guarantees for BFGS where t is number of iterations.
2. With the added assumption that the Hessian is Lipschitz the paper shows that, after sufficient number of iterations faster convergence occurs. These results are less novel given non-asymptotic superlinear convergence for BFGS is already well-studied [37,38].

The closest related papers is Qiujiang Jin, Ruichen Jiang, Aryan Mokhtari [38]. [38] considers the same function classes but focuses on exact line search rather than satisfying the Armijo-Wolfe conditions. Satisfying the Armijo-Wolfe conditions is more realistic. The major theoretical gain over [38] is that on smooth strongly convex functions [38] only proves an O( (1 - 1/\kappa^{1.5})^t ) suboptimality bound whereas this paper shows a O( (1 - 1/\kappa)^t ) suboptimality bound.

Since the O( (1 - 1/\kappa)^t ) suboptimality bound is worse than the O( (1 - 1/\kappa^{1/2})^t ) suboptimality bound of AGD its not clear that O( (1 - 1/\kappa)^t ) is the best suboptimality guarantee for BFGS. This paper would have been much stronger if the authors had shown either a better suboptimality guarantee or a lower bound on the performance of BFGS to show that O( (1 - 1/\kappa)^t ) is tight.

Minor comments:
- [37] was published in mathematical programming in 2023 but you currently cite the arxiv version